# Towards understanding potential atmospheric contributions to abrupt climate changes: Characterizing changes to the North Atlantic eddy-driven jet over the last deglaciation

Heather J. Andres[1] and Lev Tarasov[1]

[1]Memorial University of Newfoundland, St. John's, NL, CANADA

**Correspondence:** Heather J. Andres (handres@mun.ca)

**Abstract.** Abrupt climate shifts of large amplitude were common features of the Earth's climate as it transitioned into and out of the last full glacial state approximately twenty thousand years ago, but their causes are not yet established. Mid-latitudinal atmospheric dynamics may have played an important role in these climate variations through their effects on heat and precipitation distributions, sea ice extent, and wind-driven ocean circulation patterns. This study characterises deglacial winter wind changes over the North Atlantic (NAtl) in a suite of transient deglacial simulations using the PlaSim earth system model (run at T42 resolution) and the TraCE-21ka (T31) simulation. Though driven with yearly updates in surface elevation, we detect multiple instances of NAtl jet transitions in the PlasSim simulations that occur within 10 simulation years and a sensitivity of the jet to background climate conditions. Thus, we suggest that changes to the NAtl jet may play an important role in abrupt glacial climate changes.

We identify two types of simulated wind changes over the last deglaciation. Firstly, the latitude of the NAtl eddy-driven jet shifts northward over the deglaciation in a sequence of distinct steps. Secondly, the variability of the NAtl jet gradually shifts from a Last Glacial Maximum (LGM) state with a strongly preferred jet latitude and a restricted latitudinal range to one with no single preferred latitude and a range that is at least 11° broader. These changes can significantly affect ocean circulation. Changes to the position of the NAtl jet alter the location of the wind forcing driving oceanic surface gyres and the limits of sea ice extent, whereas a shift to a more variable jet reduces the effectiveness of the wind forcing at driving surface ocean transports.

The processes controlling these two types of changes differ on the upstream and downstream ends of the NAtl eddy-driven jet. On the upstream side over eastern North America, the elevated ice sheet margin acts as a barrier to the winds in both the PlaSim simulations and the TraCE-21ka experiment. This constrains both the position and the latitudinal variability of the jet at LGM, so the jet shifts in sync with ice sheet margin changes. In contrast, the downstream side over the eastern NAtl is more sensitive to the thermal state of the background climate. Our results suggest that the presence of an elevated ice sheet margin in the south-eastern sector of the North American ice complex strongly constrains the deglacial position of the jet over eastern North America and the western North Atlantic as well as its variability.

# 1 Introduction

The last deglaciation encompassed a period of large-scale global warming of the Earth's surface climate, with regional patterns of millennial-timescale variability (Blunier et al., 1998; Shakun and Carlson, 2010; Clark et al., 2012). The Oldest Dryas (OD, 19-14.7ka BP, which includes Heinrich Stadial 1, 17.5-14.7ka BP), the Bølling-Allerød (B-A, 14.7-12.8ka BP), and the Younger Dryas (YD, 12.8-11.7ka BP) denote consecutive periods of stadial and interstadial conditions with rapid adjustments between them (Hammer et al., 1986; Grachev and Severinghaus, 2005; Clark et al., 2012). Signatures of these climate variations are present in the mid- to high-latitudes of both hemispheres with increasing amplitude toward the poles, although the signs tend to be anti-phased between the two hemispheres and of lower amplitude in the south (Shakun and Carlson, 2010). Such variability is also present in proxy indicators for variables other than temperature. Ocean circulation proxies indicate a deepening of the Atlantic Meridional Overturning Circulation (AMOC) and a northward shift in the northern boundary of the subtropical gyre with shifts from stadial to interstadial conditions in northern high latitudes (and vice versa) (e.g. Gherardi et al. (2009); Benway et al. (2010); Clark et al. (2012); Repschläger et al. (2015)). Abrupt shifts have also been detected in reconstructed lake levels and speleothem data in the subtropics and Indian and African monsoon regions, suggesting latitudinal shifts of the InterTropical Convergence Zone (ITCZ) (e.g. Jacob et al. (2007); Mohtadi et al. (2016)).

Proposed hypotheses explaining the presence of such variability during the deglaciation commonly centre around deepwater formation changes in response to freshwater anomalies in the regions where deepwater is formed (e.g. Rooth (1982); Broecker et al. (1985, 1989); Tarasov and Peltier (2005); Bradley and England (2008); Hu et al. (2010); Keigwin et al. (2018)). In simulations, abrupt reductions of the AMOC induced by hosing are successful at explaining the abruptness of cooling in the extratropical North Atlantic (NAtl), Nordic Seas, Arctic and Eurasia, the reduced precipitation over the NAtl and Europe, and the southward shift of the ITCZ (Kageyama et al., 2010, 2013). They are less successful at explaining the amplitude of temperature changes over Greenland and Europe during the last deglaciation (Clark et al., 2012), particularly when hosing amounts are constrained to realistic values (Kageyama et al., 2010).

Freshwater forcing may not be the only driver of the abrupt climate changes of the last deglaciation. Modern Earth System Models (ESMs) are now exhibiting abrupt climate changes of similar magnitude under slowly-varying or constant boundary conditions (Knorr and Lohmann, 2007; Peltier and Vettoretti, 2014; Zhang et al., 2014; Brown and Galbraith, 2016; Zhang et al., 2017; Klockmann et al., 2018), due to the bistability of the AMOC (Stommel, 1961; Broecker et al., 1985; Knorr and Lohmann, 2007; Zhang et al., 2014, 2017) and/or thermohaline instabilities involving the interactions of the ocean, sea ice and potentially atmosphere (Knorr and Lohmann, 2007; Dokken et al., 2013; Peltier and Vettoretti, 2014; Brown and Galbraith, 2016; Vettoretti and Peltier, 2016; Klockmann et al., 2018; Vettoretti and Peltier, 2018). Winter sea ice extent changes in model simulations alone are sufficient to reproduce deglacial January temperature changes in northwestern Europe (Renssen and Isarin, 2001), and changes to the surface ocean heat transports can affect the rates of deepwater formation (Lozier et al., 2010; Häkkinen et al., 2011; Muglia and Schmittner, 2015). Since low-level wind patterns over the Arctic, Greenland and NAtl help constrain winter sea ice extent in this region (Venegas and Mysak, 2000) and affect surface ocean heat transports there through the application of surface wind stresses (Lozier et al., 2010; Häkkinen et al., 2011; Li and Born, 2019), these winds may play

an important role in setting the conditions required for abrupt deglacial changes or be involved in the abrupt transitions themselves. However, in order to assess this potential, deglacial changes to lower-tropospheric winds must be first identified.

Of the two dominant features of mid-latitude atmospheric circulation patterns over the NAtl, the subtropical and eddy-driven jets, the eddy-driven jet has the largest presence in the lower troposphere, and thus the most potential to change wind stress and thereby ocean circulation. The eddy-driven jet (or polar front jet or jet stream) is a narrow band of fast, westerly winds that arises from the momentum-flux convergence of atmospheric synoptic-scale eddies (e.g. extratropical cyclones with lifespans of days) (Lee and Kim, 2003; Barnes and Hartmann, 2011). These baroclinic eddies are primarily created by the strong temperature gradients along the boundary between polar and tropical air masses and/or the wind shear on the flank of the subtropical jet (Lee and Kim, 2003; Justino et al., 2005). Since these two phenomena occur at different latitudes, the position of the eddy-driven jet depends on their relative strengths (Lee and Kim, 2003):

– under a weak subtropical jet as over the NAtl today, the eddy-driven jet tends to be distinct from the subtropical jet and located in the mid-latitudes. Its variability is dominated by latitudinal changes in the position of the jet.

– under a strong subtropical jet as in the North Pacific (NPac) today, the eddy-driven jet tends to lie along the poleward flank of the subtropical jet. Its variability tends to be dominated by fluctuations in strength rather than latitudinal position.

Additional eddy sources that localize the eddy-driven jet in longitude as well as latitude include the baroclinicity of land-sea boundaries or other surface temperature gradients, and the breaking of Rossby waves (see Box 1 of Cohen et al. (2014) for an overview of the phenomena affecting and affected by the eddy-driven jet). Due to the greater temperature contrasts between equator and pole or land and sea during a hemisphere's winter, the eddy-driven jet is stronger and more spatially localized during winter. These conditions make it more likely that the eddy-driven jet will impact the atmosphere/ocean system during winter. As such, we restrict our attention in this study to the northern winter, or DJF. However, summer changes to the winds and climate during the deglaciation are of greater importance to the rates and locations of ice sheet melt and retreat, so we will address those changes in a following study.

Since the eddy-driven jet's position depends on the characteristics of the eddy field, any changes to locations or rates of eddy production and decay will change the jet. However, the jet itself is a source of eddies in the form of midlatitude cyclones, or storm tracks. Thus, variations in the jet's position over time arise from a complex process of feedback interactions between changes to the eddies generated by the background climate, and changes to the eddies arising from the jet's response (Cohen et al., 2014). The type of variability depends on the latitudinal position of the jet. Eddy-driven jets located closer to the equator (and the subtropical jet) or the pole tend to vary in strength (pulse) due to dynamical limits provided by background wind conditions in these regions, while jets in the central midlatitudes tend to meander latitudinally (Barnes and Hartmann, 2011). Depending on the type of variability, the patterns of heat and moisture transported by the jet will be either concentrated to a narrow band of latitudes around the jet's mean position or spread over a large area.

Under LGM boundary conditions, with ice sheet cover extending to the mid-latitudes over North America (NAmer) and over Fennoscandia (Dyke, 2004; Hughes et al., 2016), both the position of the ITCZ (and presumably the subtropical jet) (Arbuszewski et al., 2013) and the regions of baroclinicity are different from present-day. The topographic and thermal barriers

presented by the ice sheet complex over NAmer, in particular, alter the stationary wave field, the pattern of transient eddy production and decay, and the locations of baroclinic zones around the NH (Rind, 1987; Cook and Held, 1988; Roe and Lindzen, 2001a, b; Justino et al., 2005; Li and Battisti, 2008; Lofverstrom et al., 2014; Merz et al., 2015). These effects have their strongest manifestation just downstream of the North American Ice Sheet complex (NAIS), over the NAtl. However, the influences on the western and eastern sides of the NAtl have been found to differ in atmospheric simulations: the position of the western side of the NAtl jet is more affected by ice sheet orography (though the relevant aspect of orography has not been identified prior to our study), and the position of the eastern side of the jet is primarily influenced by stationary and/or transient eddies (Kageyama and Valdes, 2000; Lofvestrom et al., 2016). Regional jet position in turn will likely affect different aspects of the climate system. The behaviour of the jet over the eastern NAtl may strongly affect the routing of storm tracks and the jet's contribution to poleward atmospheric heat transport. The position of the western side of the jet over eastern NAmer is of particular importance to the wind-driven ocean circulation through Sverdrup transport (Li and Born, 2019). Given the above and that there is no reason to expect equal and synchronous changes in eastern and western NAtl jet positions over deglaciation, both positions should be diagnosed. Following the convention in recent literature, we also use the jet's mean latitudinal position and its east-west difference in latitudinal position (differencing regions marked by black boxes in Figure 4), or its tilt, as additional metrics.

NAtl eddy-driven jet characteristics in LGM simulations for multiple models from the Paleoclimate Modelling Intercomparison Project (PMIP) 3 show little consistency, except that the jets are all stronger and less variable than in corresponding preindustrial simulations (P. Hezel, personal communication). However, in simulations based on the NCAR family of models (CCSM3, CCSM4 and their components), the NAtl eddy-driven jet at LGM follows a less tilted path and is stronger and less latitudinally-variable than at present (Li and Battisti, 2008; Lofverstrom et al., 2014; Merz et al., 2015). The reasons for these jet characteristics remain unexplained. Nevertheless, changes to the path and variability of the jet are expected to result in changes to the distributions of heat and precipitation over Western Europe during this period.

Sometime during the last deglaciation, the pattern and variability of simulated midlatitude winds change, especially in response to the ice sheet topography changes (Cook and Held, 1988; Rind, 1987; Justino et al., 2005; Li and Battisti, 2008; Merz et al., 2015). However, there is little information as to whether these changes occurred at the time of the abrupt climate transitions. In the TraCE-21ka deglacial simulation (Liu et al., 2009; He, 2011; Liu et al., 2012), the NAtl jet exhibits two characteristic states: a strong, stable, zonal glacial jet and a weaker, latitudinally-variable, tilted interglacial jet (Lofverstrom and Lora, 2017). The transition from the one jet state to the other is abrupt and coincident with the separation of the Cordilleran and Laurentide ice sheets at 13.89 ka BP (Lofverstrom and Lora, 2017). A jet shift at 13.89ka BP lies during the middle of the B-A, which would rule out its playing an important, proximal role in triggering abrupt climate transitions during the last deglaciation. However, it is difficult to assess the representativeness of the timing of changes associated with ice sheets in the TraCE-21ka experiment, since that simulation only enacted ice sheet and corresponding meltwater changes via step functions applied at irregular intervals (He, 2011). Additionally, the ice sheet reconstruction used, ICE-5G, over-predicts the height of the Keewatin dome with respect to more recent reconstructions (Tarasov et al., 2012; Peltier et al., 2015) and is inconsistent with present-day uplift data (Tarasov and Peltier, 2004).

In summary, previous work has shown that lower-tropospheric winds over the North Atlantic have the ability to alter sea ice extent and surface ocean circulations in a manner that can reproduce abrupt climate changes detected over Greenland. Simulations have shown that these winds do change characteristics from the start to the end of the last deglaciation, although only a single study has examined how these changes may have progressed in time. Due to the manner in which boundary conditions were updated in the latter study as well as problems with their chosen ICE5-G boundary conditions, it is difficult to assess whether the timing of the wind changes that occurred are actually representative of past changes. Therefore, this study diagnoses the changes undergone by the NAtl eddy-driven jet from the LGM to the preindustrial period in multiple transient deglacial experiments using boundary conditions that are updated every simulation year following the specifications of the PMIP4 (Ivanovic et al., 2016). These simulations are performed using a modified version of the Planet Simulator (PlaSim) version 16, an Earth System Model with a primitive equation atmosphere and simplified physics parametrizations (Fraedrich, 2012; Lunkeit et al., 2012). As such, these simulations can help elucidate whether atmospheric dynamical changes have the potential to play important roles in the abrupt climate changes of the last deglaciation and provide an important comparison against PMIP4 deglacial studies performed using more complex models.

## 2 Methods

The experiments discussed in this paper consist of an ensemble of four transient simulations of the last deglaciation generated using a modified version of PlaSim, as well as a suite of sensitivity studies (FixedOrbGHG, FixedGlac, PDTopo and DarkGlac). All deglacial simulations are initialized from the same initial conditions, which are derived from year 2567 of an equilibrated LGM spin-up started from present-day. Ensemble members differ by the radiation parameter set used, all of which were identified as optimized configurations for reproducing radiative fields during the preindustrial period. Further details of the tuning procedure are available in Supplemental Section S1.0.1. A subset of the same boundary conditions are applied in all of the fully transient experiments, which are described in Table 1.

The goals of the sensitivity experiments are to isolate the contributions of ice sheet height, ice sheet albedo, and orbital and GHG forcings to the jet changes we detect, by fixing one forcing at a time and running otherwise identical simulations over the entire deglaciation. Each sensitivity experiment was repeated for all tuning configurations employed in the fully-transient experiments. Note that the results of the DarkGlac (artificially low ice albedo) experiments were not significantly different than the fully-transient runs due to the reflectivity of snow on top of the dark ice, so their results are not discussed.

Characteristics of PlaSim are described in Section 2.1, and the boundary conditions are described in Section 2.2. Finally, the model's performance at simulating LGM climate with these boundary conditions is described in Section 2.2.5.

### 2.1 Planet Simulator

PlaSim consists of the Portable University Model of the Atmosphere (PUMA) wet primitive equation atmosphere model, the Simulator for Biospheric Aspects (SIMBA) dynamic vegetation model, a zero-layer thermodynamic sea ice model, a slab-ocean model and the Large-Scale Geostrophic (LSG) ocean model (Fraedrich, 2012; Lunkeit et al., 2012). PUMA is an atmospheric

**Table 1.** Characteristics of numerical simulations presented in this study. The time-varying forcings are described in section 2.2, and are either transient (T) or fixed (F). Topography includes ice sheet topographic variations as well as corresponding changes to surface roughness and land mask. GHG represents greenhouse changes enacted through effective $CO_2$ concentrations.

| Name | Timespan | | Time-varying Forcings | | | |
|------|----------|---------|------------|-----------------|---------|---------|
|      | Start (BP) | End (BP) | Topography | Land ice albedo | GHG | Orbital |
| FullyTrans1-4 | 21000 | 0 | T | T | T | T |
| FixedOrbGHG1-4 | 21000 | 0 | T | T | F (LGM) | F (LGM) |
| FixedGlac1-4 | 21000 | 0 | F (LGM) | F (LGM) | T | T |
| PDTopo1-4 | 21000 | 0 | F (past1000) | T | T | T |
| DarkGlac1-4 | 21000 | 0 | T | F (past1000) | T | T |

general circulation model (GCM) whose dynamical core is based on the primitive equations. The primary simplifications in this component of PlaSim are found in the physical parametrizations incorporated in the model: for example, carbon dioxide is

20 the only greenhouse gas whose radiative effects are considered and the radiative transfer scheme is much simpler (and thereby much faster) than that used in current state of the art GCMs. Herein we use 10 vertical levels at a spectral resolution of T42 (approximately 2.8°x2.8°), which has been previously shown to be sufficiently high to resolve phenomena of interest to the eddy-driven jet (Barnes and Hartmann, 2011; Lofverstrom and Liakka, 2018) while enabling fast enough model run times to make multiple deglacial experiments feasible. The dynamical atmospheric solutions are generated in spectral space, while the

25 remaining calculations (e.g. phase changes, heat exchange with the land, sea ice or slab ocean, and any changes in those sub-components) occur in real space on a Gaussian grid with 64 latitude points and 128 longitude points. The only exception to this is LSG, which is run at 2.5°x5° horizontal resolution.For every atmospheric time step, the land, sea ice and slab-ocean components are called sequentially using the same grid and land-sea mask (a schematic is available in Supplemental Figure S1).

Rather than specifying a fixed deep-ocean heat flux to the mixed layer ocean, PlaSim estimates these fluxes by executing LSG every 32 atmospheric time steps (equivalent to 4.5 days). LSG is a three-dimensional, global, ocean general circulation model that solves the primitive equations implicitly under assumptions of large spatial and temporal scales (Maier-Reimer et al., 1993). This formulation permits stable solutions on longer time steps than other components of PlaSim with the trade-off that it filters out gravity waves and barotropic Rossby waves (Maier-Reimer et al., 1993). Since the time steps of LSG are so long, a slab-ocean model is used as an intermediary between LSG and the rest of the model in order to allow the ocean to respond to abrupt or short-lived phenomena. The slab-ocean model is fixed to a 50m depth, which corresponds to the depth of the top layer of LSG. Thus, at the start of an LSG integration, fields in the top layer of LSG are initialized to those from the slab-ocean

5 model, and heat fluxes and wind stress fields are read from the sea ice and atmosphere components, respectively. The LSG

integration is performed, and a spatial map of differences between the mixed-layer temperature at the end and the start of the LSG time step are calculated. These temperature differences are used to define a map of deep-ocean heat fluxes, which are subdivided by the number slab-ocean time steps before the next LSG integration and applied as bottom boundary conditions to the slab-ocean model. Thus, under constant atmospheric conditions, the slab ocean model relaxes toward the LSG solution. Under changing atmospheric conditions, the surface component of the ocean will tend toward a mixture of the LSG solution and a thermal response to the surface forcing.

For the experiments performed in this study, the PlaSim model was modified to allow time-varying boundary conditions. Changing boundary conditions include land-sea mask (implemented for all model components except LSG, whose land-sea mask remains at present-day conditions throughout), topography, ice mask, orographic component of surface roughness, orbital configuration and greenhouse gas concentration. Note that although the land mask was altered consistently with changing ice sheet mass, no corresponding salinity anomalies were introduced to the oceans. However, surface runoff routing was recalculated every time the ice sheet configuration changed.

Further details about PlaSim are available in Supplemental Section S1.

## 2.2 Boundary Conditions

PlaSim boundary conditions are updated every simulation year. However, for all of the transient simulations presented, the forcings are accelerated by a factor of ten. This acceleration was not found to alter the main conclusions of this study when tested against a single unaccelerated run, but it is expected to lengthen the apparent timescales of processes. For example, if the ocean's mixed layer takes approximately 30 simulation years to fully adjust to a change in atmospheric boundary conditions, it will appear to take 300 years from the perspective of forcing changes. More particulars about the differences between the accelerated and unaccelerated runs are presented in Appendix section A2. Where the time between boundary condition updates is shorter than the temporal resolution available for a boundary condition dataset, the boundary conditions are linearly interpolated in time. For ice sheet and land-sea masks, the interpolated values are then rounded to zero or one with a cutoff of 0.5.

### 2.2.1 Land-sea mask, land topography, land-ice mask

Changes in land-sea mask, land topography and land-ice mask are derived from the GLAC1-D deglacial chronology generated with the three-dimensional glacial systems model (GSM) that includes a thermomechanically-coupled ice sheet model, visco-elastic bedrock response, and various other components (cf Tarasov et al., 2012; Briggs et al., 2013, and references therein). The North American (Tarasov et al., 2012) and Eurasian components (Tarasov et al., 2014) are obtained from large ensemble Bayesian inversions against large sets of geophysical and geological observations. The Antarctic component (Briggs et al., 2014) is a best scoring run (against data constraints) from a large ensemble and the Greenland component (Tarasov and Peltier, 2003) is an older hand-tuned model. These four components have then been post-processed to create a gravitationally self-consistent global deglaciation chronology with a temporal resolution of 100 years. These data are interpolated spatially to the

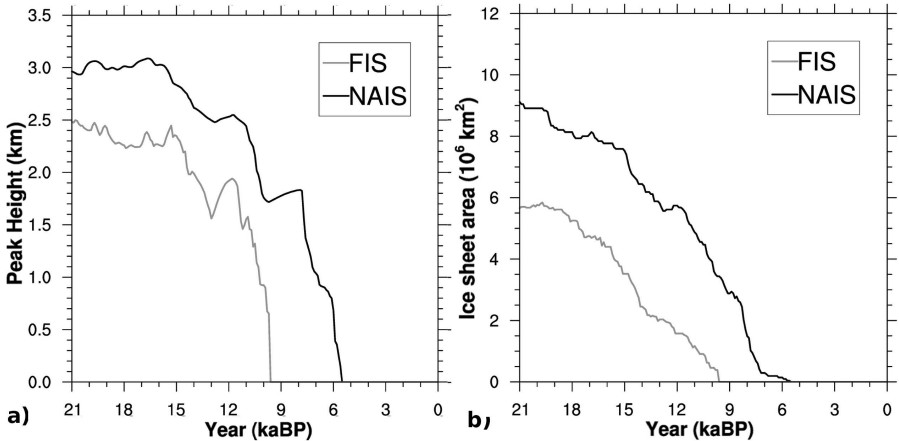

**Figure 1.** a) Peak elevation in ice sheet-covered areas (bedrock elevation plus ice sheet thickness) and b) ice sheet area for the component of the Laurentide ice sheet and Eurasia (FIS).

model grid, with land-sea mask defined so the topography of ocean grid cells lies below the contemporaneous sea level. The data are also linearly interpolated in time in order to provide updates every simulation year (i.e. every 10 forcing years).

Time series of ice sheet areas and peak elevations for North America (east of the Rockies) and Fennoscandia are plotted in Figure 1. In both the North American ice sheet complex (NAIS) and Fennoscandian ice sheet complex (FIS), ice sheet margins begin retreating before peak heights change noticeably. The ice sheet areas decrease approximately linearly, with pauses for both the NAIS and the FIS during the Younger Dryas (approximately 13 to 12ka BP). In contrast, the rate of peak height decrease accelerates with time. Thus, the bulk of peak height changes occur from 12ka BP to 6ka BP. The FIS completely disappears by 9 ka, while a significant Labrador ice dome is present at 8 ka (and largely dissipates by 6 ka).

### 2.2.2 Surface roughness

The component of surface roughness due to orography is calculated from the topography data described above via a similar method to what was used for the default present-day roughness provided with the model (Tibaldi and Geleyn, 1981). For each T42 grid cell, the roughness is equal to the variance of all 0.5°x0.5° ice sheet grid cells contained within it divided by an effective higher-resolution grid cell length. This calculation is performed by first conservatively remapping the elevation and the square of the elevation from the higher-resolution grid to the lower-resolution grid. The variance is then the difference between the square of the remapped elevation and the remap of the squared elevation. The effective higher-resolution grid cell length is the square root of the area of the T42 grid cell divided by the number of higher-resolution grid cells per T42 grid cell (taken here to be the ratio of the total number of grid cells globally in each grid).

### 2.2.3 Orbital configuration

Orbital parameters are calculated internally within the model given an orbital forcing year. The equations follow Berger (1978), and the orbital year is updated every simulation year.

### 2.2.4 Greenhouse gas concentration

The only greenhouse gas explicitly handled in PlaSim is carbon dioxide ($CO_2$). In this study, other trace gases are accounted for by defining an effective $CO_2$ concentration value that yields radiative changes equivalent to the combination of $CO_2$, nitrous oxide ($N_2O$) and methane ($CH_4$). Effective $CO_2$ concentrations were defined with respect to reference $CO_2$ concentrations at year 22.3ka BP using the equations in Ramaswamy et al. (2001). This reference year was chosen, because both $N_2O$ and $CH_4$ values were at a relative minimum at that time, and this year precedes the period of interest for our study. Data for $CO_2$, $N_2O$ and $CH_4$ concentration changes over the deglaciation are consistent with the prescriptions of the PMIP4 Deglacial experiment except for $CO_2$, which uses an older dataset (see Ivanovic et al. (2016) and data sources Luthi et al. (2008) Meinshausen et al. (2017), and Loulergue et al. (2008)). Effective $CO_2$ concentration values are updated every simulation year, and are linearly interpolated in time between available data points as needed.

### 2.2.5 Model Evaluation

We compare the climate conditions during the first and last century of the fully-transient PlaSim simulations (corresponding to forcing years 21-20ka BP and 1ka BP to 1950CE, respectively, due to acceleration) to the results of LGM and past1000 experiments in the Climate Modelling Intercomparison Project (CMIP) 5. Only CMIP5 simulations that include experiments using the same model configuration for both LGM and past1000 are included here, as tabulated in Supplementary Table S1.

In Figures 2 and 3, near-surface temperatures (T2m) and sea ice concentrations are plotted for ensemble averages of the first and last centuries of the fully-transient PlaSim simulations. Hatching in Figure 2 indicates where the PlaSim ensemble average T2m lie within the range spanned by the CMIP5 multi-model ensemble members (interpolated to the same resolution). Red and gold lines in Figure 3 identify CMIP5 multi-model ensemble maximum and minimum sea ice extents during the same season, respectively. The surface climate is colder and sea ice more extensive during LGM in the PlaSim simulations than the CMIP5 multi-model ensemble members for both DJF and JJA. This is particularly true in the northern high latitudes, where temperatures over the Arctic Ocean lie below all CMIP5 ensemble members and sea ice is anomalously extensive in the NPac and on the eastern side of the NAtl. In contrast, Antarctic temperatures lie within the range of CMIP5 models, and Southern Ocean sea ice is close to the CMIP5 maximum.

Most of these disagreements are resolved by the past1000, when PlaSim-predicted temperatures and sea ice concentrations lie within the CMIP5 range in most regions. However, temperatures remain anomalously cold over the Arctic during DJF and over oceans in the midlatitudes in both hemispheres and seasons, and sea ice extent exceeds the CMIP5 maximum in the Southern Ocean during JJA. These differences are discussed in more detail in Appendix section A1.

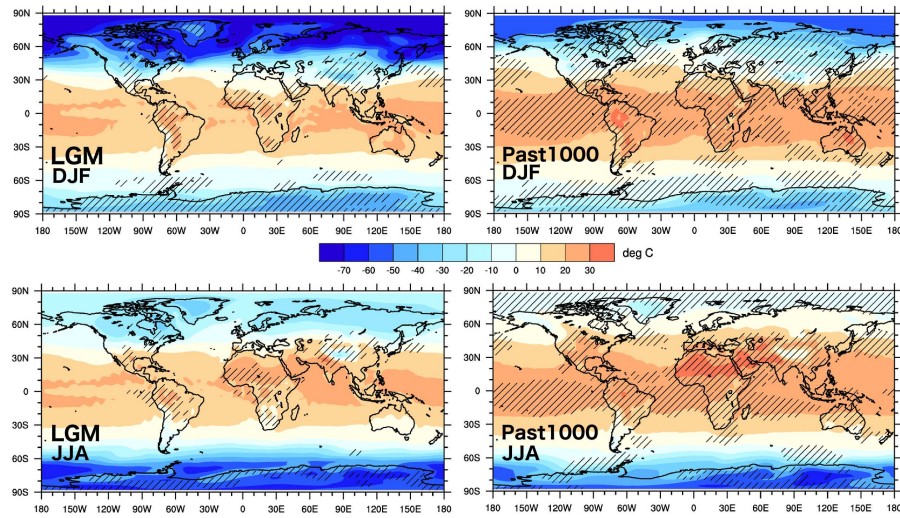

**Figure 2.** 2m temperatures averaged over all PlaSim ensemble members for indicated seasons and periods. Hatching indicates where the PlaSim ensemble average values lie within the range of CMIP5 models.

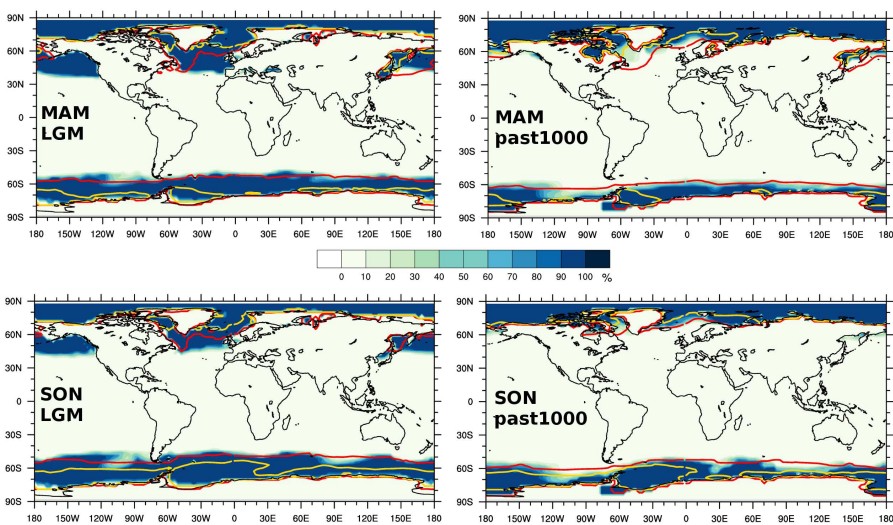

**Figure 3.** Maps of sea ice concentration averaged over all PlaSim ensemble members for indicated seasons and periods. The CMIP5 multi-model maximum extents denoted by the 15% concentration line are plotted for corresponding seasons and experiments in the red lines. The CMIP5 multi-model minimum extents are plotted in the gold lines.

Lower-level zonal winds from both PlaSim and CMIP5 averages are plotted in Figure 4 for DJF with the sea ice margin outlined in blue. Zonal winds during the past1000 in the PlaSim simulations lie within the range covered by the CMIP5 multi-model ensemble except over North America and central Eurasia, where winds are stronger in PlaSim. Also, the NPac jet is

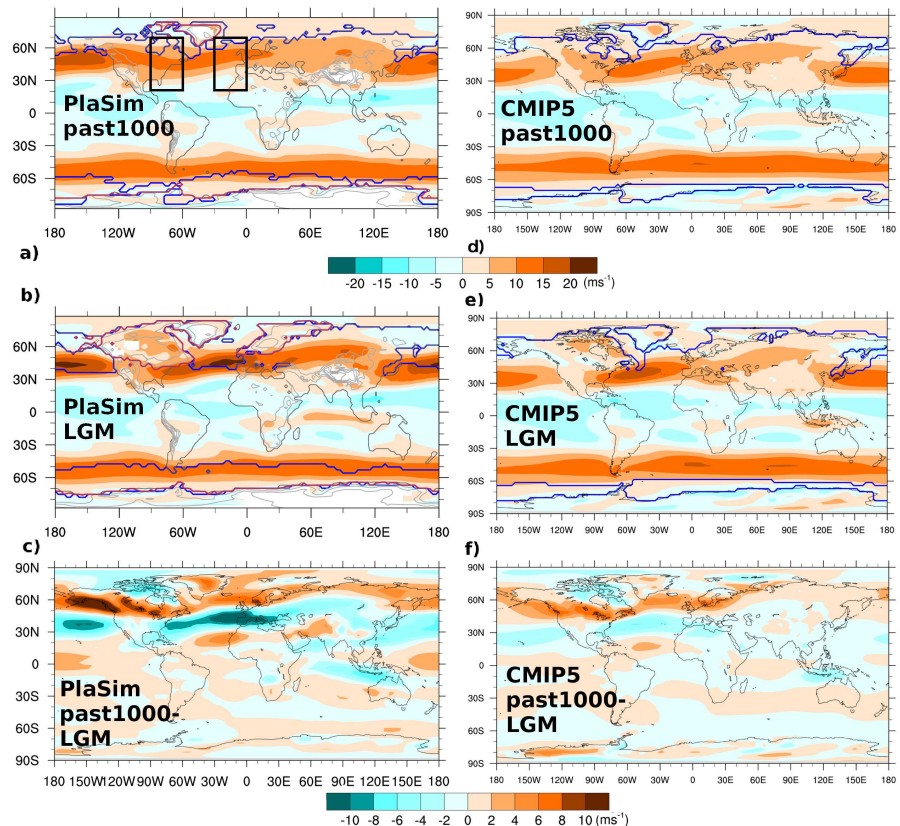

**Figure 4.** Zonal winds averaged over 700 to 925hPa and averaged over all PlaSim ensemble members or CMIP5 multi-model ensemble members for DJF and indicated periods. Where the land surface impinges on the vertical range, winds are interpolated to not bias the vertical average. Further discussion can be found in Section 4. Differences between climatologies for these two periods are in the bottom row. Blue lines mark the outline of the 15% concentration line for sea ice, and purple lines mark the land ice mask in PlaSim simulations. Black boxes outline the regions over which the western and eastern components of the NAtl jet are evaluated.

displaced further north and is more tilted in the PlaSim past1000 simulations. For the LGM, northern midlatitude zonal winds from the PlaSim simulations are stronger than those from the CMIP5 multi-model ensemble. Also, the strongest winds of both
5 the NAtl and the NPac jets are shifted further east toward the the eastern margins of their respective ocean basins. We speculate that this eastern shift is connected to the much more southern extent of sea ice on the eastern side of the NAtl, as was found in CAM3 simulations forced by present-day ice sheets with LGM sea surface temperatures and sea ice extent (Lofvestrom et al., 2016). In spite of these specific differences, the pattern of wind changes from the LGM to the past1000 are similar (but differ in magnitude) between the CMIP5 and PlaSim runs (including during JJA, not shown).

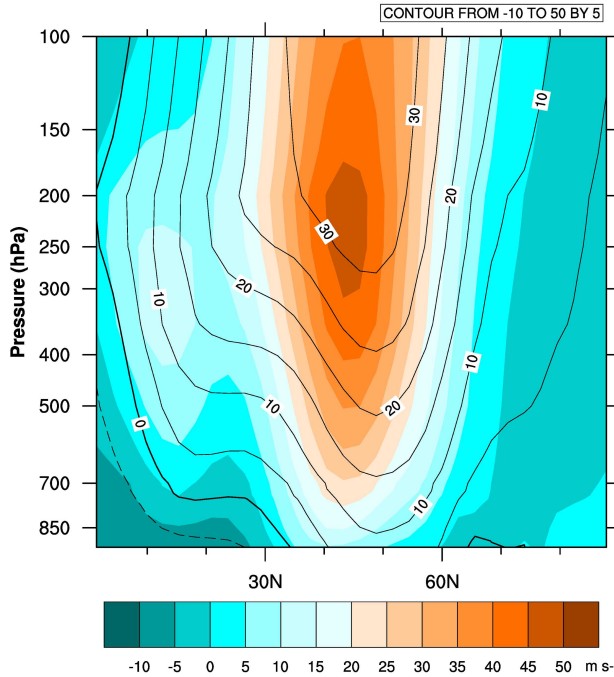

CONTOUR FROM -10 TO 50 BY 5

-10 -5 0 5 10 15 20 25 30 35 40 45 50 m s-1

**Figure 5.** Ensemble average of zonal wind profiles for the NAtl for LGM in coloured contours and past1000 in contour lines.

## 3 Results

### 3.1 Differences between LGM and past1000 climates

We begin by comparing in more detail the characteristics of the NAtl eddy-driven jets during LGM and past1000. As in Woollings et al. (2010), the algorithm used to detect the latitude of the eddy-driven jet for the remainder of this analysis finds the location of maximum zonal winds averaged over the NAtl basin within 15°N to 75°N and over atmospheric levels 700hPa to 925hPa. Where ice sheet elevation exceeds 925hPa, we interpolate the winds to these levels to not bias the resulting average. This choice does not alter the conclusions of our study substantially and is discussed in further detail in Section 4. Unlike Woollings et al. (2010), jet latitudes are defined from monthly data (without low-pass filtering) over longitudes of 90°W to 0°W. Unless otherwise indicated, the monthly jet latitudes are aggregated over 10 consecutive DJF periods to generate jet latitude frequencies. NAtl eddy-driven jet tilt is defined to be the difference between the jet latitudes calculated in the same manner from zonal winds averaged over 90°W to 60°W and 30°W to 0°W, denoted the western and eastern regions of the jet, respectively, and outlined by black boxes in Figure 4.

The NAtl eddy-driven jet is stronger, narrower, and shifted equatorward at LGM compared to past1000 in these simulations according to wind profiles in Figure 5 and histograms for lower-level jet latitudes in Figure 6. The equatorward-shifted position of the NAtl eddy-driven jet at LGM is not due to a strengthening of the subtropical jet, as the LGM subtropical jet is weaker

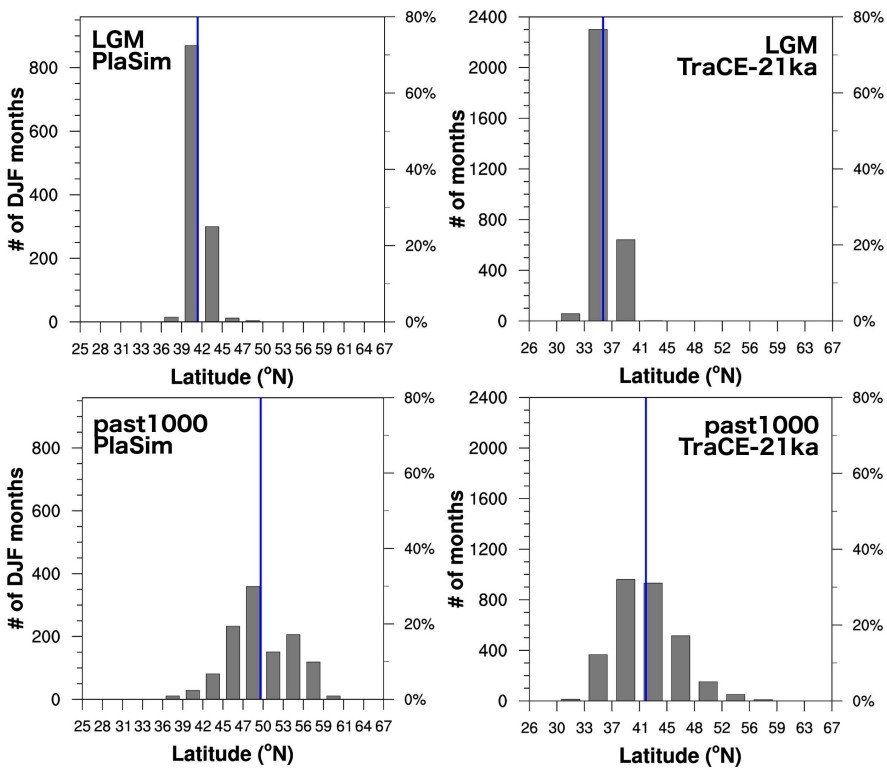

**Figure 6.** Histograms of latitudes corresponding to peak NAtl zonal winds for all PlaSim ensemble members (left column) and the TraCE-21ka data (right column) during indicated periods. Monthly jet latitude statistics are aggregated over 100 simulation years and four ensemble members for PlaSim and 1000 simulation years for the TraCE-21ka simulation. Vertical blue lines indicate mean jet latitudes for the time period.

10   and also shifted equatorward. The mean eddy-driven jet latitude shifts from 41°N to 50°N (with a grid cell length of 2.8°) from LGM to the past1000. A shift of similar size is observed on both the western and eastern regions of the eddy-driven jet (Figure 7), so there is little change in mean tilt values between these two periods (Figure 8). This result stands in contrast with results from Li and Battisti (2008), and Lofverstrom et al. (2014), where at LGM the NAtl eddy-driven jet is less tilted than during the preindustrial or present period. This issue will be discussed in more detail in Section 4.

   Although the range of jet latitudes broadens from the LGM to the past1000, the increase is asymmetrical. The lowest latitude occupied by the jet remains mostly unchanged between these two periods, while the highest latitude occupied by the

5    jet increases by 11°. This asymmetry is consistent with the subtropical jet providing a dynamical limit on the southernmost position of the eddy-driven jet (Barnes and Hartmann, 2011). Such asymmetry is apparent on both sides of the jet, although there is a larger northward increase in the range of jet latitudes over the western region than the eastern region. Thus, the range of jet tilts (downstream minus upstream latitude) increases from LGM to past1000 slightly more in the negative direction.

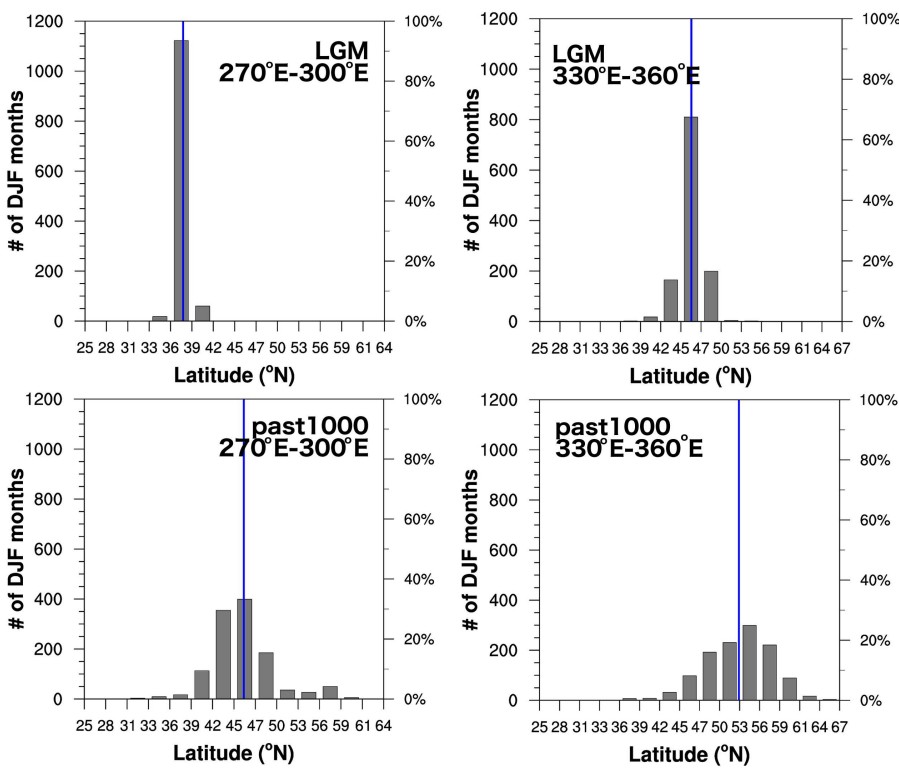

**Figure 7.** Histograms of latitudes corresponding to peak NAtl zonal winds for all PlaSim ensemble members over the western and eastern regions of the jet during indicated periods. Vertical blue lines indicate mean jet latitudes for the time period.

Changes in the NAtl eddy-driven jet from LGM to past1000 in the PlaSim simulations are compared against changes in the only publicly-available fully-transient, coupled deglacial simulation (TraCE-21ka, Liu et al. (2009); He (2011); Liu et al. (2012)) in the right columns of Figures 6 and 8. Jet histograms for TraCE-21ka are calculated in the same way as for the PlaSim simulations, except that the data are binned according to the lower resolution of this simulation (T31, equivalent to 3.7° of latitude). Although the mean jet positions differ between the two datasets, the changes from the LGM to the past1000 are quite similar between TraCE-21ka and the PlaSim ensemble. In TraCE-21ka, the mean jet latitude shifts poleward by approximately 6° (less than a grid cell smaller than the change in the PlaSim ensemble). Also, the minimum jet latitude remains unchanged, and the maximum jet latitude increases by 18°.

Differences between the PlaSim ensemble and the TraCE-21ka simulation primarily arise with respect to the jet tilt. The mean jet tilt increases by approximately 2°. Note that this value is less than that calculated in Lofverstrom and Lora (2017) from TraCE-21ka data (between 3° and 4°), but they calculated jet tilt from upper-tropospheric winds and different longitude ranges in the western and eastern regions of the NAtl jet (270-300°E and 330-360°E in this study versus 280-290°E and 340-350°E). However, either value of jet tilt change in the TraCE-21ka simulation is larger than that detected in the PlaSim simulations, and asymmetries in the increase in the range of jet tilt are opposite for these two datasets. Examining the eastern

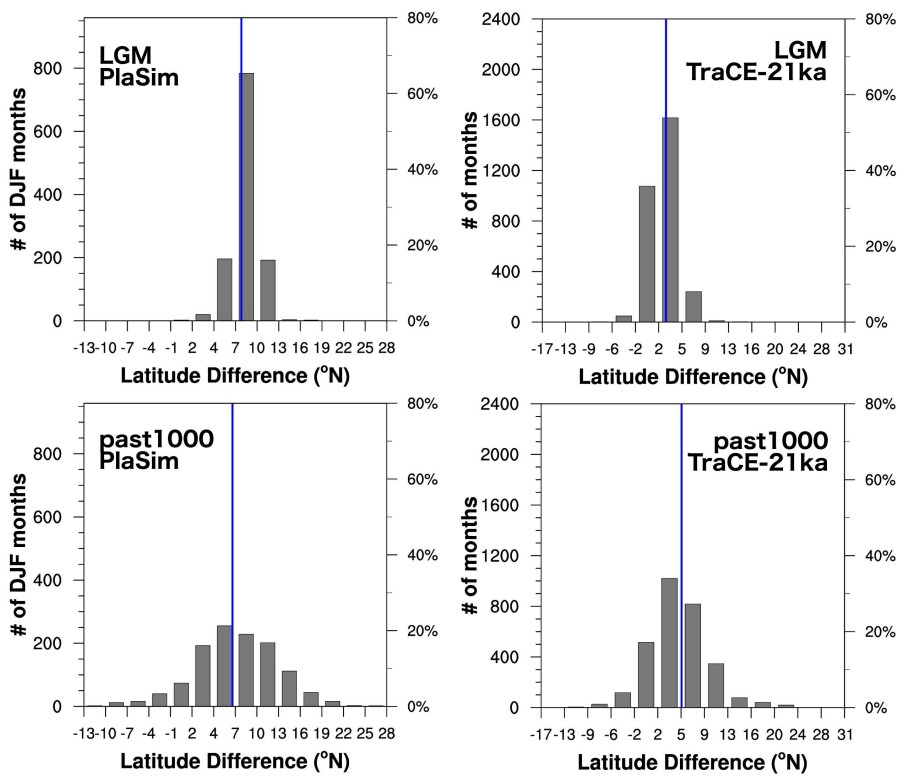

**Figure 8.** Histograms of jet tilt for the PlaSim ensemble (left column) and TraCE-21ka fully-transient simulation (right column) over indicated periods. Vertical blue lines mark the mean jet tilt for each period.

and western sides of the jet separately, the distributions in both regions change more similarly from LGM to past1000 in TraCE-21ka than in the PlaSim simulations (see Supplemental Figure S2). Also, there is no evidence of jet bimodality in any of the regions for the TraCE-21ka experiment, which may be due to the jet's being located further south.

Having shown that there are changes in both the jet location and its distribution from the LGM to the past1000 in the PlaSim simulations, and that these changes are similar to those occurring in the TraCE-21ka experiment, we now analyse how these changes arise in time.

## 3.2 Transitioning from LGM to past1000

The motivation of this study is to assess the potential impact of atmospheric changes over the NAtl on the abrupt deglacial changes detected in Greenland ice cores, so we start by discussing climate changes over Greenland. None of the transient simulations presented here produce abrupt transitions between stadial and interstadial conditions over Greenland at the historical times of the OD, B-A, or YD. Instead, the accelerated runs all show a single abrupt increase in Greenland temperatures (not shown) that occurs at different times for each simulation between 12ka BP and 5ka BP. This appears to be a manifestation of

internal variability of the model, as the timings of the changes do not coincide between ensemble members and an identically-forced simulation without acceleration exhibits five such abrupt changes over the course of the deglaciation. Further details about this phenomenon will be discussed in another paper.

Due to the absence of abrupt climate changes over Greenland, we conclude that the changes in the position, tilt and variability of the NAtl eddy-driven jet are not sufficient on their own to generate large-amplitude, abrupt climate changes in PlaSim. It may be that feedbacks between the atmosphere, ocean, land ice and sea ice that are not captured in the simulations here are important in abrupt changes. For example, one very plausible process that is missing from PlaSim is the effect of winds on sea ice. Furthermore, this absence of simulated abrupt climate change does not rule out the possibility that the discerned atmospheric dynamical changes were important to historical abrupt climate changes through their controls on the background climate state. Thus, we characterize the atmospheric changes present in the accelerated transient simulations and leave further assessments of their implications for future work.

Deglacial changes to NAtl jet distributions are presented via frequency plots that use colour to illustrate the percentage of months in ten successive winter seasons (DJF) that the jet occupies a given latitude band or tilt plotted as a function of latitude and time. Ten years of winters are chosen as a balance between providing sufficient statistics (30 months) to be able to characterise the distribution of jet characteristics and a desire for high temporal resolution to assess the abruptness of changes. Only ensemble statistics are presented, as deglacial changes over time occurred similarly in all ensemble members. Frequency plots for jet latitude and tilt changes over the deglaciation for each ensemble member individually can be found in Supplemental Figures S3 and S4.

As discussed in Section 3.1, the low-level, NAtl jet is situated further south and exhibits a narrower range of latitudinal variability in the FullyTrans runs at the start of the deglaciation compared to its end. In Figure 9, peak winds occur at the same latitude for more than two-thirds of the months from 21ka BP to approximately 19ka BP, whereas the jet occupies no latitude for more than 40% of the time from 8ka BP onward. The two different types of deglacial jet changes (the latitudinal shift and the change in the shape of its distribution) occur separately over the deglaciation. In the first type of change, the median jet latitude shifts northward three times over the deglaciation in all ensemble members. The shifts occur around 19ka BP, 14ka BP and at 11.0ka BP in all simulations, but the abruptness is not the same for all shifts. The first transition is clearly gradual, as the jet spends nearly a century of simulation (a millennium of forcing) with its time nearly equally split between two grid cells before shifting to spending most of its time at the more northern latitude. In contrast, the third transition occurs within a decade of simulation (a century of forcing) and shows little evidence of such mixed states.

In the second type of jet change evident in Figure 9, the frequency of time that the NAtl eddy-driven jet spends at its median latitude decreases, and the range of jet latitudes increases. This reduction in frequency begins after 11ka BP and is completed by 8ka BP, but its timing differs between ensemble members (see Supplemental Figure S3). Note that the development of an isolated northern branch of mean jet latitudes sometime between 15 and 14ka BP in Figure 9 does not indicate that winds passing south of the ice sheet are undergoing large latitudinal shifts. Rather, it is an artifact due to the strongest zonal winds being identified in the northern branch of a split jet stream over North America. Although this split jet stream is present from

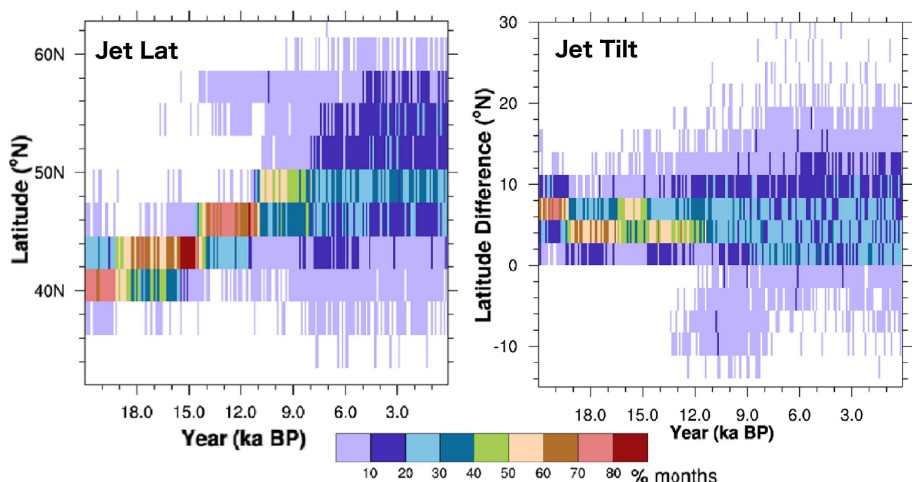

**Figure 9.** Frequency maps of NAtl mean jet latitudes and tilt averaged over 700hPa to 925hPa and aggregated over 10 successive winter seasons and all ensemble members of the FullyTrans experiment. Winds are averaged over 270°E to 360°E when calculating jet latitude, and tilt is defined as the difference in latitudes calculated over 330°E to 360°E and 270°E to 300°E. Frequencies represent the percentage of months (out of a total of 120 months) that the jet was identified at a particular latitude, where each latitude bin has a width of 2.8° at T42.

25   LGM (see Figure 4b), the northern wind branch is weaker than the southern branch until this time, so the jet detection algorithm does not identify the jet there. This issue is discussed in more detail in Section 3.3 and Supplemental Section S2.

    Ensemble statistics for the NAtl jet tilt in the FullyTrans runs in Figure 9 show oscillations between the most common jet tilt of LGM and a more zonal configuration. These changes in jet tilt reflect the very different behaviours with time of the western (270°E to 300°E) and eastern (330°E to 360°E) sides of the NAtl eddy-driven jet, which are shown in Figures 10 and

30   11. Nevertheless, by the end of the deglaciation, the jet on both its western and eastern sides has shifted northward by a similar amount, leading to little net change in jet tilt.

    The western side of the jet over eastern NAmer is very focussed and narrowly-distributed, with more than 80% of the winter months spent at the same latitude for most of the deglaciation. The most commonly-occupied latitude shifts northward twice in this region, at 19.3ka BP and 14.6ka BP, and each shift is completed within a decade of simulation. The timing of these transitions match the first two (more gradual) shifts in the basin-averaged jet and are consistent with the historical timings of the OD and B-A. They also correspond to reductions in the jet tilt. The transition away from a single, commonly-occupied jet latitude over eastern NAmer occurs more abruptly than for the jet as a whole at 10.8ka BP.

    In contrast, in Figure 11 the eastern side of the jet over the eastern NAtl is less focussed than the west, with the jet occupying

5   its most common latitude 50 to 70% of the time. There is a single, gradual, northward shift in the eastern region, occurring between 16 and 15ka BP. This change is a little later than the time of increasing jet tilt. The eastern side of the jet moves away from any single, commonly-occupied latitude after 11.4ka BP, near the historical end of the YD and start of the Holocene.

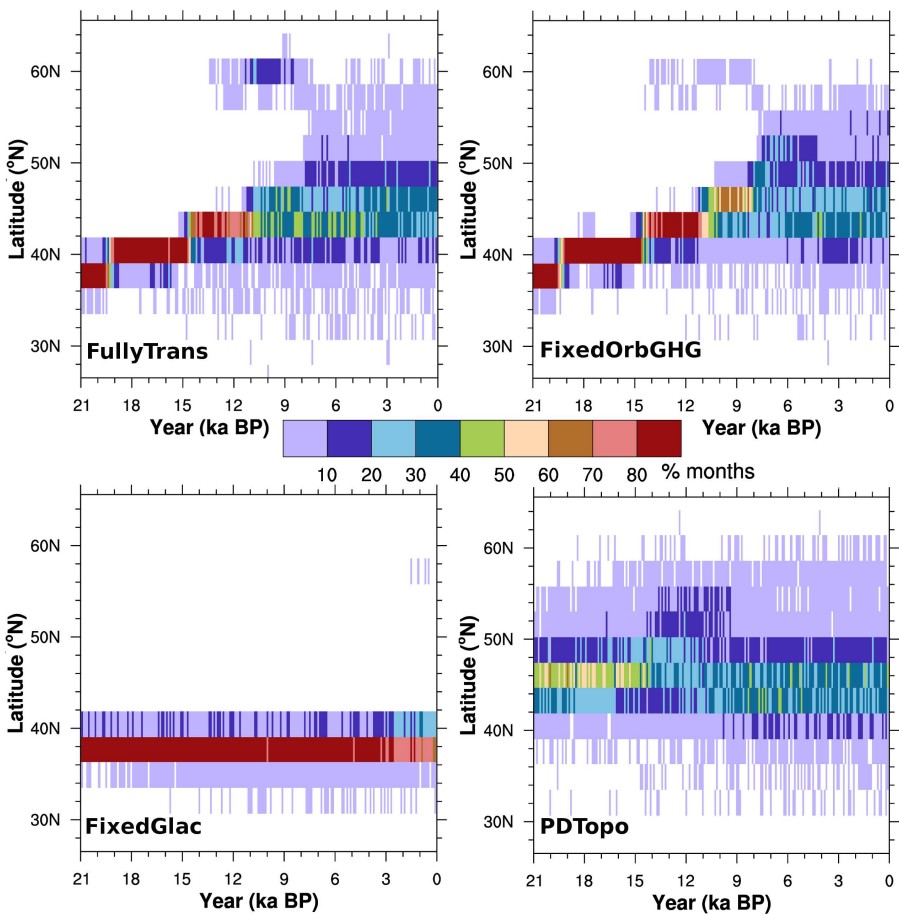

**Figure 10.** Frequency maps of NAtl, western jet latitudes based on winds over 700hPa to 925hPa and 270°E to 300°E in 10 successive winter seasons accumulated over all ensemble members of the FullyTrans, FixedOrbGHG, FixedGlac and PDTopo experiments. Colours indicate the percentage of months with peak zonal winds within each latitude bin of width 2.8° at T42.

The TraCE-21ka experiment exhibits both latitudinal jet shifts and a shift away from a single, most commonly-occupied jet latitude, although the timings of these jet changes are not the same as in the PlaSim simulations. At LGM, the jet is highly focussed with a narrow range of values, and its primary latitude shifts northward two times over the deglaciation (Figure 12). Unlike the PlaSim simulations, the first northward jet shift occurs within one decade of simulation at 13.87 ka BP, which coincides with a change in the ice sheet boundary conditions and freshwater forcing provided to the model (He, 2011). As in the PlaSim simulations, this change is driven by a shift on the western side of the jet, plotted in the bottom-left panel of Figure 12. The second shift is much more gradual and occurs from 13 to 12.5ka BP. This second transition appears to be an artifact of more gradual and asynchronous changes in the western and eastern sides of the jet.

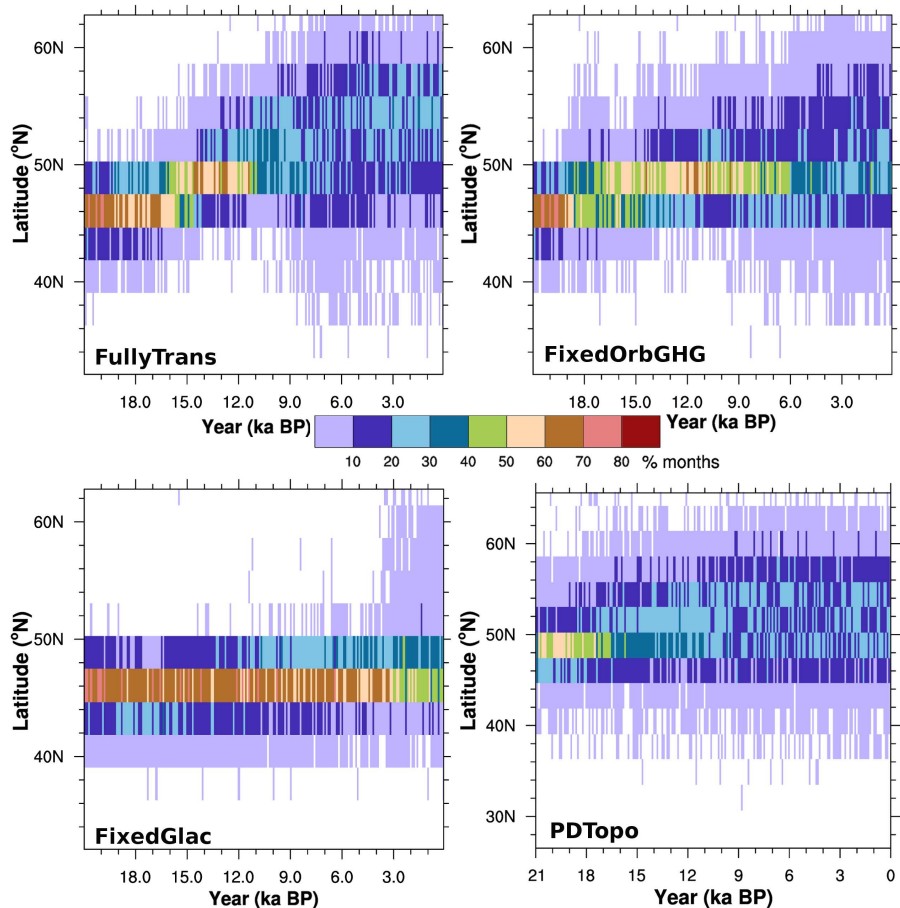

**Figure 11.** Frequency maps of ensemble-average, eastern NAtl jet latitudes based on winds over 700hPa to 925hPa and 330°E to 360°E in 10 successive winter seasons for FullyTrans, FixedOrbGHG, FixedGlac, and PDTopo experiments. Colours indicate the percentage of months with the difference in jet latitudes between 330°E to 360°E and 270°E to 300°E within each bin of width 2.8°.

The transition away from the NAtl jet occupying a single latitude more than 50% of the time in TraCE-21ka occurs more gradually and begins a little earlier than in the PlaSim simulations. This transition is complete by approximately 11ka BP and the range of jet values continues to increase until 6ka BP.

The NAtl jet tilt in Figure 12 calculated from the TraCE-21ka experiment exhibits no change over the deglaciation. This result stands in contrast to Lofverstrom and Lora (2017), who diagnosed a rapid increase in jet tilt at 13.89ka BP. The source of this discrepancy is discussed further in Section 4.

Thus, the NAtl jet exhibits changes in position and distribution that occur independently of each other in both PlaSim and TraCE-21ka simulations. Some of these changes coincide with historical climate changes. Since these jet characteristics and the timings of their changes differ on the eastern and western sides of the NAtl jet, we attribute them separately in the next section.

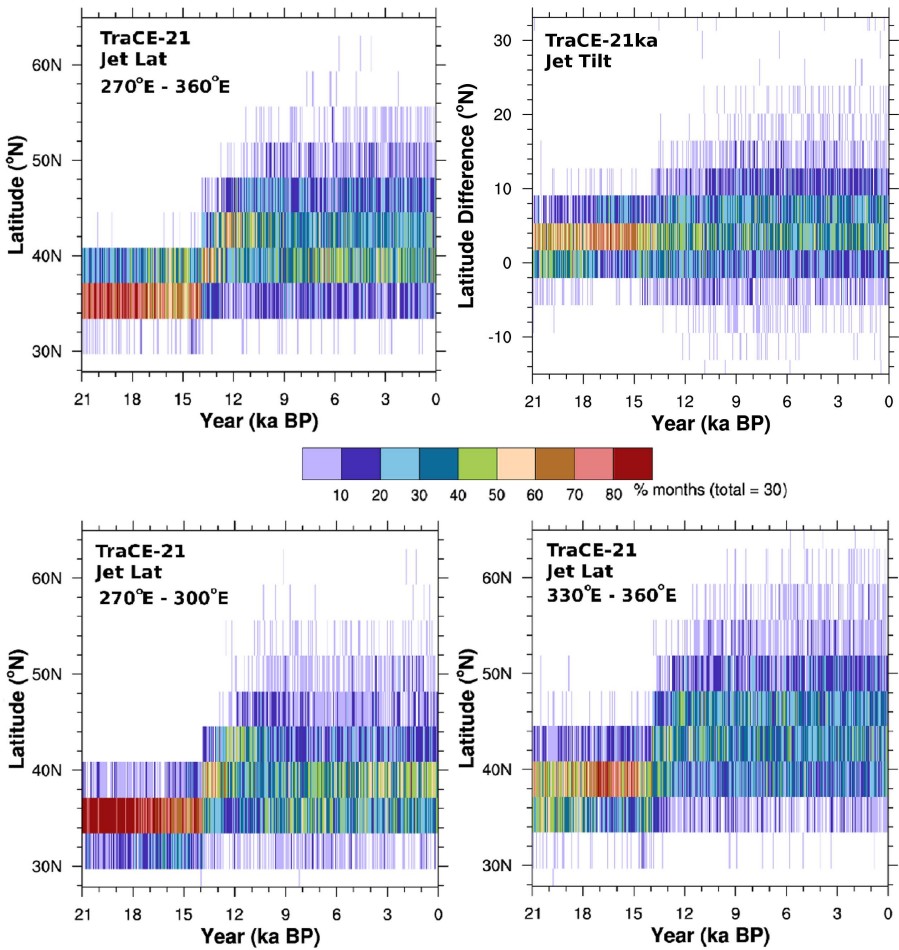

**Figure 12.** Frequency maps for the NAtl jet derived from the TraCE-21ka deglacial experiment over 700hPa to 925hPa for jet latitudes evaluated over longitudes 270°E to 360°, as well as the upstream (270°E-300°E) and downstream (330°E-360°E) regions and their difference (i.e. jet tilt).

## 3.3 Attributing causes of simulated jet transitions

In order to attribute the deglacial jet changes to boundary condition changes, we turn to the sensitivity experiments, plotted in Figures 10 and 11 for the western and eastern regions, respectively. Sensitivity results for the mean jet position and tilt are presented in Supplemental Figures S6 and S7.

Ice sheets provide the primary controls over the deglacial jet changes described in the previous section. Simulations with fixed LGM ice sheets (FixedGlac in Figures 10 and 11) reproduce neither the deglacial changes to the most commonly-occupied jet latitude nor the bulk of changes to its variability on both sides of the jet. This effect is most prominent for the western side of the jet, which shows almost no change over the deglaciation when the ice sheets are fixed to their LGM state. In the east, the

most common position of the NAtl jet does not change under fixed LGM ice sheets, but the frequency of time the jet spends at this latitude decreases, and its range of variability increases. Only orbital and greenhouse gas forcings are changing at this time, so this may indicate a sensitivity to those forcings (perhaps mediated by the changing sea ice extent and sea surface 5 temperatures).

Other sensitivity experiments can help decompose which attributes of the ice sheets are enacting this control on the NAtl jet. The PDTopo experiment isolates the thermal forcing associated with ice sheets' relatively high albedo from the orographic forcing due to their elevation by imposing present-day land topography in combination with time-varying glacial surface albedo. Thus, at LGM, ice sheets are infinitessimally thin but extensive. In neither the east nor the west is the NAtl jet as 10 focussed, or as equatorward-shifted in PDTopo (Figures 10 and 11) as it is at LGM in the FullyTrans runs or throughout the FixedGlac runs. Consequently, we conclude that the ice sheet albedo alone is not the primary controlling factor on the NAtl jets, and that the elevation of ice sheets is important. However, it is not clear from the present experiments whether orographic changes to the ice sheets alone are sufficient to explain the jet changes, or whether the ice sheets need to be reflective. Since the ice sheets became quickly covered with highly-reflective snow in the DarkGlac simulations, that experiment did not resolve 15 this question.

Additional experiments are required to unequivocally isolate the regions of the ice sheets whose orography is affecting the NAtl jet. Yet, a relationship can be identified between the latitude of the western side of the jet and the minimum latitude of the south-eastern margin of the NAIS in scatter plots based on the FullyTrans and TraCE-21ka experiments (Figure 13).

The western side of the jet either reaches or lies south of the south-eastern NAIS margin at all times in the FullyTrans 20 simulations and in the TraCE-21ka experiment. The only exception to this is when the wind branch along the north-eastern slope of the NAIS is identified by the algorithm as having the fastest winds (points located within the blue boxes in Figure 13).

There is no such clear relationship with the latitude of the eastern side of the jet (not shown), nor with the latitude of the western side of the jet when the ice sheet is infinitessimally thin in PDTopo (right side of Figure 13). Since the albedo of the south-eastern margin of the NAIS is insufficient on its own to restrict the latitude of the western side of the jet, it appears that the 25 margin must be elevated to restrict the jet. Whether this barrier effect is enacted via dynamical processes alone or is amplified by the thermal gradient associated with the ice sheet's albedo and lapse rate is not distinguishable from the experiments here.

Thus, we find that the change in variability of the NAtl eddy-driven jet during the deglaciation from a latitudinally-constrained to a latitudinally-variable regime are best explained by the effect of the ice sheet margin on the winds in the western region of the jet.

30 – The western side of the jet is more focussed to a single latitude than the eastern side, particularly when the NAIS extends well into the midlatitudes. This is because the ice sheet margin is constraining the winds in the western region of the jet, but not in the eastern region.

– The most commonly-occupied latitude of the western side of the jet shifts northward in a step-wise fashion. The ice sheet mask and topography are specified on the model grid, so the jet is able to move northward into a grid cell as soon as the ice sheet retreats from it. Without the ice sheet barrier effect, the most common tilt in the PDTopo experiment is

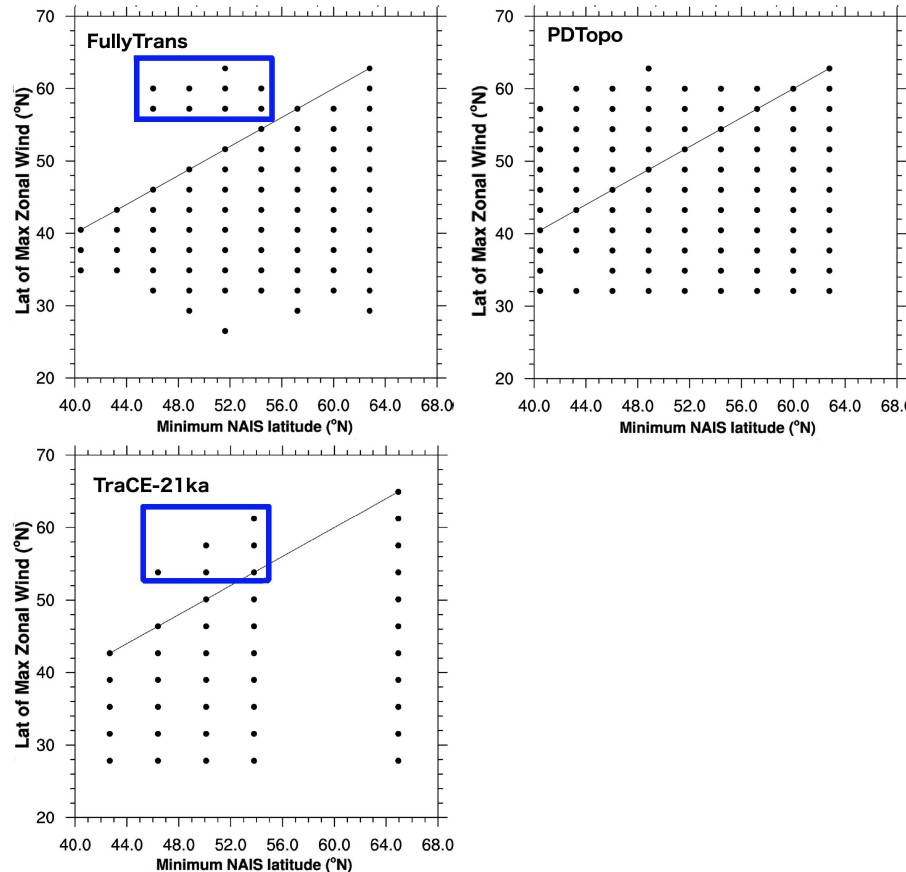

**Figure 13.** Scatter plots for monthly NAtl jet latitudes from 700hPa to 925hPa on the western side of the Atlantic (270°E - 300°E) versus the minimum latitude of the NAIS ice mask east of 270°E for all FullyTrans and PDTopo ensemble members and TraCE-21ka. Note the PlaSim latitudes are calculated on the T42 grid, and the TraCE-21ka latitudes are calculated on the T31 grid. Blue boxes identify points associated with the northern jet latitude branch.

much smaller than in FullyTrans and does not change over the deglaciation (Supplemental Figure S7), as the western and eastern sides of the jet appear to shift together.

– The distribution of the western side of the jet during most of the deglaciation is strongly skewed with the jet spending the bulk of its time at the northern boundary of its range. In contrast, the jet distribution in the eastern region is more symmetrical. The ice sheet only acts as a barrier to the western region of the jet, which creates a strong wind shear along the northern edge of the jet there. This shear causes eddies to break along this boundary and accelerate the flow locally, which keeps the jet preferentially in its northernmost position. Since there is no such constraint on the jet in the eastern 10 region, it is free to vary equally in both directions around its most common position.

Although ice sheet orography appears most important to NAtl jet changes, it does not provide the only controls on the jet. Fixing the radiative forcings to their LGM values (FixedOrbGHG in Figures 10 and 11) delays the transition away from a strongly preferred jet latitude on both sides of the NAtl eddy-driven jet, but particularly in the east. The timing of the first two western jet latitude shifts are unaffected by changes to the radiative forcings since these are primarily driven by ice sheet changes. However, an additional shift occurs after 11ka BP in the western region of the NAtl jet when the radiative forcings are fixed to their LGM values, and the jet shifts away from a single, most commonly-occupied jet location later than before at 8ka BP. In the eastern region, the shift in latitudes occurs earlier than in FullyTrans, but the jet never reaches as northern a position as it does in FullyTrans. Thus, orbital configuration and greenhouse gases appear to be key for the transition of the jet away from a narrow, peaked latitudinal distribution to a broader, less peaked distribution, particularly in the eastern region of the NAtl jet.

Overall, the results presented here indicate that changes to the western side of the jet appear to be predominantly controlled by changes in the position of the south-eastern NAIS margin, while changes to the position of the eastern side of the jet are sensitive to the thermal background conditions of the climate (which the ice sheets also affect).

## 4  Discussion and Conclusions

This paper explores the question of whether winter eddy-driven jet changes over the North Atlantic could have contributed to the abrupt climate changes detected in Greenland ice cores over the last deglaciation. Lower-tropospheric winds are most likely to contribute to abrupt, large-scale climate changes either by directly triggering changes in surface ocean circulation patterns and/or sea ice extent (and thereby deepwater formation rates), or by altering the state of these components in a way that makes them unstable to perturbations from another source. Thus, we have assessed the timing and abruptness of changes to the latitudinal position and variability of the North Atlantic (NAtl) eddy-driven jet in a small ensemble of transient deglacial simulations using the PlaSim model and the TraCE-21ka simulation. Since none of the NAtl jet changes in the PlaSim simulations lead to large-scale, abrupt climate changes in those simulations, our results provide no support for a direct, causal role for jet changes in triggering historical abrupt climate changes. It is unclear the extent to which model limitations (such as purely thermodynamic sea ice) and run acceleration are contributing to this negative result. Regardless, since the jet undergoes abrupt changes over the deglaciation, we can not discount the possibility that the jet set the stage for such events to occur.

Assessing the abruptness of NAtl jet changes is problematic, because the abruptness of a phenomenon depends on its context. During the deglaciation, changes are generally considered abrupt if they occur within a couple of decades and are of sufficient amplitude to appear unusual compared to background climate variability. In assessing the abruptness of jet latitudinal changes, we must also address the fact that the wind data is gridded; a latitudinal change in the position of the NAtl jet will always involve a discrete step from one latitude band to another. However, as discussed in Section 3.2, we can distinguish between gradual latitudinal changes where the jet evenly splits its time between two adjacent latitude bands while it transitions from one to the other and more abrupt changes where there is no such mixed state. Thus, we consider a jet change to be abrupt (given

the decadal resolution of our jet diagnostics) when the most commonly-occupied jet position shifts from one grid cell to the next without an intermediate mixed state.

All of the fully-transient deglacial simulations presented here show that the NAtl eddy-driven jet shifted northward from the Last Glacial Maximum to the preindustrial period, and its latitudinal variability increased. These characteristics match those derived from other studies (Li and Battisti, 2008; Lofverstrom et al., 2014; Merz et al., 2015). However, unlike those studies, neither the PlaSim simulations nor TraCE-21 show much change in jet tilt between these two periods. This contrasts with previous work on TraCE-21ka that identifies an abrupt increase in tilt in this dataset that occurred at 13.89ka BP (Lofverstrom and Lora, 2017). These two TraCE-21ka results were obtained from different wind data extracted from the same dataset: lower-tropospheric winds are analysed in this study, while Lofverstrom and Lora (2017) examine upper tropospheric winds at 250hPa. There are additional differences in the range of longitudes used to specify the western and eastern regions of the NAtl jet: 280 to 290°E and 340 to 350°E, respectively in Lofverstrom and Lora (2017) versus 270 to 300°E and 330 to 360°E in this study. We are able to reproduce the results obtained by Lofverstrom and Lora (2017) (except for the timing, which we date to 13.87ka BP) when we calculate the jet tilt in the same manner as they did (figures are presented in Supplemental Figure S5, alongside corresponding figures using the methodology employed in this study). Since Li and Battisti (2008) and Lofverstrom et al. (2014) also derive their tilt results from upper-tropospheric winds, it is not clear how much their results are in conflict with those presented here (Merz et al. (2015) analyse lower-tropospheric winds, but do not discuss tilt changes).

The novelty of this present study compared to those previously mentioned is that it analyses transient changes in the jet over the entire deglaciation in multiple experiments with a small ensemble of model runs. Additionally, we do not assume a Gaussian distribution of jet characteristics which is the implicit assumption of studies that solely characterize the jet via its mean and standard deviation. Instead, through frequency plots, we present changes in time of the jet distribution itself by showing the percentage of winter months that the jet spends at each latitude for every decade of simulation.

The deglacial NAtl jet changes arise in a set of PlaSim simulations where boundary conditions vary smoothly in time. Nevertheless, the jet changes via three well-separated northward shifts in latitudinal position (at 19ka BP, 14ka BP and 11.0ka BP) followed by a gradual transition away from a latitudinally-constrained regime of variability that is complete by 8ka BP in all simulations. The TraCE-21ka experiment exhibits two shifts in the most common position of the jet (at 13.87ka BP and 13ka BP) contemporaneous with the start of a gradual increase in latitudinal variability.

Interpreting the detected jet changes is complicated by the fact that the NAtl jet calculated over longitudes 270°E to 360°E mixes characteristics of the upstream and downstream regions. The evolution of the jet in these two regions differs in timing, abruptness and dependence on background model conditions. The first two of the mean jet shifts detected in the PlaSim simulations are driven by changes in the position of peak winds on the upwind region of the eddy-driven jet over eastern North America. These shifts are each accomplished within a decade of simulation (century of forcing), and their timings are consistent between the accelerated simulations, an unaccelerated run, and an accelerated transient simulation starting from a warmer initial state. Only in the TraCE-21ka experiment are the timings of these shifts different. In contrast, only a single shift in peak winds between 16 and 14ka BP is evident in the downstream region of the eddy-driven jet over the eastern side of the

NAtl. The shift on the eastern side of the jet occurs over 200 simulation years (two millennia of forcing), and its timing differs between runs with and without acceleration, in the warmer transient simulation, and in the TraCE-21ka experiment.

The different characteristics of changes in the upstream and downstream regions of the North Atlantic jet correspond to different mechanisms of change. While previous work shows that the western side of the jet is more strongly affected by ice sheet orography and the eastern side by stationary and/or transient eddies (Kageyama and Valdes, 2000; Lofvestrom et al.,

2016), we identify a more specific relationship between the North American ice sheet complex and the western side of the jet. In both the PlaSim and TraCE-21ka experiments, low-level winds over eastern North America are highly focussed to the latitude of the ice sheet margin and vary only south of this margin, even though the ice sheet surface lies below the top of our low-level wind range. When this margin retreats, the winds tend to shift northward with it, but not before. It is the orography of the south-eastern margin of the North American Ice Sheet complex that appears to play the most important role in this effect,

as simulations with spatially extensive but infinitessimally thin ice sheets do not reproduce this effect. However, whether the thermal gradients associated with the elevated ice sheet surface contribute to this effect in a subsidiary way is not clear from this study.

From these results, we hypothesize that different ice sheet reconstructions represented at different resolutions will lead to different timings and numbers of simulated jet shifts over eastern North America. This was indeed the case in this study, as the

timing and number of northward jet shifts in the PlaSim simulations (T42 resolution with GLAC1-D updated every simulation year) and TraCE-21ka (T31 with ICE-5G updated approximately every 500-1000 years) differed. If this hypothesis is correct and this mechanism is as important in reality as it is in these two models, then our knowledge of the southernmost position of regional ice provides a northern bound on the historical position of the jet.

In contrast, the downstream side of the jet over the eastern North Atlantic does not show the same sensitivity to the marginal

position of the North American ice complex, but it is still affected by the presence of elevated ice sheets. Whether this control is exerted via changes to the stationary waves (e.g. Kageyama and Valdes (2000); Lofverstrom et al. (2014)), transient eddies (e.g. Merz et al. (2015)), surface thermal gradients from sea ice and sea surface temperatures (e.g. Li and Battisti (2008)), some combination of these or some other mechanism altogether is not fully discernable from this study. There is some evidence that sea ice and sea surface temperatures may play a role, as the range of jet latitudes increases after abrupt sea ice retreats and sea surface temperature warmings in the FixedGlac experiment, and the eastern jet distribution is centred around different latitudes at the end of the deglaciation in FullyTrans and PDTopo even though their forcings are the same at this time.

Orbital and greenhouse gas forcings contribute to the spreading of the distribution of jet latitudes away from its peak value in both regions of the jet. However, this effect is most prominent in the eastern North Atlantic, where these forcings also affect

5  the jet's latitudinal position after the deglaciation. In general, it appears that the characteristics of the jet on the eastern side of the North Atlantic are sensitive to changes in the background climate state, likely through changes to the positions of the polar front (and the growth of associated eddies) and sea ice margin.

As such, it would be difficult to estimate historical changes to the jet in this region from model simulations, since the pattern of thermal responses to changes in boundary conditions is sensitive to model parametrizations for processes like cloud physics,

and feedbacks between the atmosphere, ocean and sea ice. Thus, the effect of changing boundary conditions likely varies between models and even between simulations using the same model but different initial boundary condition states.

One advantage of the design of the PlaSim simulations compared to TraCE-21ka for studies such as this one is that ice sheet characteristics were updated every simulation year in the PlaSim runs. Enacting these boundary condition changes gradually provided better estimates of the timing of jet changes and made attributing these changes easier. The TraCE-21ka experiment updated ice sheet and meltwater prescriptions every few hundred to a few thousand simulation years and held them fixed in between (He, 2011). This likely contributed to the lack of change in the jet over North America in TraCE-21ka until 13.87 ka BP, when its ice sheet area and topography were adjusted from 15ka BP specifiations to 14ka BP values. At the same time, Northern Hemisphere freshwater routing was shifted from the Mackenzie River to the Nordic Seas and St. Lawrence River, and Weddell Sea hosing was turned off. Lofverstrom and Lora (2017) attributed the increase in jet tilt at this time to the opening of the Cordilleran and Laurentide ice sheets, but there were many other factors that may have contributed to this change as well.

Note that the ice sheet elevation in some grid cells exceeds the bottom pressure level of the analysis range, so we interpolated the winds to these levels to not bias the resulting average. The land surface does not pass above the 700hPa pressure level in any of the grid cells included in our analysis, so we are not introducing winds to a region where there was none. The impact of this choice was tested on the 850hPa level and found to not change the jet results (see Supplemental Figure S9) except when the jet is calculated over the entire longitudinal range. In this case, excluding grid cells from the analysis on the western side of the region effectively weights the mean jet toward characteristics of the eastern region. This issue illustrates the sensitivity of this diagnostic to the longitudinal range that is employed. Due to this sensitivity and the important differences in jet characteristics in the two regions, we argue that analyses over the western and eastern jet regions are more instructive than the more commonly-used jet latitude and tilt metrics, and would encourage other authors to present these metrics as well.

Finally, given the constraint provided by the ice sheet on the western side of the jet, we can imagine scenarios whereby abrupt jet changes could have occurred in this region in the past.

1. the ice sheet margin exhibited an abrupt retreat or abrupt thinning, allowing the jet over the western NAtl to abruptly shift northward, or

2. other processes, the origin of which are not presently clear, constrained the regional jet position for an interval of time, after which the jet rapidly adjusted to the ice sheet margin.

It is unlikely that relevant changes in the ice sheet margin occur on timescales of decades, so it appears unlikely that changes to the upstream end of the North Atlantic jet directly caused past abrupt climate changes. We have less insight into the conditions that may lead to abrupt jet transitions in the eastern region of the jet, so it is difficult to assess their plausibility. However, even gradual changes to the jet in either region could have enabled past abrupt changes by altering the background climate conditions in such a way that another triggering process could occur. Alternatively, relatively gradual jet changes may be able to trigger abrupt climate changes, depending on the non-linearity of the climate system in the NAtl region (e.g. feedbacks associated with the location of the sea ice edge). Better understanding of possible climate system sensitivities to NAtl jet changes will require additional sensitivity experiments that include dynamic sea ice.

## 5 Acknowledgements

*Code availability.* TEXT

*Data availability.* TEXT

*Code and data availability.* TEXT

*Sample availability.* TEXT

## Appendix A: Testing Experimental Design

There are two aspects to the design of this study which could possibly affect the results presented here. These include the model's high-latitude cold bias with respect to CMIP5 LGM simulations (described in Section 2.2.5), and the acceleration of
the forcings by a factor of 10. The effects of each of these aspects have been assessed using additional simulations and are discussed below.

### A1  Cold Arctic at LGM

As described in Section 2.2.5, the PlaSim simulations exhibit colder conditions in the polar regions during winter compared to the CMIP5 multi-model ensemble, particularly in the Northern Hemisphere. The strength of these temperature differences
decreases over the deglaciation until by the last millennium the 2m temperatures generated by PlaSim lie mostly within the range of CMIP5 experiments. The location of the largest temperature differences over snow-covered regions suggests that the treatment of snow albedo may at least partly explain them.

Most high-latitude land is covered with a perennial snow cover in the LGM simulations that is metres thick. The albedo of snow in PlaSim is treated as a linear function of surface temperature with cutoff minimum and maximum values of 0.4
and 0.8, respectively. Most problematically, PlaSim's default albedo scheme allows the snow albedo to return to fresh snow values after periods of snow melt, without the requirement of additional snowfall. This unphysical albedo scheme was tested in Koltzow (2007) using the HIRHAM model, and it was found to underestimate satellite-derived clear-sky snow albedo at temperatures above -6°C and overestimate the albedo at temperatures below -8°C. These characteristics are consistent with the climate differences detected in the PlaSim simulations for LGM: they are strongest over snow-covered regions during cold
seasons.

To test the influence of this bias, we modified the snow albedo scheme in unforested regions in a manner analogous to Helsen et al. (2017). After a snowfall of at least 20 cm water equivalent, the snow albedo is reset to a maximum value of 0.85. With less snowfall than that, the increase in albedo is scaled linearly between its maximum value and its previous value according to the relative amount of snowfall. When there has been no snowfall, the snow begins to age via an exponential decay via a temperature-dependent time constant toward a temperature-dependent minimum albedo value. This value ranges linearly between an upper limit for firn (albedo 0.75) at temperatures below -5°C and a lower limit for wet snow (albedo 0.55) at temperatures above the melt temperature. The time constant for this decay is defined in Equation A1, where tau is the time constant in days and T is the surface temperature in Kelvin.

$$\tau = 30 - 29.875 * MIN[1, MAX[0, ((T - 268.16)/(273.15 - 268.16))^{0.25}]] \tag{A1}$$

This parametrization of the time constant gives values of 3 hours for temperatures at the melting point, 4 days for -2°C and 30 days for -5°C.

This updated snow albedo parametrization was applied in PlaSim to snow cover over land and sea ice, and a new LGM spin-up and accelerated transient experiment were performed. Sea ice is much less extensive at LGM in the new transient simulation as seen in Figure A1 and temperature biases with respect to the CMIP5 multi-model ensemble are reduced. However, Arctic winter temperatures are still outside the range of the CMIP5 models at LGM. The remaining model differences suggest the need for additional comparisons in the tuning procedure that constrain the seasonal cycle, which would better constrain the changes in the climate under different forcing conditions.

The jet latitudes in the transient run using the new albedo parametrization are very similar to those in the original accelerated transient simulations in the upstream region of the jet. However, in the jet as a whole and in the downstream regions plotted in Figure A2, the transition to a more distributed jet without a single, most commonly-occupied latitude occurs much earlier over the eastern NAtl, by 15ka BP instead 11ka BP. These changes are consistent with the conclusions presented here that the constraints on the upstream end of the jet are primarily caused by the position of the south-eastern margin of the NAIS, while the controls on the east are based on the thermal state of the atmosphere and sea surface temperatures. A warmer initial state with less extensive sea ice achieves a similar jet state over the eastern NAtl earlier.

**A2 Simulation Acceleration**

Another run was generated identical to FullyTrans1 except without acceleration to identify what influence the ten times acceleration may have had on the results presented here and to better assess the abruptness of the transitions. Frequency plots for the jet latitudes in all regions are shown to year 6.3ka BP in Figure A3. Jet latitude shifts in the western region of the jet occur at the same time as in the accelerated runs, and the most abrupt transition between 15ka BP and 14ka BP occurs within 80 years. Over the eastern region, the jet changes occur earlier than without acceleration. Thus, allowing the ocean to adjust to the changing climate conditions changes the timing of the jet shifts over the eastern NAtl and the timing of the transition away from a single, most commonly-occupied jet latitude. Notably, there are quasi-periodic, short-lived episodes throughout

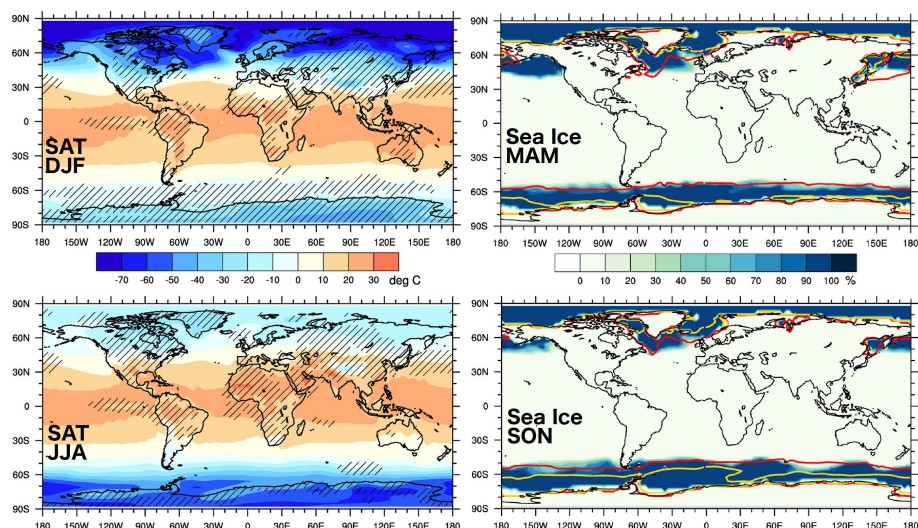

**Figure A1.** 2-metre temperatures and sea ice concentration at LGM in the first century of a single transient simulation employing the new snow albedo parametrization compared to CMIP5 multi-model minimum and maximum values. Hatching represents regions where climatological values lie within the range of CMIP5 models, while red lines indicate CMIP5 maximum sea ice extent and yellow lines indicate CMIP5 minimum extent.

the deglaciation in this run when the jet position in the east shifts northward, and its range extends much further north. These periods are coincident with episodes of abrupt warming in the run.

*Author contributions.* HJA and LT designed the experiments. HJA modified the PlaSim model code, performed the simulations and analysed the output. HJA prepared the manuscript with contributions from LT.

*Competing interests.* The authors declare that they have no conflict of interest.

*Disclaimer.* TEXT

*Acknowledgements.* The authors thank Camille Li for her insightful comments and Taimaz Bahadory for his editing. Funding for HJA was provided by ArcTrain Canada and the German Federal Ministry of Education and Research (BMBF) in its Research for Sustainability initiative (FONA) through the PalMod project. All calculations were performed using HPC services from the Centre for Health Informatics

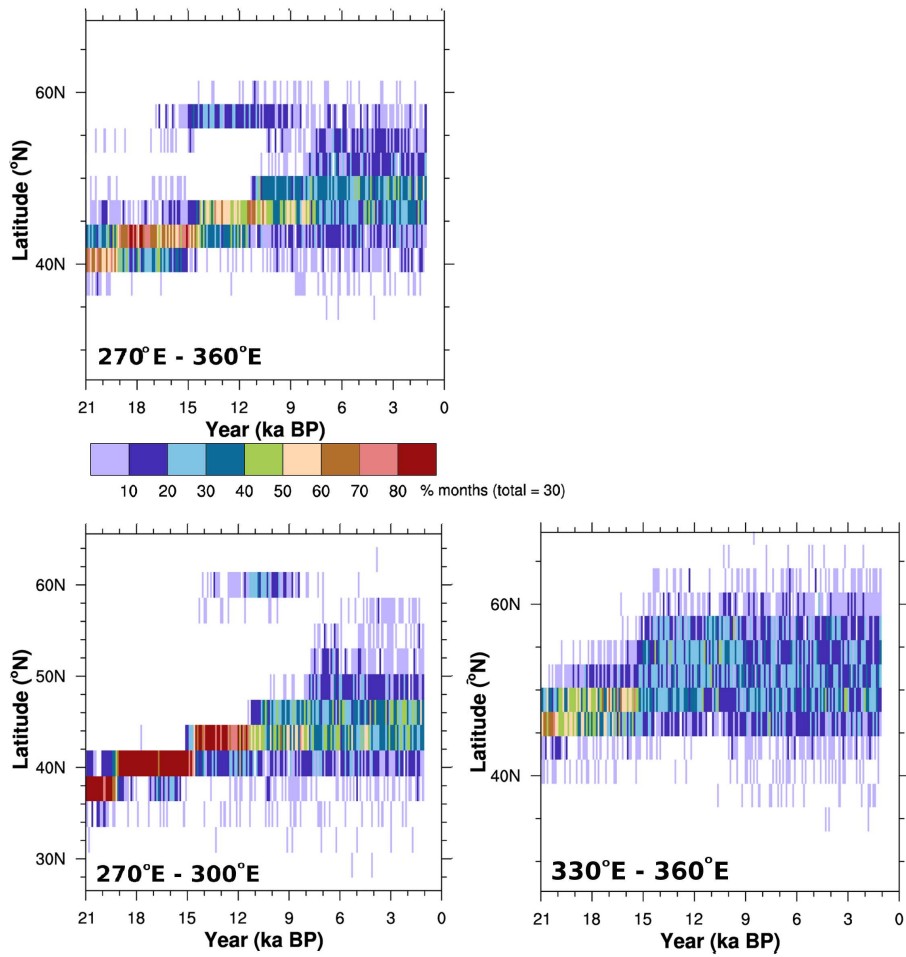

**Figure A2.** Frequency plots for North Atlantic eddy-driven jet latitudes as a function of time for the transient deglacial simulation using a revised snow albedo parametrization.

and Analytics at the Memorial University of Newfoundland. TraCE-21ka was made possible by the DOE INCITE computing program, and supported by NCAR, the NSF P2C2 program, and the DOE Abrupt Change and EaSM programs.

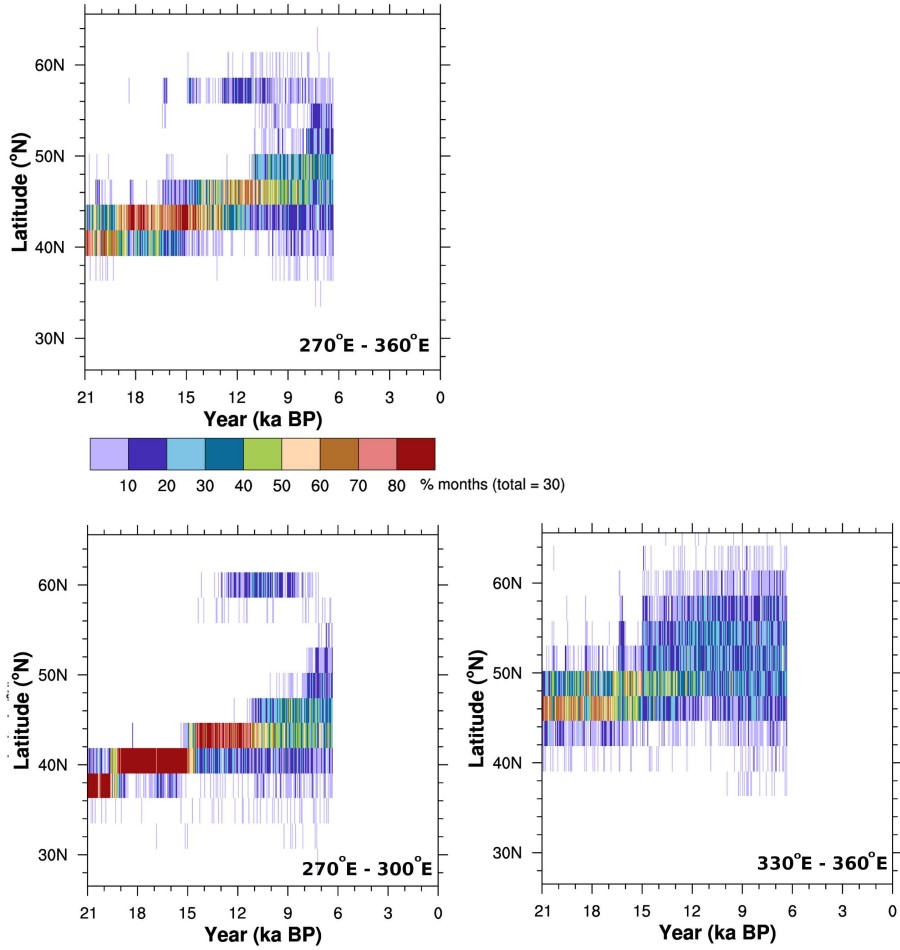

**Figure A3.** Frequency plots for North Atlantic eddy-driven jet latitudes as a function of time for the unaccelerated transient deglacial simulation.

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
