# Peer review of "Towards understanding potential atmospheric contributions to abrupt climate changes: Characterizing changes to the North Atlantic eddy-driven jet over the last deglaciation"

_Climate of the Past, 2018_

## Referee Comment (RC1) · Anonymous Referee #1 · 28 Dec 2018

**Review of:**
Andres and Tarasov, Towards understanding potential atmospheric contributions to abrupt climate change: Characterizing changes to the North Atlantic eddy-driven jet over the last deglaciation, Climate of the Past Discussion

This work is potentially of interest to the geosciences community.

**Summary:**
This manuscript describes a set of transient simulations of the last deglaciation (from LGM to present), investigating the evolution of the North Atlantic jet stream characteristics, such as axial tilt and meridional variability. Data from two different circulation models are analyzed: the Planet Simulator (an intermediate complexity model), and the TraCE-21ka simulation that was run with NCAR CCSM3 (a coupled atmosphere-ocean general circulation model).

In agreement with pervious studies, the authors find that the jet was narrow, fairly zonally oriented, and had a low meridional variability at the LGM and in the first part of the deglaciation, followed by an abrupt change in the jet characteristics (widening, increased axial tilt and latitudinal variability) around 14 ka (14,000 years ago). Altered jet characteristics are also identified at around 19 ka and 11 ka; to my knowledge, neither of the latter two have previously been discussed in the literature.

A number of sensitivity experiments are also carried out where the impact of different forcing agents is examined in isolation, e.g. ice sheets (and topography in isolation), greenhouse gas concentrations, and orbital parameters. The main conclusion is that the ice sheet topography has the strongest influence on the jet characteristics, especially over the western ocean basin where it is very latitudinally stable when the ice sheet is present. The eastern side of the jet is proposed to be more influenced by the thermal state of the background climate, which, in turn, is shaped by the ice sheet topography, greenhouse gas concentrations, and orbital parameters.

The paper is well written and presents a coherent and succinct story. I recommend publication, subject to some intermediate revisions. My main concerns are outlined below, followed by more specific line-by-line comments.

**Major concerns:**
**Lack of explanations**:
The authors provide an explanation for the suppressed variability of the western side of the jet stream (though I personally think that one key aspect is missing – see comment below). However, the more complicated part of the story, explaining the jet characteristics over the eastern ocean basin, is not discussed at the same level of detail. I think this is a shame and I encourage the authors to do more analysis to improve that part of the story.

**Lack of dedicated discussion section:**
The authors have decided to jump straight from the results section to the conclusions without having a proper discussion section where your findings are put in perspective with the existing literature. This makes it hard to get a sense for how your results differ from earlier studies and how they contribute to our understanding of the atmospheric circulation during the deglaciation. The discussion section is in my mind the most important part of a paper, so its omission feels instinctively wrong and may give this paper less traction than it deserves.

**Line-by-line comments:**
**Page 1, line 5:**
Remove "we performed" and specify that PlaSim is an intermediate complexity model.

**Page 2, line 17:**
Missing space between sentences

**Page 3, line 12:**
"Low levels of the atmosphere" is ambiguous. Perhaps better to say "lower troposphere".

**Page 3, line 13:**
Missing word. The extratropical jet stream is due to "momentum-flux convergence" by synoptic eddies.

**Page 4, line 11:**
Lofverstrom et al. (2016) showed that stationary waves can influence the jet characteristics in the eastern North Atlantic at the LGM as well.

**Page 4, line 23:**
Sentence starting with "These changes.." is incorrect. To the best of my knowledge, none of the papers cited here suggest that the subtropical and midlatitude jets entered a merged state over the N Atlantic at the LGM; see, e.g., Fig. 1 in Li and Battisti (2008) where there is a clear separation between the subtropical and eddy driven jets.

**Page 4, line 26:**
Missing space between sentences.

**Page 4, lines 26-28:**
None of the papers cited here investigated that explicitly.

**Page 4, line 32:**
"The timing.." meaning here is not clear.

**Page 5, line 5 and section 2.1:**
Perhaps nit-pick but this not technically correct. PUMA (the dynamical core of PlaSim) is indeed a dry primitive equation model. However, the extra layer of physics on top of the dynamical core makes PlaSim more than a primitive equation model. More correct to say that it is a simplified general circulation model or, better yet, an Earth-system model of intermediate complexity (EMIC).

**Page 5, line 28:**
It was recently shown by Lofverstrom and Liakka (2018) that T42 resolution is sufficient to reasonably capture planetary waves in simulations of the LGM climate.

**Page 5, line 29:**
The description here is not correct. The Gaussian grid is the 128 x 64 cell grid in real space that the data is outputted on ("Gaussian" refers to how the grid is generated). The primitive equation are partially solved in spectral space (wave space) and are thus transformed between grid space (on the Gaussian grid, in this case 128 x 64 grid points in lon x lat) and the spectral representation in wave space, which supports at most 42 harmonics in the zonal and meridional direction, respectively. (Hence the name T42, where the T is short for

"truncation" or more specifically "triangular truncation").

**Page 6, line 1:**
Not sure if I understand how the LSG models works. Is it a dynamic model that only runs in the mixed layer? If yes, how can a realistic ocean circulation be established if there is no deep ocean? Do you parameterize fluxes between the deep ocean and mixed layer? If yes, how are these fluxes calculated? What is the depth of the mixed layer? Prescribed or dynamic? Studies have shown that the mixed layer depth was substantially greater at the LGM (e.g. Sherriff-Tadano et al., 2018), which can have profound implications for the ocean heat contents and energy exchange between the ocean and atmosphere.

**Page 6, line 17:**
Please clarify, you update the boundary conditions every simulation year, but with 10x acceleration, meaning that you effectively only run every 10 years from LGM to PI. Is that correct?

**Page 7, line 12:**
"...temporal resolution of 100 years". This seems to conflict with the synchronous update of boundary conditions described above.

**Page 7, line 5:**
Please clarify how this process works. You can't fit an even number of 0.5° grid cells in a T42 cell (which is around 2.8°), so there must be some partial overlapping cells. Also, what does "effective higher-resolution grid cell length" mean?

**Page 8, line 18:**
Ivanovic et al. (2016) is double cited.

**Page 8, line 22:**
Century should probably be millennium here (you discuss 21 ka - 20 ka and 1 ka to 1950), right?

**Page 9, line 10:**
Sentence can be simplified; e.g.: "Also, the path of the NPac jet" -> "Also, the NPac jet"

**Page 9, line 14:**
I agree with this assessment and a similar conclusion was reached by Lofverstrom et al. (2016); see their discussion about sensitivity simulations with extensive sea ice in the eastern N Atlantic (their Fig. 6).

**Page 16, line 6:**
Write out explicitly that you are referring to Fig. 11 here.

**Page 16, line 5:**
Typo? ...range or its tilt -> ...range of its tilt (?)

**Page 16, line 15:**
What standard metrics?

**Page 16, line 17:**
Replace "instead" with "as well", and remove "For those interested".

**Page 17, line 5:**
Meaning here is not clear. Do you mean flat ice sheets (i.e., only accounting for the albedo effect)? Also, the ice sheet height is not the only thing influencing the circulation. As you say elsewhere, the spatial extent is also important.

**Page 18, line 1:**
How did you arrived at this specific number (725 m)?

**Page 18, line 9:**
Typo? "jet does not always move the the latitude of the jet."

**Page 19, lines 3-10:**
This explanation is a bit too simplistic. I agree that the presence of the ice sheet constrains the jet latitude in the west, presumable in part because of obstruction of the flow by the topography. However, the thermal gradient at the southern ice margin can influence the flow in a similar fashion (this is not mentioned here as far as I can see) – both the change in albedo at the ice sheet margin, and the adiabatic cooling of the flow by the implied elevation difference. The modern (PI) jet is also less variable in the western ocean basin because of the strong thermal gradient at the sea-ice edge. This is clearly a different mechanism than the presence of a big ice sheet, but the effect is similar.

**Page 21, line 25:**
Meaning here is not clear — this seems to be the definition of a shift in the jet latitude.

**Page 22, line 16:**
What figure is discussed here?

**Page 22, line 25:**
I would encourage you to think a little bit more about this and try to give a mechanistic explanation for this phenomena. Doesn't have to be a full explanation, but at least something that adds a little bit more to the story.

**Figure 1:**
What ice sheet remained in North America through the Holocene and is 1.5 km thick?

**Figure 4:**
Panels showing LGM and past1000 are mixed up (LGM is shown in middle panels).

**Figure 4:**
The top of the LGM ice sheets (and indeed some modern topography) is higher than the 700 hPa isobar. I would advice against extrapolating the wind field in these regions and instead treat it as missing data, as extrapolation can cause some weird effects when doing statistical analysis (e.g. when determining the latitude of the strongest winds in the western N Atlantic).

**Figure 6 – 8:**
Use the same range on the spines on the right hand side for easier comparison (e.g., in Fig. 6: 0–80 % in top panels and 0–35 % in lower panels).

**Figure 9:**
10 successive years is a bit ambiguous because it can be done in at least two different

ways: (1) a sliding mean where the input and output arrays have the same length; (2) form decadal averages where the input array is 10x longer than the output array. These methods will yield slightly different results. I doubt that the difference will be of sufficient magnitude to challenge your conclusions, but this type of information is important for reproducibility.

**Figure 9 – 11 and 13:**
Write out lat and lon bounds and pressure level(s) used in statistics.

**Figure 11:**
Caption appears to be wrong as you show latitude here, not difference in latitude across the N Atlantic.

**Figure 12:**
Use same latitude range on vertical axis for easier comparison.

**References**

Ivanovic, R., Gregoire, L., Kageyama, M., Roche, D., Valdes, P., Burke, A., Drummond, R., Peltier, W. R., and Tarasov, L.: Transient climate simulations of the deglaciation 21-9 thousand years before present (version 1)-PMIP4 Core experiment design and boundary conditions, Geoscientific Model Development, 9, 2563–2587, 2016.

Li, C. and Battisti, D.: Reduced Atlantic Storminess during Last Glacial Maximum: Evidence from a Coupled Climate Model, J. Climate, 21, 3561–3579, 2008.

Lofverstrom, M. and Liakka, J.: The influence of atmospheric model resolution in a climate model-forced ice sheet simulation, The Cryosphere, 12, 1499–1510, doi: https://doi.org/10.5194/tc-12-1499-2018, 2018.

Lofverstrom, M., Caballero, R., Nilsson, J., and Messori, G.: Stationary wave reflection as a mechanism for zonalising the Atlantic winter jet at the LGM, Journal of the Atmospheric Sciences, 73, 3329–3342, doi:10.1175/JAS-D-15-0295.1, 2016.

Sherriff-Tadano, S., Abe-Ouchi, A., Yoshimori, M., Oka, A., and Chan, W.-L.: Influence of glacial ice sheets on the Atlantic meridional overturning circulation through surface wind change, Climate Dynamics, 50, 2881–2903, 2018.

---

## Author Comment (AC1) · 19 Jan 2019

The authors thank Anonymous Referee #1 for his/her considered comments.

Changes to the text and detailed corrections regarding typos and such will be provided at the end of this round of review, except where they affect the meaning of the text. More substantive comments are addressed here. Note unless otherwise indicated, abbreviated versions of our responses below will be inserted into the revised text.

Reviewer's comments are provided in italics

*Lack of explanations : The authors provide an explanation for the suppressed variability of the western side of the jet stream (though I personally think that one key aspect is missing – see comment below). However, the more complicated part of the story, explaining the jet characteristics over the*

*eastern ocean basin, is not discussed at the same level of detail. I think this is a shame and I encourage the authors to do more analysis to improve that part of the story.*

We agree with this reviewer that it is important to disentangle the various possible contributors to deglacial jet changes on the eastern side of the North Atlantic (Natl). However, this can't be done to the same level as the western jet with the existing set of ensemble runs. The potential contributors that we have considered to the eastern jet changes include:

- Stationary and transient eddy responses to ice sheet topography changes,
- Thermal effects on latitudinal surface temperature gradients in the North Atlantic (and possibly sea ice) due to ice sheet area and height (over North America and Fennoscandia), greenhouse gas concentrations and orbital forcing,
- Indirect effects associated with the pinning of the western side of the jet to a particular latitude, and
- Indirect effects associated with ocean circulation responses to the boundary condition changes.

We will add to the revised text the following conclusions that we can draw based on our existing runs.

1. The presence of elevated and extensive ice sheets provides the dominant controls on the jet characteristics on both sides of the NAtl jet. When the ice sheets are fixed to their LGM configurations in FixedGlac, little change is seen in the position or distribution of the jet on both its western and eastern sides. One notable exception to this is after 4ka BP, when the eastern side of the jet shows a reduced frequency at its preferred latitude and an expansion of its range. This timing is coincident with abrupt warming events in the FixedGlac experiments that are accompanied with abrupt retreats in Northern Hemisphere sea ice extent.
2. The eastern side of the NAtl jet is more sensitive to the background climate state than the western side of the jet is. This is evident when examining the differences between the FullyTrans and FixedOrbGHG experiments. By present day, the western side of the jet is mostly unaffected by the orbital and GHG components being fixed to LGM values. In contrast, the eastern side of the jet is centred approximately 5° further south when orbital and greenhouse gas components are fixed to LGM.
3. The position of the western side of the jet has limited control on the position of the eastern side of the jet. This conclusion is arrived at by considering the difference between western and eastern jet response to the FixedOrbGHG experiment discussed above and by examining the PDTopo experiment,. For the latter, the position of the western side of the jet is approximately 8° further north than in the FullyTrans experiments before 20ka BP. In these same experiments

at that time, the preferred eastern position of the NAtl jet is only shifted approximately 3º further north compared to its position in the FullyTrans.

4. The ocean state has an effect on the position of the jet on the eastern side of the NAtl. The differences in the eastern position of the jet between FullyTrans and PDTopo from 4ka BP onward indicate that although their forcings are the same or very similar, the history of the simulation affects the position of the jet on the eastern side of the NAtl. Such a difference is not apparent on the western side of the jet.

What is not clear from the simulations presented here is the role of Eurasian ice topography and extent (ie versus that of the NAIS) and the relative role of background temperature versus temperature gradient. Given the length and content of the current paper, these last questions will be answered in a future study.

*Lack of dedicated discussion section:The authors have decided to jump straight from the results section to the conclusions without having a proper discussion section where your findings are put in perspective with the existing literature. This makes it hard to get a sense for how your results differ from earlier studies and how they contribute to our understanding of the atmospheric circulation during the deglaciation. The discussion section is in my mind the most important part of a paper, so*

*its omission feels instinctively wrong and may give this paper less traction than it deserves.*

We will add on a subsection to the Results and Discussion section where we will more clearly summarize our results in the context of the existing literature and discuss implications.

*Page 4, line 23: Sentence starting with "These changes.." is incorrect. To the best of my knowledge, none of the papers cited here suggest that the subtropical and midlatitude jets entered a merged state over the N Atlantic at the LGM; see, e.g., Fig. 1 in Li and Battisti (2008) where there is a clear separation between the subtropical and eddy driven jets. Page 4, lines 26-28: None of the papers cited here investigated that explicitly.*

We see that the citation here was misleading in that it appears to attribute the identification of merged jets to those papers, rather than just the data from which we concluded the change in distribution of heat and moisture transports. We will remove the citation altogether and make it clear that these are the authors' inferences based on those papers.

Nevertheless, we would argue that the separation between the jets at LGM is not as clear in all cases as the reviewer suggests. For example, in Figure 3 of Li and Battisti (2008), there is little difference in the Atlantic zonal wind profiles of the latitude of the peak winds at 200 hPa and at the surface differing at LGM (indication of the positions of subtropical and eddy-driven jets in Eichelberger and Hartmann (2007)). In Figure 2 of Lofverstrom et al (2014), the 800hPa winds across the North Atlantic during LGM are highly zonal, and there is little evidence of any separation between the subtropical and eddy-driven jets. Finally, in Figure 4 of Merz et al (2015), there appears to be a clearer separation between the subtropical and eddy-driven jets at LGM than during Present Day, or a less merged jet state, much as in our study. Delineating whether a jet is merged or not is problematic, since it is based on the separation of the distributions of the subtropical and eddy-driven jets. How much separation can there be and still be considered merged? The only definition for a merged jet that I have encountered is in Harnick et al (2014), who define the Zonal Jet Index as the anomaly with respect to monthy or seasonal climatology of the maximum value of the zonal derivative of the latitude with peak zonal winds. They

define a threshold for the jet to be merged as being a negative value in this derivative with time that exceeds one standard deviation. Perhaps it will be more helpful for us to replace the text in question with the following. "These changes are consistent with what would be expected if the North Atlantic jet approached a more "merged" state at LGM, although not all of these studies show evidence of this. Irregardless, changes to the path of the jet are expected to result in changes in the distributions of heat and precipitation over Western Europe during this time."

*Page 4, line 32: "The timing.." meaning here is not clear.*

We state in the previous sentence that the jet transition detected by Lofverstrom and Lora (2017) occurred at the separation of the Cordilleran and Laurentide ice sheets at 13.89ka BP. A transition at 13.89 ka BP would be unlikely to explain either the abrupt warming into the B-A (occurring nearly a millennium earlier), nor the abrupt cooling at the start of the YD (occurring a millennium later).

*Page 5, line 5 and section 2.1: Perhaps nit-pick but this not technically correct. PUMA (the dynamical core of PlaSim) is indeed a dry primitive equation model. However, the extra layer of physics on top of the dynamical core makes PlaSim more than a primitive equation model. More correct to say that it is a simplified general circulation model or, better yet, an Earth-system model of intermediate complexity (EMIC).*

Most importantly, we would like to point out that PlaSim solves the moist primitive equations, not the dry. We agree with the referee that PlaSim is more than just a primitive equation model but so is any current generation Earth System Model. Our main goal in pointing out that the foundation of the atmospheric model is based on the moist primitive equations is to illustrate that the dynamics have not been simplified beyond what is common in many Earth System Models today. Rather, the simplifications in the atmospheric model arise mainly in the parametrizations included: no treatment of volcanic or anthropogenic aerosols, only a single greenhouse gas species explicitly accounted for, etc. We will revisit how we describe the model in this manuscript to make the above clear, but we intentionally avoided the term EMIC. EMIC has become a vague term encompassing a wide range of models with different combinations of sophisticated and simplified components.

*Page 5, line 29: The description here is not correct. The Gaussian grid is the 128 x 64 cell grid in real space that the data is outputted on ("Gaussian" refers to how the grid is generated). The primitive equation are partially solved in spectral space (wave space) and are thus transformed between grid space (on the Gaussian grid, in this case 128 x 64 grid points in lon x lat) and the spectral representation in wave space, which supports at most 42 harmonics in the zonal and meridional direction, respectively. (Hence the name T42, where the T is short for "truncation" or more specifically "triangular truncation").*

We agree with the referee and will correct the wording in the text accordingly.

*Page 6, line 1: Not sure if I understand how the LSG models works. Is it a dynamic model that only runs in the mixed layer? If yes, how can a realistic ocean circulation be established if there is no deep ocean? Do you parameterize fluxes between the deep ocean and mixed layer? If yes, how are these fluxes calculated? What is the depth of the mixed layer? Prescribed or dynamic? Studies have shown that the mixed layer depth was substantially greater at the LGM (e.g. Sherriff-Tadano et al., 2018), which can have profound implications for the ocean heat contents and energy exchange between the ocean and atmosphere.*

LSG is a three-dimensional general circulation model for the entire ocean. It solves the primitive

equations for the ocean under assumptions of large spatial and time scales, which filters out relatively fast components like Kelvin waves. Given these assumptions and that LSG solves its equations implicitly, the time steps used are much longer than would be commonly used for other dynamical ocean models (in this study, approximately 4 simulation days). However, in order to allow the model to respond more quickly than this to abrupt or short-lived changes at the ocean's top surface, a mixed-layer ocean model is used as an intermediary between LSG and the rest of the model. The mixed-layer model is fixed to a 50m depth, which corresponds to the depth of the top layer of LSG. LSG itself does not have a fixed mixed-layer depth, and fluxes between the deep ocean and the mixed-layer are calculated as part of an LSG integration. The mixed-layer ocean model is made to relax gradually toward the LSG solution over LSG's timestep via an applied bottom-boundary heat flux, but it is also free to respond to changing thermal forcings at its top boundary from the atmosphere and sea ice. The above details have been added to the supplement (or appendix).

*Page 6, line 17: Please clarify, you update the boundary conditions every simulation year, but with 10x acceleration, meaning that you effectively only run every 10 years from LGM to PI. Is that correct?*

As far as the forcings go, the referee's description is correct. However, since the model is run continuously forward in time under these accelerated conditions, the solution will not be the same as if we extracted one of every 10 years from an unaccelerated simulation. The differences are described in Appendix A2 and are most noticeable in phenomena with a decadal or longer response timescale. For example, if the ocean's mixed layer takes approximately 30 simulation years to fully adjust to a change in atmospheric boundary conditions, the forcings will have progressed 300 years in this time. Thus, the timescales of these responses will appear lengthened.

*Page 7, line 12: "...temporal resolution of 100 years". This seems to conflict with the synchronous update of boundary conditions described above.*

As stated on page 7, lines 1-3, not all of the boundary conditions were available at annual or decadal timescales. Where they weren't available, we interpolated their values linearly in time between available bracketing time points. Thus, all of the boundary conditions were updated at the start of every simulation year.

*Page 7, line 5: Please clarify how this process works. You can't fit an even number of 0.5° grid cells in a T42 cell (which is around 2.8°), so there must be some partial overlapping cells. Also, what does "effective higher-resolution grid cell length" mean?*

We calculated the variance of higher-resolution grid cells within the T42 grid cells in the following manner. Writing the variance as the SUM(x^2)-(SUM(x))^2 and using conservative remapping as an area-weighted sum over overlapping regions in grid cells between the two grids (for a description of conservative remapping, see Jones, 1999), we conservatively remapped both the elevation and the square of the elevation from the higher-resolution grid to T42. We then squared the remapped elevation and took the difference as in the equation above to get the variance. The number of high-resolution grid cells contributing to this variance was approximated by the ratio of the total number of grid cells in the global high-resolution grid divided by the number of T42 grid cells. The effective higher-resolution grid cell length was then defined by dividing the area of the T42 grid cell evenly over the number of high-resolution grid cells contributing to the variance and taking the square root (assuming each of these grid cells are squares).

Page 8, line 22: Century should probably be millennium here (you discuss 21 ka – 20 ka and 1 ka to

1950), right?

Since the PlaSim simulations were generated with the forcings accelerated in time by a factor of ten, a millennium of forcing changes elapsed during the first or last centuries of the simulations. Thus, we were analysing 100 years of output and comparing them against a millennium of data from unaccelerated simulations.

Page 16, line 15: What standard metrics?

"Standard" here is intended to mean "commonly-used." The definitions employed here for jet latitudinal position and tilt arise from Woollings et al (2010) and a combination of Woollings and Blackburn (2012) and Lofverstrom and Lora (2017), respectively. However, variants on these definitions have been used in many of the papers discussed in the Introduction.

*Page 17, line 5: Meaning here is not clear. Do you mean flat ice sheets (i.e., only accounting for the albedo effect)? Also, the ice sheet height is not the only thing influencing the circulation. As you say elsewhere, the spatial extent is also important.*

The PDTopo experiment is defined with all forcings varying in time except the ice sheet thickness. The thickness of the ice sheets are fixed to present-day values, so the land elevation remains the same as during present day at all times. Nevertheless, the ice sheet area varies from an LGM extent to present-day (i.e an infinitessimally thin ice sheet). This allows the role of the ice sheet orography to be separated from the influence of its albedo and the rest of the forcings.

We do not claim that ice sheet height is the only thing influencing the circulation. Rather, the text says, "The component of the ice sheets that appears most important to this effect is their elevation." We support this claim by noting in the results of the PDTopo experiment (with time dependent ice area) that at LGM, the primary location of the jet is not shifted equatorward and is only slightly more focussed than during present-day. Since we know that the ice sheet is the primary control for the western side of the jet being equatorward-shifted and highly focussed via the results of the FixedGlac experiment (where all other forcings vary in time, but the western side of the jet remains predominantly in the same state throughout the deglaciation), this suggests that either the ice sheet orography or the combined effect of the orography and area are creating these effects. Since our DarkGlac experiment differed very little from the full-forcing runs due to extensive snow cover, we can not differentiate between these two possibilities. However, we can say that an elevated ice sheet is required for this effect .

*Page 18, line 1: How did you arrived at this specific number (725 m)?*

We determined the elevation threshold above which the jet appeared latitudinally restricted by empirical testing. The southernmost latitude of the ice sheet in eastern North America was identified by applying a mask for regions at or above the elevation value being tested and identifying its southernmost latitude value in this region. Then, as in Figure 12, the North Atlantic jet latitude for each month was plotted against the corresponding ice sheet minimum latitude. If the jet latitude ever exceeded the ice sheet latitude in any of the FullyTrans simulations, then that elevation value was rejected as the threshold.

*Page 19, lines 3-10: This explanation is a bit too simplistic. I agree that the presence of the ice sheet constrains the jet latitude in the west, presumable in part because of obstruction of the flow by the*

*topography. However, the thermal gradient at the southern ice margin can influence the flow in a similar fashion (this is not mentioned here as far as I can see) – both the change in albedo at the ice sheet margin, and the adiabatic cooling of the flow by the implied elevation difference. The modern (PI) jet is also less variable in the western ocean basin because of the strong thermal gradient at the sea-ice edge. This is clearly a different mechanism than the presence of a big ice sheet, but the effect is similar.*

We reject the hypothesis that the albedo change along the southern margin of the ice sheet plays an important role in constraining the jet that we detect over North America, because there is no such constraining effect in the PDTopo experiment (See Figure 12).  The PDTopo experiment includes a time-evolving, but infinitessimally-thin ice sheet, so the albedo change along the ice sheet's southern margin varies the same way in time in the PDTopo experiments as it does in the FullyTrans runs. Thus, the ice sheet must be elevated in order for this barrier effect to occur.   Our analyses can not distinguish between whether the elevated barrier operates via a dynamical effect alone or also via a thermal effect associated with the temperature gradient along the southern margin of the ice sheet.

*Page 21, line 25:  Meaning here is not clear – this seems to be the definition of a shift in the jet latitude.*

We agree that the wording used does not effectively bring out our point.  What we intended to argue here is that the frequency maps calculated from the Trace-21ka data do not show any change in the preferred jet tilt between 14 and 13ka BP.  These results are contrary to what is suggested by the results of Lofverstrom and Lora (2017), but there are two important differences in the methodology here from what they used.  Firstly, although both studies examine the Trace-21ka data, the abrupt increase in jet tilt presented in Lofverstrom and Lora (2017) was detected at 250 hPa, whereas we analysed jet changes over 700-925hPa.  Secondly, the jet tilt was defined in Lofverstrom and Lora (2017) as the difference in jet positions between 10-20ºW and 70-80ºEW, whereas we defined the jet tilt as the difference between 0-30ºW and 60-90ºW.  When we alter our analysis conditions to match those of Lofverstrom and Lora (2017), we see that the preferred angle of jet tilt does not change around 13.9 ka BP.  Rather, the frequency of time spent at this tilt and less tilted values decreases, while the range and frequency of tilts increases for more positive values.  This combination of phenomena matches the abrupt increase in mean jet tilt presented by Lofverstrom and Lora (2017).

*Page 22, line 16:  What Figure is discussed here?*

Figure 13, bottom left plot.

*Page 22, line 25:I would encourage you to think a little bit more about this and try to give a mechanistic explanation for this phenomena. Doesn't have to be a full explanation, but at least something that adds a little bit more to the story.*

We will expand on our discussion as discussed at the start of this document.

*Figure 1:  What ice sheet remained in North America through the Holocene and is 1.5 km thick?*

Figure 1 plots the peak elevation of ice sheet-covered areas in North America and Fennoscandia.  There is no 1.5km thick ice sheet over North America during the Holocene, but there are regions that have an elevation in excess of 1.5km that are covered by ice: glaciers in the Rockies, for example.

*Figure 4: Panels showing LGM and past1000 are mixed up (LGM is shown in middle panels).*

This is correct. We will fix the labels.

*Figure 9: 10 successive years is a bit ambiguous because it can be done in at least two different ways: (1) a sliding mean where the input and output arrays have the same length; (2) form decadal averages where the input array is 10x longer than the output array. These methods will yield slightly different results. I doubt that the difference will be of su#cient magnitude to challenge your conclusions, but this type of information is important for reproducibility.*

Neither of these methods were used to generate the plots of Figure 9 as no averages were performed. Instead, monthly jet latitudes and tilt were collected for every month in DJF for 10 successive years. Then, the fraction of time that the jet spent in each latitude/tilt bin was calculated by summing over the number of months with a jet in the given bin and dividing by the total number of months in the sample (= 30 months).

*Figure 11: Caption appears to be wrong as you show latitude here, not difference in latitude across the N Atlantic.*

The reviewer is correct. The caption will be changed.

Heather Andres and Lev Tarasov

---

## Referee Comment (RC2) · Anonymous Referee #2 · 24 Jan 2019

Towards understanding potential atmospheric contributions to abrupt climate changes: Characterizing changes to the North Atlantic eddy-driven jet over the last deglaciation by Andres and Tarasov.

In this manuscript the Authors describe a suite of experiments that show how the North Atlantic jet may have evolved over the last deglaciation. The Authors present results from simulations in which individual boundary conditions are changed in order that their effect on the circulation can be described. Such an approach is to be applauded as it makes clear the role that the various boundary conditions play overcoming the hand-waviness that exists in the analysis of many paleoclimate model simulations. Furthermore, the Authors compare their results directly with other model simulations to highlight the role that model physics might play. Overall it is a very thorough study.

The authors show that the evolution of the eddy driven jet is different on the eastern and western side of the Atlantic, with changes on the western side caused by the ice sheet topography, and changes on the eastern side influenced by all glacial forcings.

Overall I learned a lot from this paper and it is a great addition to the paleoclimate literature. It was, however, let down by some rather weak analysis in parts. There are also a number of sections that would be well to be rewritten as it was only possible to get the meaning on the third or fourth reading.

I detail below some specific comments and suggestions.

**Introduction**

The Introduction is quite variable. The description of the North Atlantic jet is great, and gives a reader unfamiliar with the topic a good start. I don't understand why there's an extensive description of the mechanisms for abrupt change in the climate during the deglaciation, because much does not appear to be relevant for the subsequent analysis. For example, it's only in the Introduction that changes in the AMOC are mentioned - why include this? I suggest focussing the Intro only on information that is pertinent for the paper itself.
The Intro, as written, just stops. It lacks a clear statement of the problem that this paper addresses. It conveys much information but doesn't link it together: how does jet variability have the potential to impact the climate during the deglaciation, especially abrupt changes? How is the paper going to answer this question?

**Section 3.2**
What is the point in this section? At the end of this section I don't konw what I am supposed to have learned. It would help to have a clear finishing paragraph to summarise this section. At present it is a selection of disparate facts. If you focus on the observations which support you summary, noting any discrepancies it may help add a narrative structure. It would help if you were to link the discussion here with the discussion comparing the jet in the east and west to the discussion of tilt - how do the changes at either end affect tilt?

**Tilt v east/west changes**
Reading the manuscript one gets a sense that the tilt of the jet and the changes in the jet in the east and west are two separate entities, which they evidently aren't. Historically, the focus has been on tilt, so it makes sense to at least look at this. However, I really like the description of how the east and west vary as this adds nuance to the very simplistic view of tilt. What's lacking the manuscript though is much of a bridge between the two. How can we interpret the previous discussions about tilt in the context of your results?

**General comments**

I'm not sure "oscillation" is a useful term to describe periods like the Younger Dryas. Oscillation implies some set of physics that gives a distinct cycle, and it's not clear to me that any changes which occur over the last deglaciation can be described as cycles. A better term would be "variability", which encapsulates the fact that there are changes without implying any cyclical physics. You even make this point on page 2 line 28. The presence or absence of cycles is not important for the interpretation in the paper so I'd suggest the less loaded term "variability"|.

p3 l5 - *"During the deglaciation, changes to the variability of low-level winds can alter gyre transports (and to a lesser degree, wind position and strength"* Not sure I understand this, surely variability of the low level wind **is** changes in the wind position and strength?

p3 l27 - jet not yet.

p5 l29 - how many ensemble members are there?

p5 l30 - *"Gaussian grid that is then used for diabatic calculations"* not sure what diabatic calculations are.

p8 l22 - You don't mention anything about model spin up? How is the model initialized?

p9 l7 - I'm not sure what *"peak zonal winds"* means. Perhaps just say strongest winds?

p9 l16 *"the pattern of wind changes from the LGM to the past1000 are similar in both CMIP5 and PlaSim runs (including during JJA, not shown) even though the differences are stronger in the PlaSim transient simulations."* Saying transient simluations at the end of this sentence makes it sound like there's an extra set of simulations - "the transient simulations" - as well as the normal "runs". This is not the case?

p10 l9 - I'm not sure how you get from *"10 consecutive DJF periods"* to the histogram on figure 6. 10 DJF periods surely gives 30 months, yet the frequencies on Fig 6. show 100s of months. Is this due to all the ensemble members?

p12 l6 - *"tilt .. shifts slightly higher"* - better as tilt becomes steeper.

p12 l7 - It would be interesting to compare the spread in histograms between Trace21ka and PlaSim. You discuss spread for PlaSim on p11, why not Trace21ka too?

p12 l14 - There needs to be a discussion of "abrupt". What constitutes abrupt: changes over what timescale? When looking at jet shifts in a coarse resolution model, a movement from on one grid box to the next come across as abrupt, tipping point like, but is actually just a smooth rapid change. To be abrupt suggests some set of non-linear feedbacks giving a larger response than the input would suggest. A linear response to a large change to me is not abrupt, just rapid. This is totally personal, but to avoid anyone misinterpreting what **you** mean by abrupt you need to define it.

P12 - I don't see a figure which shows the temperature over Greenland. Why are you showing the temperature over Greenland, anyway. What does it represent in this study looking at jet shifts.

p13 l5 - What exactly are you saying in this paragraph? I don't see what the point is.

p14 - Transition seems to be used to describe two different things here. From line 8 there are the three events called transitions then on l10 transition is used to describe the way things change. It doesn't help that on l10 it says *"A second type of transition"* without ever being clear what the first type of transition is. This whole paragraph l2 onwards is really difficult to understand. I'm struggling to be more helpful and think of suggestions to improve it, but a reader is really going to struggle with this.

p15 l4 - This seems to contradict p11 l5 which says that the jet tilt doesn't change much from LGM to past 1000.

p15 l13 - *"The preferred latitude shifts northward twice within a single decade of simulation, at 19.3ka BP and 14.6ka BP. The timing of these transitions match the more gradual shifts in the jet as a whole and two occasions when the tilt is reduced. They are also consistent with the historical timing of the start of the OD and B-A."* I do not understand this pair of sentences.

p16 l14 *"This separation makes it much easier to identify what changes are occurring and attribute their causes than examinations of the mean jet position over the entire range or its tilt.. "* This is an awkward sentence.

p17 l4 - this is an interesting point, any conjecture as to why orbit and GHG matter?

p18 l10 - *"Yet, the jet does not always move to the latitude of the jet"* Is an odd sentence.

p19 l3 - *"The consequences of this restriction are that the western end of the jet is more focussed relative to the eastern side, particularly when the NAIS extends well into the midlatitudes, and that the northern range of the western side of the jet increases much more over the deglaciation than its southern range"*
Perhaps rewrite as: "There are two consequences of this restriction. First the western end of the jet is more focussed relative to the eastern side, particularly when the NAIS extends well into the midlatitudes. Second, the northern range of the western side of the jet increases much more over the deglaciation than its southern range"

p19 l10 - *"The preferred tilt in the PDTopo experiment is near zero"*. The tilt doesn't change much in this simulation and stays at its past1000 value, which you show and argue does have a tilt, 5 degrees in Fig. 8. I agree that the ***change*** is near zero, but not that there is no tilt.

p19 l15 - *"In contrast, the shift in latitudes occurs earlier on the eastern side of the jet, and no further change to the preferred range of jet latitudes occurs following this."* This sentence exemplifies why, I think, this paper is so confusing. What is important in this sentence is *"the eastern side of the jet":* this is what is being compared to the preceding sentence. Yet the way that this sentence is structured puts this midway through the sentence, slightly buried. Thus it takes very careful reading to parse the sentence. If you wrote it as: "In contrast, on the eastern side of the jet the shift in latitudes occurs earlier..." it would be much more obvious what's going. It may be that this reviewer is a bit stupid, but with the long sentences that you use any help that a reader can get would be good. I'd have a look through the paper for more instances of this inverting of sentences. Given the quality of the English in this review you may, however, choose to ignore the stylistic recommendations of this reviewer.

p22 l1 - This section of the conclusions relates to the weakest part of the main text, much of what is in this part of the conclusions was not clear from the preceding sections.

p22 l10 colon inappropriate here.

p22 l21 - it would be clearer to say "through two phenomena: first, .... second ...." the two descriptions are so long you need to make it clear where one stops and the other starts.

p23 l7 - *"Conversely, the sensitivity of the jet position on the eastern side of the North Atlantic to the background climate state implies that it would be difficult to estimate historical changes to the jet in this region from model simulations, since estimates would vary between models and between simulations with different boundary conditions."* Need to explain this more. Surely, because the orbit and GHG are better constrained than ice sheets, this response will be better simulated?

p23 l12 - I disagree. The assumption here is that the surface temperature response is linear with respect to the jet latitude: as jet latitude increases so will temperature in direct proportion. But, if there are non-linear feedbacks it could be that when the jet reaches a certain latitude abrupt changes in temperature are possible. For example, imagine the jet is well south of the sea ice margin but gradually moves north due to a slow retreat of the ice sheet. At some point the jet will be over the sea ice margin and a different set of feedbacks become possible. Thus you can get abrupt changes in temperature from a smoothly varying jet/ice sheet.

Figure 1(a) - There's an enormous remnant ice sheet over NAm: 1.7km at 0ka. What is this?

Figure 5 - Its hard to judge in this figure how the amplitude of the jet changes. If you highlight one isotach, 20m/s, in both the contours and colours it would make it simpler to see how the structure of the jet differs.

---

## Author Comment (AC2) · 22 Apr 2019

The authors thank Anonymous Referee #1 for his/her considered comments.

Reviewer's comments are provided in italics.

*Lack of explanations : The authors provide an explanation for the suppressed variability of the western side of the jet stream (though I personally think that one key aspect is missing - see comment below). However, the more complicated part of the story, explaining the jet characteristics over the eastern ocean basin, is not discussed at the same level of detail. I think this is a shame and I encourage the authors to do more analysis to improve that part of the story.*

We agree with this reviewer that it is important to disentangle the various possible contributors to deglacial jet changes on the eastern side of the North Atlantic (NAtl). However, this can't be done to the same level as the western jet with the existing set of ensemble runs. The potential contributors that we have considered to the eastern jet changes include:

- Stationary and transient eddy responses to ice sheet topography changes,
- Thermal effects on latitudinal surface temperature gradients in the North Atlantic (and possibly sea ice) due to ice sheet area and height (over North America and Fennoscandia), greenhouse gas concentrations and orbital forcing,
- Indirect effects associated with the pinning of the western side of the jet to a particular latitude, and
- Indirect effects associated with ocean circulation responses to the boundary condition changes.

We will add to the revised text the following conclusions that we can draw based on our existing runs.

1. The presence of elevated and extensive ice sheets provides the dominant controls on the jet characteristics on both sides of the NAtl jet. When the ice sheets are fixed to their LGM configurations in FixedGlac, little change is seen in the position or distribution of the jet on both its western and eastern sides. One notable exception to this is after 4ka BP, when the eastern side of the jet shows a reduced frequency at its preferred latitude and an expansion of its range. This timing is coincident with abrupt warming events in the FixedGlac experiments that are accompanied with abrupt retreats in Northern Hemisphere sea ice extent and warmings of NAtl sea surface temperatures.
2. The eastern side of the NAtl jet is more sensitive to the background climate state than the western side of the jet is. This is evident when examining the differences between the FullyTrans and FixedOrbGHG experiments. By present day, the western side of the jet is mostly unaffected by the orbital and GHG components being fixed to LGM values. In contrast, the eastern side of the jet is centred approximately 5° further south when orbital and greenhouse gas components are fixed to LGM.
3. The position of the western side of the jet has limited control on the position of the eastern side of the jet. This conclusion is arrived at by examining the PDTopo experiment, where the position of the western side of the jet is approximately 8° further north than in the FullyTrans experiments before 20ka BP. In these same experiments at that time, the preferred eastern position of the NAtl jet is shifted approximately 3° further north.
4. The ocean state has an effect on the position of the jet on the eastern side of the NAtl. The differences in the eastern position of the jet between FullyTrans and PDTopo from 4ka BP onward indicate that although their forcings are the same or very similar, the history of the simulation affects the position of the jet on the eastern side of the NAtl. Such a difference is not apparent on the western side of the jet.

What is not clear from the simulations presented here is the role of Eurasian ice topography and extent

(ie versus that of the NAIS) and the relative role of background temperature versus temperature gradients. Given the length and content of the current paper, these last questions will be answered in a future study.

*Lack of dedicated discussion section:The authors have decided to jump straight from the results section to the conclusions without having a proper discussion section where your findings are put in perspective with the existing literature. This makes it hard to get a sense for how your results differ from earlier studies and how they contribute to our understanding of the atmospheric circulation during the deglaciation. The discussion section is in my mind the most important part of a paper, so its omission feels instinctively wrong and may give this paper less traction than it deserves.*
We have changed the Results and Discussion section to Results and changed Conclusions to Discussion and Conclusions. In the Discussion and Conclusions, we have provided much more context for our results in light of existing literature and discussed its implications.

For example, "All of the fully-transient deglacial simulations presented here show that the NAtl eddy-driven jet shifted northward from the Last Glacial Maximum to the preindustrial period, and its latitudinal variability increased. These characteristics match those derived from other studies (Li and Battisti, 2008; Lofverstrom et al., 2014; Merz et al., 2015). However, unlike those studies, neither the PlaSim simulations nor TraCE-21 show much change in jet tilt between these two periods."

"The novelty of this present study compared to those previously mentioned is that it analyses transient changes in the jet over the entire deglaciation in multiple experiments. Additionally, instead of characterizing the jet via the Gaussian statistics of its mean and standard deviation, we present the changes with time of the jet distribution itself."

*Page 1, line 5: Remove "we performed" and specify that PlaSim is an intermediate complexity model.*
We have made the description more precise. Revised text is "This study characterises deglacial winter wind changes over the North Atlantic (NAtl) in a suite of transient deglacial simulations using PlaSim, an intermediate-resolution earth system model with simplified physical parametrizations, and the TraCE-21ka simulation."

*Page 2, line 17: Missing space between sentences*
There is a space in the latex source file. It is not being rendered in this case, due to the number of characters on the line.

*Page 3, line 12: "Low levels of the atmosphere" is ambiguous. Perhaps better to say "lower troposphere".*
Done. Revised text is "Of the two dominant features of mid-latitude atmospheric circulation patterns in the NAtl, the subtropical and eddy-driven jets, the eddy-driven jet has the largest presence in the lower troposphere."

*Page 3, line 13: Missing word. The extratropical jet stream is due to "momentum-flux convergence" by synoptic eddies.*
Done. Revised text is "The eddy-driven jet (or polar front jet or jet stream) is a narrow band of fast, westerly winds that arises from the momentum-flux convergence of atmospheric synoptic-scale eddies (e.g. extratropical cyclones with lifespans of days) (Lee and Kim, 2003; Barnes and Hartmann, 2011)."

*Page 4, line 11: Lofverstrom et al. (2016) showed that stationary waves can influence the jet*

*characteristics in the eastern North Atlantic at the LGM as well.*

Revised text is "However, the influences on the western and eastern sides of the NAtl have been found to differ in atmospheric simulations: the position of the western side of the jet is more affected by ice sheet orography, and the position of the eastern side of the jet is primarily influenced by stationary and/or transient eddies (Kageyama and Valdes, 2000; Lofvestrom et al., 2016)."

*Page 4, line 23: Sentence starting with "These changes.." is incorrect. To the best of my knowledge, none of the papers cited here suggest that the subtropical and midlatitude jets entered a merged state over the N Atlantic at the LGM; see, e.g., Fig. 1 in Li and Battisti (2008) where there is a clear separation between the subtropical and eddy driven jets. Page 4, lines 26-28: None of the papers cited here investigated that explicitly.*

We see that the citation here was misleading in that it appears to attribute the identification of merged jets to those papers, rather than just the data from which we concluded the change in distribution of heat and moisture transports. We will remove the citation altogether and make it clear that these are the authors' inferences based on those papers.

Nevertheless, we would argue that the separation between the jets at LGM is not as clear in all cases as the reviewer suggests. For example, in Figure 3 of Li and Battisti (2008), there is little difference in the Atlantic zonal wind profiles of the latitude of the peak winds at 200 hPa and at the surface differing at LGM (indication of the positions of subtropical and eddy-driven jets in Eichelberger and Hartmann (2007)). In Figure 2 of Lofverstrom et al (2014), the 800hPa winds across the North Atlantic during LGM are highly zonal, and there is little evidence of any separation between the subtropical and eddy-driven jets. Finally, in Figure 4 of Merz et al (2015), there appears to be a clearer separation between the subtropical and eddy-driven jets at LGM than during Present Day, or a less merged jet state, much as in our study. Delineating whether a jet is merged or not is problematic, since it is based on the separation of the distributions of the subtropical and eddy-driven jets. How much separation can there be and still be considered merged? The only definition for a merged jet that I have encountered is in Harnick et al (2014), who define the Zonal Jet Index as the anomaly with respect to monthy or seasonal climatology of the maximum value of the zonal derivative of the latitude with peak zonal winds. They define a threshold for the jet to be merged as being a negative value in this derivative with time that exceeds one standard deviation.

Revised text is "These changes are consistent with what would be expected if the North Atlantic jet approached a more "merged" state at LGM, although not all of these studies show evidence of this. Irregardless, changes to the path of the jet are expected to result in changes to the distributions of heat and precipitation over Western Europe during this period."

*Page 4, line 26: Missing space between sentences.*
Done. This was an artifact of LaTex's paragraph formatting.

*Page 4, line 32: "The timing.." meaning here is not clear.*
We state in the previous sentence that the jet transition detected by Lofverstrom and Lora (2017) occurred at the separation of the Cordilleran and Laurentide ice sheets at 13.89ka BP. A transition at 13.89 ka BP would be unlikely to explain either the abrupt warming into the B-A (occurring nearly a millennium earlier), nor the abrupt cooling at the start of the YD (occurring a millennium later).

Revised text is "A jet shift at 13.89ka BP lies during the middle of the B-A,..."

*Page 5, line 5 and section 2.1: Perhaps nit-pick but this not technically correct. PUMA (the dynamical*

*core of PlaSim) is indeed a dry primitive equation model. However, the extra layer of physics on top of the dynamical core makes PlaSim more than a primitive equation model. More correct to say that it is a simplified general circulation model or, better yet, an Earth-system model of intermediate complexity (EMIC).*

Most importantly, we would like to point out that PlaSim solves the moist primitive equations, not the dry. We agree with the referee that PlaSim is more than just a primitive equation model but so is any current generation Earth System Model. Our main goal in pointing out that the foundation of the atmospheric model is based on the moist primitive equations is to illustrate that the dynamics have not been simplified beyond what is common in many Earth System Models today. Rather, the simplifications in the atmospheric model arise mainly in the parametrizations included: no treatment of volcanic or anthropogenic aerosols, only a single greenhouse gas species explicitly accounted for, etc. We intentionally avoided the term EMIC, because it has become a vague term encompassing a wide range of models with different combinations of sophisticated and simplified components.

Revised text, "PUMA is an atmospheric general circulation model whose dynamical core is based on the wet primitive equations. The primary simplifications in this component of PlaSim are found in the physical parametrizations incorporated in the model: for example, carbon dioxide is the only greenhouse gas whose radiative effects are considered and the radiative transfer scheme is much simpler (and thereby much faster) than that used in current state of the art GCMs."

*Page 5, line 28: It was recently shown by Lofverstrom and Liakka (2018) that T42 resolution is sufficient to reasonably capture planetary waves in simulations of the LGM climate.*
This reference was added. Revised text is "Herein we use 10 vertical levels at a spectral resolution of T42 (approximately 2.8°x2.8°), which has been previously shown to be sufficiently high to resolve phenomena of interest to the eddy-driven jet (Barnes and Hartmann, 2011; Lofverstrom and Liakka, 2018) while enabling fast enough model run times to make multiple deglacial experiments feasible."

*Page 5, line 29: The description here is not correct. The Gaussian grid is the 128 x 64 cell grid in real space that the data is outputted on ("Gaussian" refers to how the grid is generated). The primitive equation are partially solved in spectral space (wave space) and are thus transformed between grid space (on the Gaussian grid, in this case 128 x 64 grid points in lon x lat) and the spectral representation in wave space, which supports at most 42 harmonics in the zonal and meridional direction, respectively. (Hence the name T42, where the T is short for "truncation" or more specifically "triangular truncation").*

We agree with the referee. Revised text is "The dynamical atmospheric solutions are generated in spectral space, while the remaining calculations (e.g. phase changes, heat exchange with the land, sea ice or slab ocean, and any changes in those sub-components) occur in real space on a Gaussian grid with 64 latitude points and 128 longitude points. The only exception to this is LSG, which is run at 2.5°x5° horizontal resolution."

*Page 6, line 1: Not sure if I understand how the LSG models works. Is it a dynamic model that only runs in the mixed layer? If yes, how can a realistic ocean circulation be established if there is no deep ocean? Do you parameterize fluxes between the deep ocean and mixed layer? If yes, how are these fluxes calculated? What is the depth of the mixed layer? Prescribed or dynamic? Studies have shown that the mixed layer depth was substantially greater at the LGM (e.g. Sherriff-Tadano et al., 2018), which can have profound implications for the ocean heat contents and energy exchange between the*

*ocean and atmosphere.*

LSG is a three-dimensional general circulation model for the entire ocean. It solves the primitive equations for the ocean under assumptions of large spatial and time scales, which filters out relatively fast components like Kelvin waves. Given these assumptions and that LSG solves its equations implicitly, the time steps used are much longer than would be commonly used for other dynamical ocean models (in this study, approximately 4 simulation days). However, in order to allow the model to respond more quickly than this to abrupt or short-lived changes at the ocean's top surface, a mixed-layer ocean model is used as an intermediary between LSG and the rest of the model. The mixed-layer model is fixed to a 50m depth, which corresponds to the depth of the top layer of LSG. LSG itself does not have a fixed mixed-layer depth, and fluxes between the deep ocean and the mixed-layer are calculated as part of an LSG integration. The mixed-layer ocean model is made to relax gradually toward the LSG solution over LSG's timestep via an applied bottom-boundary heat flux, but it is also free to respond to changing thermal forcings at its top boundary from the atmosphere and sea ice.

Revised text is "Rather than specifying a fixed deep-ocean heat flux to the mixed layer ocean, PlaSim estimates these fluxes by executing LSG every 32 atmospheric time steps (equivalent to 4.5 days). LSG is a three-dimensional, global, ocean general circulation model that solves the primitive equations implicitly under assumptions of large spatial and temporal scales (Maier-Reimer et al.,1993). This formulation permits stable solutions on longer time steps than other components of PlaSim with the trade-off that it filters out gravity waves and barotropic Rossby waves (Maier-Reimer et al., 1993). Since the time steps of LSG are so long, a slab-ocean model is used as an intermediary between LSG and the rest of the model in order to allow the ocean to respond to abrupt or short-lived phenomena. The slab-ocean model is fixed to a 50m depth, which corresponds to the depth of the top layer of LSG. Thus, at the start of an LSG integration, fields in the top layer of LSG are initialized to those from the slab-ocean model, and heat fluxes and wind stress fields are read from the sea ice and atmosphere components, respectively. The LSG integration is performed, and a spatial map of differences between the mixed-layer temperature at the end and the start of the LSG time step are calculated. These temperature differences are used to define a map of deep-ocean heat fluxes, which are subdivided by the number slab-ocean time steps before the next LSG integration and applied as bottom boundary conditions to the slab-ocean model. Thus, under constant atmospheric conditions, the slab ocean model relaxes toward the LSG solution. Under changing atmospheric conditions, the surface component of the ocean will tend toward a mixture of the LSG solution and a thermal response to the surface forcing."

*Page 6, line 17: Please clarify, you update the boundary conditions every simulation year, but with 10x acceleration, meaning that you effectively only run every 10 years from LGM to PI. Is that correct?*

As far as the forcings go, the referee's description is correct. However, since the model is run continuously forward in time under these accelerated conditions, the solution will not be the same as if we extracted one of every 10 years from an unaccelerated simulation. The differences are described in Appendix A2 and are most noticeable in phenomena with a decadal or longer response timescale. For example, if the ocean's mixed layer takes approximately 30 simulation years to fully adjust to a change in atmospheric boundary conditions, the forcings will have progressed 300 years in this time. Thus, the timescales of these responses will appear lengthened.

Revised text is "This acceleration was not found to alter the main conclusions of this study when tested with a single unaccelerated run, but it is expected to lengthen the apparent timescales of processes. For example, if the ocean's mixed layer takes approximately 30 simulation years to fully adjust to a change in atmospheric boundary conditions, it will appear to take 300 years from the perspective of forcing changes."

*Page 7, line 12: "...temporal resolution of 100 years". This seems to conflict with the synchronous update of boundary conditions described above.*

As stated on page 7, lines 1-3, not all of the boundary conditions were available at annual or decadal timescales. Where they weren't available, we interpolated their values linearly in time between available bracketing time points. Thus, all of the boundary conditions were updated at the start of every simulation year.

Revised text is "These data are interpolated spatially to the model grid, with land-sea mask defined so the topography of ocean grid cells lies below the contemporaneous sea level. The data are also linearly interpolated in time in order to provide updates every simulation year (i.e. every 10 forcing years)."

*Page 7, line 5: Please clarify how this process works. You can't fit an even number of 0.5° grid cells in a T42 cell (which is around 2.8°), so there must be some partial overlapping cells. Also, what does "effective higher-resolution grid cell length" mean?*

We calculated the variance of higher-resolution grid cells within the T42 grid cells in the following manner. Writing the variance as the $SUM(x^2)-(SUM(x))^2$ and using conservative remapping as an area-weighted sum over overlapping regions in grid cells between the two grids (for a description of conservative remapping, see Jones, 1999), we conservatively remapped both the elevation and the square of the elevation from the higher-resolution grid to T42. We then squared the remapped elevation and took the difference as in the equation above to get the variance. The number of high-resolution grid cells contributing to this variance was approximated by the ratio of the total number of grid cells in the global high-resolution grid divided by the number of T42 grid cells. The effective higher-resolution grid cell length was then defined by dividing the area of the T42 grid cell evenly over the number of high-resolution grid cells contributing to the variance and taking the square root (assuming each of these grid cells are squares).

Revised text is "For each T42 grid cell, the roughness is equal to the variance of all 0.5°x0.5° ice sheet grid cells contained within it divided by an effective high-resolution grid cell length. This calculation is performed by first conservatively remapping the elevation and the square of the elevation from the higher-resolution grid to the lower-resolution grid. The variance is then the difference between the square of the remapped elevation and the remap of the squared elevation. The effective higher-resolution grid cell length is the square root of the area of the T42 grid cell divided by the number of higher-resolution grid cells per T42 grid cell (taken here to be the ratio of the total number of grid cells globally in each grid)."

*Page 8, line 18: Ivanovic et al. (2016) is double cited.*
The second citation has been removed. Revised text is "Data for $CO_2$ , $N_2O$ and $CH_4$ concentration changes over the deglaciation are consistent with the prescriptions of the PMIP4 Deglacial experiment (see Ivanovic et al. (2016) and data sources Luthi et al. (2008) Meinshausen et al. (2017), and Loulergue et al. (2008))."

*Page 8, line 22: Century should probably be millennium here (you discuss 21 ka - 20 ka and 1 ka to 1950), right?*
Since the PlaSim simulations were generated with the forcings accelerated in time by a factor of ten, a millennium of forcing changes elapsed during the first or last centuries of the simulations. Thus, we were analysing 100 years of output and comparing them against a millennium of data from unaccelerated simulations.

Revised text is "We compare the climate conditions during the first and last century of the fully-transient PlaSim simulations (corresponding to forcing years 21-20ka BP and 1ka BP to 1950AD, respectively, due to acceleration) to the results of LGM and past1000 experiments in the Climate Modelling Intercomparison Project (CMIP) 5."

*Page 9, line 10:  Sentence can be simplified; e.g.: "Also, the path of the NPac jet" -> "Also, the NPac jet"*
Done.  Revised text is "Also, the NPac jet is displaced further north and is more tilted in the PlaSim past1000 simulations."

*Page 9, line 14: I agree with this assessment and a similar conclusion was reached by Lofverstrom et al. (2016); see their discussion about sensitivity simulations with extensive sea ice in the eastern N. Atlantic (their Fig. 6).*
Done.  Revised text is "We speculate that this eastern shift is connected to the much more southern extent of sea ice on the eastern side of the NAtl, as was found in CAM3 simulations forced by present-day ice sheets with LGM sea surface temperatures and sea ice extent (Lofverstrom et al., 2016)."

*Page 16, line 6:  Write out explicitly that you are referring to Fig. 11 here.*
Done.  Revised text is "In contrast, in Figure 11 the eastern side of the jet over the eastern NAtl is less focussed than the west, with the jet occupying its preferred latitude 50 to 70% of the time."

*Page 16, line 5:  Typo? ...range or its tilt -> ...range of its tilt (?)*
The sentence has been reworded to avoid confusion.  Revised text is "Since these jet characteristics and the timing of their changes differ on the eastern and western sides of the NAtl jet, we attribute them separately in the next section."

*Page 16, line 15:  What standard metrics? Page 16, line 17:  Replace "instead" with "as well", and remove "For those interested".*
"Standard" here is intended to mean "commonly-used."  The definitions employed here for jet latitudinal position and tilt arise from Woollings et al (2010) and a combination of Woollings and Blackburn (2012) and Lofverstrom and Lora (2017), respectively.  However, variants on these definitions have been used in many of the papers discussed in the Introduction.

Revised text is "Due to this sensitivity and the important differences in jet characteristics in the two regions, we argue that analyses over the western and eastern jet regions are more instructive than the more commonly-used jet latitude and tilt metrics, and would encourage other authors to present these metrics as well."

*Page 17, line 5: Meaning here is not clear. Do you mean flat ice sheets (i.e., only accounting for the albedo effect)? Also, the ice sheet height is not the only thing influencing the circulation. As you say elsewhere, the spatial extent is also important.*

The PDTopo experiment is defined with all forcings varying in time except the ice sheet thickness. The thickness of the ice sheets are fixed to present-day values, so the land elevation remains the same as during present day at all times.  Nevertheless, the ice sheet area varies from an LGM extent to present-day (i.e an infinitessimally thin ice sheet).  This allows the role of the ice sheet orography to be separated from the influence of its albedo and the rest of the forcings.

We do not claim that ice sheet height is the only thing influencing the circulation. Rather, the text says, "The component of the ice sheets that appears most important to this effect is their elevation." We support this claim by noting in the results of the PDTopo experiment (with time dependent ice area) at LGM, the primary location of the jet is not shifted equatorward and is only slightly more focussed than during present-day. Since we know that the ice sheet is the primary control for the western side of the jet being equatorward-shifted and highly focussed via the results of the FixedGlac experiment (where all other forcings vary in time, but the western side of the jet remains predominantly in the same state throughout the deglaciation), this suggests that either the ice sheet orography or the combined effect of the orography and area are creating these effects. Since our DarkGlac experiment differed very little from the full-forcing runs due to extensive snow cover, we can not differentiate between these two possibilities. However, we can say that an elevated ice sheet is required for this effect.

Revised text is "Ice sheets provide the primary control over the deglacial jet changes described in the previous section. Simulations with fixed LGM ice sheets (FixedGlac in Figures 10 and 11) reproduce neither the deglacial changes to preferredjet latitude nor the bulk of changes to its variability on both sides of the jet. This effect is most prominent for the western side of the jet, which shows almost no change over the deglaciation when the ice sheets are fixed to their LGM state. In the east, the preferred position of the NAtl jet does not change under fixed LGM ice sheets, but the frequency of time the jet spends at this latitude decreases, and its range of variability increases. Only orbital and greenhouse gas forcings are changing at this time, so this may indicate a sensitivity to those forcings (perhaps mediated by the changing sea ice extent and sea surface temperatures).

Other sensitivity experiments can help decompose which attributes of the ice sheets are enacting this control on the NAtl jet. The PDTopo experiment isolates the thermal forcing associated with ice sheets' relatively high albedo from the orographic forcing due to the elevation of the ice sheet by fixing ice sheet thickness to present-day values while allowing ice sheet area to vary. Thus, LGM ice sheets are infinitessimally thin but extensive. In neither the east nor the west is the NAtl jet as focussed, or as equatorward-shifted in PDTopo (Figures 10 and 11) as it is at LGM in the FullyTrans runs or throughout the FixedGlac runs. Consequently, we conclude that the ice sheet albedo alone is not the primary controlling factor on the NAtl jets, and that the elevation of ice sheets is important. However, it is not clear from the present experiments whether orographic changes to the ice sheets alone are sufficient to explain the jet changes, or whether the ice sheets need to be reflective. Since the ice sheets became quickly covered with highly-reflective snow in the DarkGlac simulations, that experiment did not resolve this question. "

*Page 18, line 1: How did you arrived at this specific number (725 m)?*
We determined the elevation threshold above which the jet appeared latitudinally restricted by empirical testing. The southernmost latitude of the ice sheet in eastern North America was identified by applying a mask for regions at or above the elevation value being tested and identifying its southernmost latitude value in this region. Then, as in Figure 12, the North Atlantic jet latitude for each month was plotted against the corresponding ice sheet minimum latitude. If the jet latitude ever exceeded the ice sheet latitude in any of the FullyTrans simulations, then that elevation value was rejected as the threshold.

Revised text is " This number was arrived at empirically and represents the lowest threshold tested that did not have instances of the western side of the NAtl jet exceeding its location."

*Page 18, line 9: Typo? "jet does not always move the the latitude of the jet."*
Fixed. Revised text is "It should be noted, the jet does not always move to the latitude of the ice sheet

margin."

*Page 19, lines 3-10:  This explanation is a bit too simplistic. I agree that the presence of the ice sheet constrains the jet latitude in the west, presumable in part because of obstruction of the flow by the topography. However, the thermal gradient at the southern ice margin can influence the flow in a similar fashion (this is not mentioned here as far as I can see) - both the change in albedo at the ice sheet margin, and the adiabatic cooling of the flow by the implied elevation difference. The modern (PI) jet is also less variable in the western ocean basin because of the strong thermal gradient at the sea-ice edge. This is clearly a different mechanism than the presence of a big ice sheet, but the effect is similar.*

We reject the hypothesis that the albedo change along the southern margin of the ice sheet plays an important role in constraining the jet that we detect over North America, because there is no such constraining effect in the PDTopo experiment (See Figure 12).  The PDTopo experiment includes a time-evolving, but infinitessimally-thin ice sheet, so the albedo change along the ice sheet's southern margin varies the same way in time in the PDTopo experiments as it does in the FullyTrans runs. Thus,  the ice sheet must be elevated in order for this barrier effect to occur.   Our analyses can not distinguish between whether the elevated barrier operates via a dynamical effect alone or whether there is a role for the thermal effects of the ice to play.

Revised text is "Other sensitivity experiments can help decompose which attributes of the ice sheets are enacting this control on the NAtl jet.  The PDTopo experiment isolates the thermal forcing associated with ice sheets' relatively high albedo from the orographic forcing due to the elevation of the ice sheet by fixing ice sheet thickness to present-day values while allowing ice sheet area to vary.  Thus, LGM ice sheets are infinitessimally thin but extensive. In neither the east nor the west is the NAtl jet as focussed, or as equatorward-shifted in PDTopo (Figures 10 and 11) as it is at LGM in the FullyTrans runs or throughout the FixedGlac runs. Consequently, we conclude that the ice sheet albedo alone is not the primary controlling factor on the NAtl jets, and that the elevation of ice sheets is important. However, it is not clear from the present experiments whether orographic changes to the ice sheets alone are sufficient to explain the jet changes, or whether the ice sheets need to be reflective.  Since the ice sheets became quickly covered with highly-reflective snow in the DarkGlac simulations, that experiment did not resolve this question."

*Page 21, line 25:  Meaning here is not clear - this seems to be the definition of a shift in the jet latitude.*

We agree that the wording used does not effectively bring out our point.  What we intended to argue here is that the frequency maps calculated from the Trace-21ka data do not show any change in the preferred jet tilt between 14 and 13ka BP.  These results are contrary to what is suggested by the results of Lofverstrom and Lora (2017), but there are two important differences in the methodology here from what they used.  Firstly, although both studies examine the TraCE-21ka data, the abrupt increase in jet tilt presented in Lofverstrom and Lora (2017) was detected at 250 hPa, whereas we analysed jet changes over 700-925hPa.  Secondly, the jet tilt was defined in Lofverstrom and Lora (2017) as the difference in jet positions between 10-20ºW and 70-80ºW, whereas we defined the jet tilt as the difference between 0-30ºW and 60-90ºW.  When we alter our analysis conditions to match those of Lofverstrom and Lora (2017), we see that the preferred angle of jet tilt does not change around 13.9 ka BP.  Rather, the frequency of time spent at this tilt and less tilted values decreases, while the range and frequency of tilts increases for more positive values.  This combination of phenomena matches the abrupt increase in mean jet tilt presented by Lofverstrom and Lora (2017).

Revised text is "Note that this value is less than that calculated in Lofverstrom and Lora (2017) from TraCE-21ka data (between 3º and 4º), but they calculated jet tilt from upper-tropospheric winds and

different longitude ranges in the western and eastern regions of the NAtl jet (270 - 300°E and 330 to 360°E in this study versus 280 - 290°E and 340 to 350°E)." "This results stands in contrast to Lofverstrom and Lora (2017), who diagnosed a rapid increase in jet tilt at 13.89ka BP. The source of this discrepancy is discussed further in Section 4." "This contrasts with previous work on TraCE-21ka that identifies an abrupt increase in tilt in this dataset that occurred at 13.89ka BP (Lofverstrom and Lora, 2017). These two TraCE-21karesults were obtained from different wind data extracted from the same dataset: lower-tropospheric winds are analysed in this study, while Lofverstrom and Lora (2017) examine upper tropospheric winds at 250hPa. There are additional differences in the range of longitudes used to specify the western and eastern regions of the NAtl jet: 280 to 290°E and 340 to 350°E, respectively in Lofverstrom and Lora (2017) versus 270 to 300°E and 330 to 360°E in this study. We are able to reproduce the results obtained by Lofverstrom and Lora (2017) (except for the timing, which we date to 13.87ka BP) when we calculate the jet tilt in the same manner as they did (figures are presented in Supplemental Figure S5, alongside corresponding figures using the methodology employed in this study). "

*Page 22, line 16: What Figure is discussed here?*
Figure 12, bottom left plot. Revised text is "As in the PlaSim simulations, this change is driven by a shift on the western side of the jet, plotted in the bottom-left panel of Figure 12."

*Page 22, line 25:I would encourage you to think a little bit more about this and try to give a mechanistic explanation for this phenomena. Doesn't have to be a full explanation, but at least something that adds a little bit more to the story.*
We will expand on our discussion as discussed at the start of this document.

Revised text is "In contrast, the downstream side of the jet over the eastern North Atlantic does not show the same sensitivity to the marginal position of the North American ice complex, but it is affected by the presence of elevated ice sheets. Whether this control is exerted via changes to the stationary waves (e.g. Kageyama and Valdes (2000); Lofverstrom et al. (2014)), transient eddies (e.g. Merz et al. (2015)), surface thermal gradients from sea ice and sea surface temperatures (e.g. Li and Battisti (2008)), or some other mechanism is not entirely apparent from this study. There is some evidence that sea ice and sea surface temperatures may play a role, as the range of jet latitudes increases after abrupt sea ice retreats and sea surface temperature warmings in the FixedGlac experiment, and the eastern jet distribution is centred around different latitudes at the end of the deglaciation in FullyTrans and PDTopo even though their forcings are the same at this time. "

*Figure 1: What ice sheet remained in North America through the Holocene and is 1.5 km thick?*
Figure 1 plots the peak elevation of ice sheet-covered areas in North America and Fennoscandia. There is no 1.5km thick ice sheet over North America during the Holocene, but there are regions that have an elevation in excess of 1.5km that are covered by ice: glaciers in the Rockies, for example. Due to this confusion, we have changed Figure 1 to only include ice sheet area and elevation east of the Rockies.

Revised caption is "a) Peak elevation in ice sheet-covered areas (bedrock elevation plus ice sheet thickness) and b) ice sheet area for the Laurentide ice sheet and Eurasia (FIS). "

*Figure 4: Panels showing LGM and past1000 are mixed up (LGM is shown in middle panels).* This is correct. The labels are fixed in the revised version.

*Figure 4: The top of the LGM ice sheets (and indeed some modern topography) is higher than the 700 hPa isobar. I would advice against extrapolating the wind field in these regions and*

*instead treat it as missing data, as extrapolation can cause some weird effects when doing*
*statistical analysis (e.g. when determining the latitude of the strongest winds in the western*
*N Atlantic).*

The reviewer raises a good point. Since our analyses were performed on vertical averages of winds over 700-925hPa, where there is an increase in wind speed with height (Figure 5), it is not clear what pressure level should be used to create a mask of topography. To get around this problem, we recalculated the jet statistics for FullyTrans1 only looking at 850 hPa and omitting grid cells where the land surface lies above this pressure level. The results of these tests are provided in Supplemental Figure S9. The timing and characteristics of jet transitions in the western and eastern regions and for the jet tilt are unchanged from those presented in the main paper. However, when the jet is averaged over longitudes 270°E to 360°E, there are differences compared to the FullyTrans1 results in Supplemental Figure 3. These differences can be attributed to changes to the longitudinal range over which the winds are being averaged when grid cells with an elevated ice surface are masked from the analysis. Due to the differences in characteristics of the jet on its western and eastern sides, changing the weighting of these two regions with time by excluding a changing number of grid cells on the west leads to a mixture of jet characteristics in the mean jet. This issue highlights the problems with using this metric. However, since the conclusions of our study are not affected by the inclusion or exclusion of winds interpolated onto levels below the ice sheet surface, we leave our analyses as is.

Revised text is "Note that the ice sheet elevation in some grid cells exceeds the bottom pressure level of the analysis range, so we interpolated the winds to these levels to not bias the resulting average. The land surface does not pass above the 700hPa pressure level in any of the grid cells included in our analysis, so we are not introducing winds to a region where there was none. The impact of this choice was tested on the 850hPa level and found to not change the jet results (see Supplemental Figure S9) except when the jet is calculated over the entire longitudinal range. In this case, excluding grid cells from the analysis on the western side of the region effectively weights the mean jet toward characteristics of the eastern region. This issue illustrates the sensitivity of this diagnostic to the longitudinal range that is employed. Due to this sensitivity and the important differences in jet characteristics in the two regions, we argue that analyses over the western and eastern jet regions are more instructive than the more commonly-used jet latitude and tilt metrics, and would encourage other authors to present these metrics as well."

Figure 4 caption "Where the land surface impinges on the vertical range, winds are interpolated to not bias the vertical average. Further discussion can be found in Section 4."

*Figure 6 - 8: Use the same range on the spines on the right hand side for easier comparison (e.g., in*
*Fig. 6: 0-80 % in top panels and 0-35 % in lower panels).*
Done.

*Figure 9: 10 successive years is a bit ambiguous because it can be done in at least two different*
*ways: (1) a sliding mean where the input and output arrays have the same length; (2) form*
*decadal averages where the input array is 10x longer than the output array. These methods*
*will yield slightly different results. I doubt that the difference will be of su#cient magnitude*
*to challenge your conclusions, but this type of information is important for reproducibility.*
Neither of these methods were used to generate the plots of Figure 9 as no averages were performed. Instead, monthly jet latitudes and tilt were collected for every month in DJF for 10 successive years. Then, the fraction of time that the jet spent in each latitude/tilt bin was calculated by summing over the number of months with a jet in the given bin and dividing by the total number of months in the sample (= 30 months/per run * 4 runs = 120 months).

Revised text is "Frequency maps of NAtl, lower-level, jet latitudes and tilt aggregated over 10 successive winter seasons and all ensemble members of the FullyTrans experiment. Frequencies represent the percentage of months (out of a total of 120 months) that the jet was identified at a particular latitude, where each latitude bin has a width of 2.8° at T42."

*Figure 9 - 11 and 13: Write out lat and lon bounds and pressure level(s) used in statistics.*
Done.

*Figure 11:  Caption appears to be wrong as you show latitude here, not difference in latitude across theN Atlantic.*
The reviewer is correct.  Revised text is "Frequency maps of ensemble-average, NAtl, lower-level, eastern jet latitude in 10 successive winter seasons for FullyTrans, FixedOrbGHG, FixedGlac, and PDTopo experiments. Colours indicate the percentage of months with the difference in jet latitudes between 330°E to 360°E and 270°E to 300°E within each bin of width 2.8°."

*Figure 12:  Use same latitude range on vertical axis for easier comparison.*
Done.

---

## Author Comment (AC3) · 22 Apr 2019

The authors thank Anonymous Referee #2 for his/her detailed comments.

Reviewer's comments are provided in italics.

*The Introduction is quite variable. The description of the North Atlantic jet is great, and gives a reader unfamiliar with the topic a good start. I don't understand why there's an extensive description of the mechanisms for abrupt change in the climate during the deglaciation, because much does not appear to be relevant for the subsequent analysis. For example, it's only in the Introduction that changes in the AMOC are mentioned - why include this? I suggest focussing the Intro only on information that is pertinent for the paper itself. P12 - I don't see a figure which shows the temperature over Greenland. Why are you showing the temperature over Greenland, anyway. What does it represent in this study looking at jet shifts.*

The larger context of the paper is whether jet shifts over the last deglaciation may have contributed to the abrupt climate changes of that period. As we state in the introduction, "deglacial, winter wind conditions over the North Atlantic may have played important roles in the detected abrupt climate changes through their effects on sea ice and/or the surface ocean circulations." This is plausible, because "... altered winter sea ice extent is sufficient to reproduce the amplitude of deglacial climate changes in northwestern Europe and explain their seasonality (Renssen and Isarin, 2001). " This question is of potential importance to anyone who studies this period, whether they collect and analyse data or perform simulations. However, analyses of atmospheric phenomena under paleo timescales, and the behaviour of the jet in particular, tend to not get a lot of attention from this broader audience. Thus, we included information about the abrupt deglacial changes in the introduction and later in the results section in order to make this connection clearer and hopefully draw attention to this idea in the broader paleo community. As changes in AMOC play a central role in most hypotheses for explaining abrupt changes, we judge it necessary to include this.

Since the reviewer found these comments out of place, we have made some changes to the text to make this motivation stronger. Revised text in the introduction is "Proposed hypotheses explaining the presence of such variability during the deglaciation commonly centre around deep- water formation changes in response to freshwater anomalies in the regions where deepwater is formed (e.g. Rooth (1982); Broecker et al. (1985, 1989); Tarasov and Peltier (2005); Bradley and England (2008); Hu et al. (2010); Keigwin et al. (2018)). In simulations, abrupt reductions of the AMOC induced by hosing are successful at explaining the abruptness of cooling in the extratropical North Atlantic (NAtl), Nordic Seas, Arctic and Eurasia, the reduced precipitation in the NAtl and Europe, and the southward shift of the ITCZ (Kageyama et al., 2010, 2013). They are less successful at explaining the amplitude of tempera- ture changes over Greenland and Europe during the last deglaciation (Clark et al., 2012), particularly when hosing amounts are constrained to realistic values (Kageyama et al., 2010).

Freshwater forcing may not be the only driver of the abrupt climate changes of the last deglaciation. Modern Earth System Models (ESMs) are now exhibiting abrupt climate changes of similar magnitude under slowly-varying or constant boundary conditions (Knorr and Lohmann, 2007; Peltier and Vettoretti, 2014; Zhang et al., 2014; Brown and Galbraith, 2016; Zhang et al., 2017; Klockmann et al., 2018), due to the bistability of the AMOC (Stommel, 1961; Broecker et al., 1985; Knorr and Lohmann, 2007; Zhang et al., 2014, 2017) and/or thermohaline instabilities involving the interactions of the ocean, sea ice and potentially atmosphere (Knorr and Lohmann, 2007; Dokken et al., 2013; Peltier and Vettoretti, 2014; Brown and Galbraith, 2016; Vettoretti and Peltier, 2016; Klockmann et al., 2018; Vettoretti and Peltier, 2018). Winter sea ice extent changes alone are sufficient to reproduce the amplitude of deglacial climate changes in northwestern Europe and explain their seasonality (Renssen and Isarin, 2001), and changes to the surface ocean heat transports can affect the

rates of deepwater formation (Lozier et al., 2010; Häkkinen et al., 2011; Muglia and Schmittner, 2015). Since low-level wind patterns over the Arctic, Greenland and NAtl help constrain winter sea ice extent in this region (Venegas and Mysak, 2000) and affect surface ocean heat transports there through the application of surface wind stresses (Lozier et al., 2010; Häkkinen et al., 2011; Li and Born, 2019), these winds may play an important role in setting the conditions required for abrupt deglacial changes or be involved in the abrupt transitions themselves. However, in order to assess this potential, deglacial changes to lower-tropospheric winds must be first identified.

Of the two dominant features of mid-latitude atmospheric circulation patterns in the NAtl, the subtropical and eddy-driven 5 jets, the eddy-driven jet has the largest presence in the lower troposphere, and thus the most potential to change wind stress and thereby ocean circulation. "

Revised text in section 3.2 is "The motivation of this study is to assess the potential impact of atmospheric changes in the NAtl over the last deglaciation on the abrupt changes detected in Greenland ice cores, so we start by discussing climate changes over Greenland."

Revised text in the Discussion and Conclusions is " This paper explores the question of whether winter eddy-driven jet changes over the North Atlantic could have contributed to the abrupt climate changes detected in Greenland ice cores over the last deglaciation."

*The Intro, as written, just stops. It lacks a clear statement of the problem that this paper addresses. It conveys much information but doesn't link it together: how does jet variability have the potential to impact the climate during the deglaciation, especially abrupt changes? How is the paper going to answer this question?*

Done. Revised text is "In summary, previous work has shown that lower-tropospheric winds over the North Atlantic have the ability to alter sea ice extent and surface ocean circulations in a manner that can reproduce abrupt climate changes detected over Greenland. Simulations have shown that these winds do change characteristics from the start to the end of the last deglaciation, although only a single study has examined how these changes may have progressed in time. Due to the manner in which boundary conditions were updated in the simulation that single study was based on and due to revised understanding of those boundary conditions, it is difficult to assess whether the timing of the wind changes that occurred are actually representative of past changes. Therefore, this study diagnoses the changes undergone by the NAtl eddy-driven jet from the LGM to the preindustrial period in multiple transient deglacial experiments using boundary conditions that are updated every simulation year following the specifications of the PMIP4 (Ivanovic et al., 2016). These simulations are performed using a modified version of the Planet Simulator (PlaSim) version 16, an Earth System Model with a primitive equation atmosphere and simplified parametrizations (Fraedrich, 2012; Lunkeit et al., 2012). As such, these simulations can help elucidate whether atmospheric dynamical changes have the potential to play important roles in the abrupt climate changes of the last deglaciation and provide an important comparison against PMIP4 deglacial studies performed using more complex models."

**Section 3.2** *What is the point in this section? At the end of this section I don't know what I am supposed to have learned. It would help to have a clear finishing paragraph to summarise this section.*

Done. Revised text is "Thus, the NAtl jet exhibits changes in position and distribution that occur independently of each other in both PlaSim and TraCE-21ka simulations. Some of these changes coincide with historical climate changes. However, since these jet characteristics and the timings of their changes differ on the eastern and western sides of the NAtl jet, we attribute them separately in the next section. "

*At present it [Sec 3.2] is a selection of disparate facts. If you focus on the observations which support you summary, noting any discrepancies it may help add a narrative structure.*

We have revised this section substantially, adding many more connecting sentences to improve the flow of ideas. Since other reviewer comments led us to move any discussions to a Discussion and Conclusions section, we chose not to add extra discussion in this section as the reviewer suggested.

e.g. "As discussed in Section 3.1, the low-level, NAtl jet is situated further south and exhibits a narrower range of latitudinal variability in the FullyTrans runs at the start of the deglaciation compared to its end. "

"The TraCE-21ka experiment reproduces both types of jet changes detected in the PlaSim simulations, although the timings of jet shifts are not the same. "

*It would help if you were to link the discussion here [Sec 3.2] with the discussion comparing the jet in the east and west to the discussion of tilt - how do the changes at either end affect tilt?* **Tilt v east/west changes** *Reading the manuscript one gets a sense that the tilt of the jet and the changes in the jet in the east and west are two separate entities, which they evidently aren't. Historically, the focus has been on tilt, so it makes sense to at least look at this. However, I really like the description of how the east and west vary as this adds nuance to the very simplistic view of tilt. What's lacking the manuscript though is much of a bridge between the two. How can we interpret the previous discussions about tilt in the context of your results?*

We added explicit comparisons of jet tilt changes extracted from this study with those from previous studies. We also made more explicit interpretation of how western and eastern jet latitude changes correspond to tilt changes.

Revised text is "A shift of similar size is observed on both the western and eastern regions of the eddy-driven jet (Figure 7), so there is little change in mean tilt values between these two periods (Figure 8)."

"Ensemble statistics for the NAtl jet tilt in the FullyTrans runs in Figure 9 show oscillations between the preferred jet tilt of LGM and a more zonal configuration. These changes in jet tilt reflect the very different behaviours with time of the western (270°E to 300°E) and eastern (330°E to 360°E) sides of the NAtl eddy-driven jet, which are shown in Figures 10 and 11. The timing of these transitions match the first two (more gradual) shifts in the jet as a whole and are consistent with the historical timings of the OD and B-A. They also correspond to reductions in the jet tilt ." "There is a single, gradual, northward shift in the eastern region, occurring between 16 and 15ka BP. This change is a little later than the time of increasing jet tilt. " "Without the ice sheet barrier effect, the preferred tilt in the PDTopo experiment is much smaller than in FullyTrans and does not change over the deglaciation (Supplemental Figure S7), asthe western and eastern sides of the jet appear to shift together. " "All of the fully-transient deglacial simulations presented here show that the NAtl eddy-driven jet shifted northward from the Last Glacial Maximum to the preindustrial period, and its latitudinal variability increased. These characteristics match those derived from other studies (Li and Battisti, 2008; Lofverstrom et al., 2014; Merz et al., 2015). However, unlike those studies, neither the PlaSim simulations nor TraCE-21 show much change in jet tilt between these two periods. "

*I'm not sure "oscillation" is a useful term to describe periods like the Younger Dryas. Oscillation implies some set of physics that gives a distinct cycle, and it's not clear to me that any changes which occur over the last deglaciation can be described as cycles. A better term would be "variability",*

*which encapsulates the fact that there are changes without implying any cyclical physics. You even make this point on page 2 line 28. The presence or absence of cycles is not important for the interpretation in the paper so I'd suggest the less loaded term "variability"|.*

We acknowledge the reviewer's point. Revised text is Mid-latitudinal atmospheric dynamics may have played an important role in these climate variations ..." "Thus, we suggest that changes to the NAtl jet may play a critical role in abrupt glacial climate changes.""The last deglaciation encompassed a period of large-scale global warming of the Earth's surface climate, with regional patterns of millennial-timescale variability ..." "Signatures of these climate variations are present in the mid- to high-latitudes of both hemispheres...""Such variability is also present in proxy indicators...""  "Proposed hypotheses explaining the presence of such variability commonly centre around deep-ocean..." "For example, altered winter sea ice extent is sufficient to reproduce the amplitude of deglacial climate changes ..." "Thus, deglacial, winter wind conditions over the North Atlantic may have played important roles in the detected abrupt climate changes..." "None of the transient simulations presented here produce abrupt transitions between stadial and interstadial conditions at the historical time of the OD, B-A, or YD."

*p3 l5 - "During the deglaciation, changes to the variability of low-level winds can alter gyre transports (and to a lesser degree, wind position and strength)" Not sure I understand this, surely variability of the low level wind **is** changes in the wind position and strength?*

The sentence was rearranged to clarify our meaning. "Since low-level wind patterns over the Arctic, Greenland and NAtl help constrain winter sea ice extent in this region (Venegas and Mysak, 2000) and affect surface ocean heat transports there through the application of surface wind stresses (Lozier et al., 2010; Häkkinen et al., 2011; Li and Born, 2019), these winds may play an important role in setting the conditions required for abrupt deglacial changes or be involved in the abrupt transitions themselves. "

*p3 l27 - jet not yet.*

Done. Revised text is "However, the eddy-driven jet is ..."

*p5 l29 - how many ensemble members are there?*

Since this section is a description of the model rather than the experiments run for this study, it doesn't seem fitting to discuss how many ensemble members were generated here. That information is provided in the introduction to this section. "The experiments discussed in this paper consist of four transient simulations of the last deglaciation generated using a modified version of PlaSim and a suite of sensitivity studies (FixedOrbGHG, FixedGlac, PDTopo and DarkGlac)." Instead, we reworded the identified sentence to focus the reader's attention on the attributes of the model. "Herein we use a spectral resolution of T42 (approximately 2.8°x2.8°), which has been previously shown to be sufficiently high to resolve phenomena of interest to the eddy-driven jet (Barnes and Hartmann, 2011; Lofverstrom and Liakka, 2018) while enabling fast enough model run times to make multiple deglacial experiments feasible."

*p5 l30 - "Gaussian grid that is then used for diabatic calculations" not sure what diabatic calculations are.*

Diabatic calculations are those that involve the exchange of heat between fluid parcels and their

environment or phase changes. The text has been changed to clarify this. Revised text is "The dynamical atmospheric solutions are generated in spectral space, while the remaining calculations (e.g. phase changes, heat exchange with the land, sea ice or slab ocean, and any changes in those sub-components) occur in real space on a Gaussian grid with 64 latitude points and 128 longitude points."

*p8 l22 - You don't mention anything about model spin up? How is the model initialized?*

Done. Revised text is "All deglacial simulations are initialized from the same initial conditions, which are derived from year 2567 of an equilibrated LGM spin-up started from present-day."

*p9 l7 - I'm not sure what "peak zonal winds" means. Perhaps just say strongest winds?*

Done. Revised text is "For the LGM, northern midlatitude zonal winds from the PlaSim simulations are stronger than those from the CMIP5 multi-model ensemble. Also, the strongest winds of both the NAtl and the NPac jets are shifted further east toward the the eastern margins of their respective ocean basins."

*p9 l16 "the pattern of wind changes from the LGM to the past1000 are similar in both CMIP5 and PlaSim runs (including during JJA, not shown) even though the differences are stronger in the PlaSim transient simulations." Saying transient simulations at the end of this sentence makes it sound like there's an extra set of simulations - "the transient simulations" - as well as the normal "runs". This is not the case?*

There is only one set of FullyTrans deglacial simulations performed with PlaSim. Revised text is "In spite of these specific differences, the pattern of wind changes from the LGM to the past1000 are similar (but differ in magnitude) between the CMIP5 and PlaSim runs (including during JJA, not shown). "

*p10 l9 - I'm not sure how you get from "10 consecutive DJF periods" to the histogram on figure 6. 10 DJF periods surely gives 30 months, yet the frequencies on Fig 6. show 100s of months. Is this due to all the ensemble members?*

The reason the numbers of months presented in Figure 6 are much larger than expected based on 10 consecutive DJF periods is that the statistics in this Figure are based on 100 simulation years for all four ensemble members. Revised text is "Unlike Woollings et al. (2010), however, these latitudes are defined from monthly data (without low- pass filtering) over longitudes of 90°W to 0°W. Unless otherwise indicated, the monthly jet latitudes are aggregated over consecutive DJF periods to generate jet latitude frequencies." in Figure 6 caption: "Histograms of latitudes corresponding to peak NAtl zonal winds for all PlaSim ensemble members (left column) and the TraCE- 21ka data (right column) during indicated periods. Monthly jet latitude statistics are aggregated over 100 simulation years and four ensemble members for PlaSim and 1000 simulation years for the TraCE-21ka simulation."

*p12 l6 - "tilt .. shifts slightly higher" - better as tilt becomes steeper.      p12 l7 - It would be interesting to compare the spread in histograms between Trace21ka and PlaSim. You discuss spread for PlaSim on p11, why not Trace21ka too?*

Done. Revised text is "Differences between the PlaSim ensemble and the TraCE-21ka simulation primarily arise with respect to the jet tilt. The mean jet tilt increases by approximately 2°. ... However, either value of jet tilt change in the TraCE-21ka simulation is larger than that detected in the PlaSim simulations, and asymmetries in the increase in the range of jet tilt are opposite for these two datasets. Examining the eastern and western sides of the jet separately, the distributions in both regions change

more similarly from LGM to past1000 in TraCE-21ka than in the PlaSim simulations (see Supplemental Figure S2). ”

*p12 l14 - There needs to be a discussion of "abrupt". What constitutes abrupt: changes over what timescale? When looking at jet shifts in a coarse resolution model, a movement from on one grid box to the next come across as abrupt, tipping point like, but is actually just a smooth rapid change. To be abrupt suggests some set of non-linear feedbacks giving a larger response than the input would suggest. A linear response to a large change to me is not abrupt, just rapid. This is totally personal, but to avoid anyone misinterpreting what **you** mean by abrupt you need to define it.*

We agree and added the following paragraph to the Discussion and Conclusions section. “Assessing the abruptness of NAtl jet changes is problematic, because the abruptness of a phenomenon depends on its context. During the deglaciation, changes are generally considered abrupt if they occur within a couple of decades and are of sufficient amplitude to appear unusual compared to background climate variability. In this study, additional complication arises from the fact that the wind data is gridded; a latitudinal change in the position of the NAtl jet will involve a discrete step from one latitude grid to another. Thus, we consider a jet change to be abrupt (given the decadal resolution of our jet diagnostics) when the jet shifts its median latitudinal position from one grid cell to the next without an intermediate period when the jet splits its time roughly equally between the two. “

*p13 l5 - What exactly are you saying in this paragraph? I don't see what the point is.*

Since those abrupt climate changes were associated with changes to surface conditions over Greenland, based on $\delta^{18}O$ ratios in Greenland ice cores, we discuss whether we see evidence of such variability in Greenland temperatures in the PlaSim simulations.  The answer is that during the historical periods of the Oldest Dryas, Bolling-Allerod and Younger Dryas, we detect changes to the jets , but they do not yield large-amplitude, abrupt climate changes over Greenland.  Instead, we detect that type of variability over Greenland at other times in the simulations.  Is this sufficient to rule out these jet changes as important to the OD, B-A or YD?  This depends on whether PlaSim represents all of the (presently unknown) feedback processes important to these events effectively enough to capture any possible link between the jet changes and the rest of the climate system and the degree to which these historical events were stochastic.
Revised text is “Due to the absence of abrupt climate changes over Greenland, we conclude that the changes in the position, tilt and variability of the NAtl eddy-driven jet are not sufficient on their own to generate large-amplitude, abrupt climate changes in PlaSim.  It may be that feedbacks between the atmosphere, ocean, land ice and sea ice that are not captured in the simulations here are important in abrupt changes. For example, one very plausible process that is missing from PlaSim is the effect of winds on sea ice. Furthermore, this absence of simulated abrupt climate change does not rule out the possibility that the discerned atmospheric dynamical changes were important to historical abrupt climate changes through their controls on the background climate state. Thus, we characterize the atmospheric changes present in the accelerated transient simulations and leave further assessments of their implications for future work. “

*p14 - Transition seems to be used to describe two different things here. From line 8 there are the three events called transitions then on l10 transition is used to describe the way things change. It doesn't help that on l10 it says "A second type of transition" without ever being clear what the first type of transition is. This whole paragraph l2 onwards is really difficult to understand. I'm struggling to be more helpful and think of suggestions to improve it, but a reader is really going to struggle with this.*

The reviewer is correct in that we attempt to categorize two types of changes to the NAtl jet over the deglaciation (jet shifts and changes to the distribution of jet frequencies) and use the word "transition" to describe both of them. Revised text is "The two different types of deglacial changes (the latitudinal shift and the change in the shape of its distribution) occur separately over the deglaciation. In the first type of change, the median jet latitude shifts northward three times over the deglaciation in all ensemble members.""In the second type of jet change evident in Figure 9, the frequency of time that the NAtl eddy-driven jet spends at its median latitude decreases, and the range of jet latitudes increases. "

*p15 l4 - This seems to contradict p11 l5 which says that the jet tilt doesn't change much from LGM to past 1000.*

Both are true. Throughout the deglaciation, the jet tilt switches back and forth between its state at LGM and one grid cell more zonal configuration. Since the distribution of jet tilts broadens at the end of the deglaciation without introducing much overall tendency in the changes to this variable, there is little change in mean jet tilt between the start and the end of the simulation (LGM and the past1000). "A shift of similar size is observed on both the western and eastern regions of the eddy-driven jet (Figure 7), so there is little change in mean tilt values between these two periods (Figure 8)." "Ensemble statistics for the NAtl jet tilt in the FullyTrans runs in Figure 9 show oscillations between the preferred jet tilt of LGM and a more zonal configuration. These changes in jet tilt reflect the very different behaviours with time of the western (270°E to 300°E) and eastern (330°E to 360°E) sides of the NAtl eddy-driven jet, which are shown in Figures 10 and 11. Nevertheless, by the end of the deglaciation, the jet on both its western and eastern sides has shifted northward by a similar amount, leading to little net change in jet tilt. "

*p15 l13 - "The preferred latitude shifts northward twice within a single decade of simulation, at 19.3ka BP and 14.6ka BP. The timing of these transitions match the more gradual shifts in the jet as a whole and two occasions when the tilt is reduced. They are also consistent with the historical timing of the start of the OD and B-A." I do not understand this pair of sentences.*

Revised text is "The preferred latitude shifts northward twice at 19.3ka BP and 14.6ka BP, and each shift is completed within a decade of simulation. The timing of these transitions match the first two (more gradual) shifts in the jet as a whole and are consistent with the historical timings of the OD and B-A. "

*p16 l14 "This separation makes it much easier to identify what changes are occurring and attribute their causes than examinations of the mean jet position over the entire range or its tilt.." This is an awkward sentence.*

Revised text is "Since these jet characteristics and the timings of their changes differ on the eastern and western sides of the NAtl jet, we attribute them separately in the next section. "

*p17 l4 - this is an interesting point, any conjecture as to why orbit and GHG matter?*

Since the position of the jet over the eastern NAtl is determined based on where transient eddies (i.e. storm tracks) form and decay, moving the location of the polar front could affect the location of the jet in this region. Orbital and greenhouse gas forcings alter the background climate conditions in different characteristic ways. Orbital changes alter the distribution of heating around the globe, particularly on sub-annual timescales. Greenhouse gas changes are known to have a disproportionate effect in polar regions. Both of these processes could move the polar front.

Revised text is "In general, it appears that the characteristics of the jet on the eastern side of the North Atlantic are sensitive to changes in the background climate state, likely through changes to the positions of the polar front (and the growth of associated eddies) and sea ice margin."

*p18 l10 - "Yet, the jet does not always move to the latitude of the jet" Is an odd sentence.*

Revised text is "Yet, the jet does not always move to the latitude of the ice sheet margin."

*p19 l3 - "The consequences of this restriction are that the western end of the jet is more focussed relative to the eastern side, particularly when the NAIS extends well into the midlatitudes, and that the northern range of the western side of the jet increases much more over the deglaciation than its southern range"*

*Perhaps rewrite as: "There are two consequences of this restriction. First the western end of the jet is more focussed relative to the eastern side, particularly when the NAIS extends well into the midlatitudes. Second, the northern range of the western side of the jet increases much more over the deglaciation than its southern range"*

"The western side of the jet is more focussed to a single latitude than the eastern side, particularly when the NAIS extends well into the midlatitudes. This is because the barrier is being applied to the winds in the western region of the jet, but not the eastern region." "The distribution of the western side of the jet during most of the deglaciation is strongly skewed with the jet spending the bulk of its time at the northern boundary of its range. In contrast, the jet distribution in the eastern region is more symmetrical. As long as the ice sheet continues to impinge on where the jet would preferentially be located in the absence of the ice sheet, the wind shear along the northern edge of the jet remains very strong. Thus, eddies tend to break along this boundary and accelerate the flow there. This keeps the jet preferentially in its northernmost position. Since there is no such constraint on the jet in the eastern region, it is free to vary equally in both directions around its mean position. "

*p19 l10 - "The preferred tilt in the PDTopo experiment is near zero". The tilt doesn't change much in this simulation and stays at its past1000 value, which you show and argue does have a tilt, 5 degrees in Fig. 8. I agree that the **change** is near zero, but not that there is no tilt.*

Revised text is "Without the ice sheet barrier effect, the preferred tilt in the PDTopo experiment is much smaller than in FullyTrans and does not change over the deglaciation (Supplemental Figure S7), as the western and eastern sides of the jet appear to shift together."

*p19 l15 - "In contrast, the shift in latitudes occurs earlier on the eastern side of the jet, and no further change to the preferred range of jet latitudes occurs following this." This sentence exemplifies why, I think, this paper is so confusing. What is important in this sentence is "the eastern side of the jet": this is what is being compared to the preceding sentence. Yet the way that this sentence is structured puts this midway through the sentence, slightly buried. Thus it takes very careful reading to parse the sentence. If you wrote it as: "In contrast, on the eastern side of the jet the shift in latitudes occurs earlier..." it would be much more obvious what's going. It may be that this reviewer is a bit stupid, but with the long sentences that you use any help that a reader can get would be good. I'd have a look through the paper for more instances of this inverting of sentences. Given the quality of the English in this review you may, however, choose to ignore the stylistic recommendations of this reviewer.*

Thank-you to the reviewer for pointing out sentences that (s)he finds difficult to parse. We tried to shorten the sentences and make the language clearer as we edited the manuscript. For the example provided, the revised text is "In the eastern region, the shift in latitudes occurs earlier than in

FullyTrans, but the jet never reaches as northern a position as it does in FullyTrans."

*p22 l1 - This section of the conclusions relates to the weakest part of the main text, much of what is in this part of the conclusions was not clear from the preceding sections.*

We have substantially revised the results section and hope that it is much clearer now.

*p22 l10 colon inappropriate here.*

Revised text is "These shifts are each accomplished within a decade of simulation (century of forcing), and their timings are consistent between the accelerated simulations, an unaccelerated run, and an accelerated transient simulation starting from a warmer initial state. "

*p22 l21 - it would be clearer to say "through two phenomena: first, .... second ...." the two descriptions are so long you need to make it clear where one stops and the other starts.*

This sentence was removed in more recent edits of the manuscript.

*p23 l7 - "Conversely, the sensitivity of the jet position on the eastern side of the North Atlantic to the background climate state implies that it would be difficult to estimate historical changes to the jet in this region from model simulations, since estimates would vary between models and between simulations with different boundary conditions." Need to explain this more. Surely, because the orbit and GHG are better constrained than ice sheets, this response will be better simulated?*

While the reviewer is correct in that our knowledge of deglacial changes in orbital forcing and greenhouse gases is less uncertain than our knowledge of past ice sheet configuration, his/her conclusion is mistaken. Climate responses to orbital and greenhouse gas forcings are complex and involve a multitude of feedbacks between different climate components and depend on subgrid parametrizations for phenomena like clouds. The combined effect of all of these processes makes predicting climate responses challenging and model-dependent. In contrast, ice sheet margins acting as a physical barrier to the winds is a very simple process that does not depend as strongly on the most uncertain components of climate models.
Revised text is "As such, it would be difficult to estimate historical changes to the jet in this region from model simulations, since the pattern of thermal responses to changes in boundary conditions is sensitive to model parametrizations for processes like cloud physics, and feedbacks between the atmosphere, ocean and sea ice. Thus, the effect of changing boundary conditions likely varies between models and even between simulations using the same model but different initial boundary condition states."

*p23 l12 - I disagree. The assumption here is that the surface temperature response is linear with respect to the jet latitude: as jet latitude increases so will temperature in direct proportion. But, if there are non-linear feedbacks it could be that when the jet reaches a certain latitude abrupt changes in temperature are possible. For example, imagine the jet is well south of the sea ice margin but gradually moves north due to a slow retreat of the ice sheet. At some point the jet will be over the sea ice margin and a different set of feedbacks become possible. Thus you can get abrupt changes in temperature from a smoothly varying jet/ice sheet.*

The reviewer raises a good point. Revised text is "It is unlikely that relevant changes in the ice sheet margin occur on timescales of decades, so it appears that changes to the upstream end of the North Atlantic jet are more likely to play an enabling role than a causal role for abrupt climate changes. Yet, we can not entirely rule out the possibility that gradual jet changes can trigger abrupt climate changes.

The non-linearity of the coupled climate system implies that gradual changes do not necessarily lead to gradual responses, particularly if there are thresholds (e.g. sea ice edge) beyond which feedbacks change. "

*Figure 1(a) - There's an enormous remnant ice sheet over NAm: 1.7km at 0ka. What is this?*

Figure 1 plots the peak elevation of ice sheet-covered areas in North America and Fennoscandia. There is no 1.5km thick ice sheet over North America during the Holocene, but there are regions that have an elevation in excess of 1.5km that are covered by ice: glaciers in the Rockies, for example.

Revised caption is "a) Peak elevation in ice sheet-covered areas (bedrock elevation plus ice sheet thickness) and b) ice sheet area for North America (NAIS) and Eurasia (FIS). " Revised text is "In Figure 1a, there appears to be an elevated remnant of the NAIS that continues to present day, which corresponds to small glaciers located in the Rocky Mountains."

*Figure 5 - Its hard to judge in this figure how the amplitude of the jet changes. If you highlight one isotach, 20m/s, in both the contours and colours it would make it simpler to see how the structure of the jet differs.*

From our perspective, highlighting a colour and adding a highlighted line isotach for LGM would likely both reduce and add confusion.

---

## Author Comment (AC4) · 3 May 2019

[revised manuscript text omitted]
 CO2, nitrous oxide (N2O) and methane (CH4). Effective CO2 concentrations were defined with respect to reference CO2 concentrations at year 22.3ka BP using the equations in Ramaswamy et al. (2001). This reference year was chosen, because both N2O and CH4 values were at a relative minimum at that time, and this year precedes the period of interest for our study. Data for CO2, N2O
- and  $CH_4$  concentration changes over the deglaciation are consistent with the prescriptions of the PMIP4 Deglacial experiment except for  $CO_2$ , which uses an older dataset (see Ivanovic et al. (2016) and data sources Luthi et al. (2008) Meinshausen et al. (2017), and Loulergue et al. (2008)). Effective  $CO_2$  concentration values are updated every simulation year, and are linearly interpolated in time between available data points as needed.

**2.2.5 Model Evaluation**

15 We compare the climate conditions during the first and last century of the fully-transient PlaSim simulations (corresponding to forcing years 21-20ka BP and 1ka BP to 1950CE, respectively, due to acceleration) to the results of LGM and past1000 experiments in the Climate Modelling Intercomparison Project (CMIP) 5. Only CMIP5 simulations that include experiments using the same model configuration for both LGM and past1000 are included here, as tabulated in Supplementary Table S1.

In Figures 2 and 3, near-surface temperatures (T2m) and sea ice concentrations are plotted for ensemble averages of the first and last centuries of the fully-transient PlaSim simulations. Hatching in Figure 2 indicates where the PlaSim ensemble average

- 20 and last centuries of the fully-transient PlaSim simulations. Hatching in Figure 2 indicates where the PlaSim ensemble average T2m lie within the range spanned by the CMIP5 multi-model ensemble members (interpolated to the same resolution). Red and gold lines in Figure 3 identify CMIP5 multi-model ensemble maximum and minimum sea ice extents during the same season, respectively. The surface climate is colder and sea ice more extensive during LGM in the PlaSim simulations than the CMIP5 multi-model ensemble members for both DJF and JJA. This is particularly true in the northern high latitudes, where
- 25 temperatures over the Arctic Ocean lie below all CMIP5 ensemble members and sea ice is anomalously extensive in the NPac and on the eastern side of the NAtl. In contrast, Antarctic temperatures lie within the range of CMIP5 models, and Southern Ocean sea ice is close to the CMIP5 maximum.

Most of these disagreements are resolved by the past1000, when PlaSim-predicted temperatures and sea ice concentrations lie within the CMIP5 range in most regions. However, temperatures remain anomalously cold over the Arctic during DJF

30 and over oceans in the midlatitudes in both hemispheres and seasons, and sea ice extent exceeds the CMIP5 maximum in the Southern Ocean during JJA. These differences are discussed in more detail in Appendix section A1.

---

## Author Response (AR2)

*Review 2 of:*
*Andres and Tarasov: Towards understanding potential atmospheric contributions to abrupt climate change: Characterizing changes to the North Atlantic eddy-driven jet over the last deglaciation, Climate of the Past Discussion*

*Summary:*
*The authors have made substantial improvements of the text from the first round of review, and I am glad to see that many of the suggestions and comments from the referees have been incorporated. I recommend that the manuscript is accepted, subject to a few minor edits.*

*My impression of this manuscript is generally positive, however I can't help but come away with a feeling that we haven't really learned a whole lot from this study that we didn't already know before. I hope I am not too unfair. The main takeaway seems to be that the LGM was a different climate state than modern, but the transition between these two states appears to be both model and forcing dependent. As you have shown, Plasim and Trace agree on some aspects of this transition, but disagree on other aspects, and it is hard to say which simulation (if any) is more truthful. I certainly don't think that Trace reflects reality despite using a comprehensive fully-coupled circulation model (the simulation has several known problems, many of which are pointed out in the text), but I am not convinced that the Plasim results are necessarily more accurate because of the simplified model code, general approach, and apparent issues reproducing the CMIP5 mean climate at the LGM. Although this study certainly takes a stride in the right direction, more studies of this kind are necessary, and we can only hope that both Palmod and iTrace will shed more light on this important issue in the not too distant future.*

We acknowledge Reviewer 1's assertion that there are aspects of deglacial jet changes that are similar and others that differ between the PlaSim simulations and TraCE-21ka. We also acknowledge that both sets of simulations have issues associated with the models, boundary conditions or methodology that may affect the applicability of aspects of their predictions to historical jet changes. However, it is not the goal of this study to pit one set of simulations against the other to gauge which one is "right." Rather, it is to compare the predictions from these two sets of runs to see whether they are saying something consistent about how the jet responds to the types of boundary condition changes occurring during the last deglaciation, and whether those changes had the potential to contribute to the abrupt climate changes that occurred during that time. From that perspective, we disagree with the reviewer's assessment of the main takeaway of this study. We would instead define the main takeaway to be that the large differences in atmospheric states between the LGM and present day were not arrived at gradually throughout the deglaciation but through a succession of changes to the position of the jet followed by a gradual increase in the range of the jet.

More to the reviewers general point, the text already indicates how this study contributes importantly to the field in multiple ways (textual examples are listed with each point):

1. We show that in a set of simulations where boundary conditions vary smoothly in time (instead of via large, discrete adjustments as in TraCE-21ka), the jet nonetheless does not transition smoothly from the LGM to the Holocene. Particularly over eastern North America, it undergoes a sequence of discrete shifts in latitude. Since the only other study that analysed jet changes in a transient deglacial experiment (Lofverstrom and Lora, 2017) analysed jet changes in TraCE-21ka, this is new information.
   "Simulations have shown that these winds do change characteristics from the start to the end of the last deglaciation, although only a single study has examined how these changes may have progressed in time. Due to the manner in which boundary conditions were updated in the simulation that single study was based on and due to revised understanding of those boundary

conditions, it is difficult to assess whether the timing of the wind changes that occurred are actually representative of past changes. Therefore, this study diagnoses the changes undergone by the NAtl eddy-driven jet from the LGM to the preindustrial period in multiple transient deglacial experiments using boundary conditions that are updated every simulation year following the specifications of the PMIP4 (Ivanovic et al., 2016)."

"The deglacial NAtl jet changes arise in a set of PlaSim simulations where boundary conditions vary smoothly in time. Nevertheless, the jet changes via three well-separated northward shifts in latitudinal position (at 19ka BP, 14ka BP and 11.0ka BP)... "

2. We show that in both PlaSim simulations and TraCE-21ka changes to the most common jet latitudinal position and the distribution of jet latitudinal positions do not necessarily occur at the same time or at the same rate.

"...the jet changes via three well-separated northward shifts in latitudinal position (at 19ka BP, 14ka BP and 11.0ka BP) followed by a gradual transition away from a latitudinally-constrained regime of variability that is complete by 8ka BP in all simulations. The TraCE-21ka experiment exhibits two shifts in the most common position of the jet (at 13.87ka BP and 13ka BP) contemporaneous with the start of a gradual increase in latitudinal variability. "

3. We show that in both PlaSim simulations and TraCE-21ka the jet over eastern North America remains south of the elevated ice sheet margin at all times, which explains its constrained latitudinal variability during LGM. Previous studies have shown that the jet over the western North Atlantic is affected by the presence of North American ice sheets (e.g. Kageyama and Valdes, 2000), but they do not identify which aspect of the ice sheets are most important to this effect (peak elevation, dome position, area, etc.).

"While previous work shows that the western side of the jet is more strongly affected by ice sheet orography and the eastern side by stationary and/or transient eddies (Kageyama and Valdes, 2000; Lofvestrom et al., 2016), we identify a more specific relationship between the North American ice sheet complex and the western side of the jet. In both the PlaSim and TraCE-21ka experiments, low-level winds over eastern North America are highly focussed to the latitude of the ice sheet margin and vary only south of this margin, even though the ice sheet surface lies below the top of our low-level wind range. When this margin retreats, the winds tend to shift northward with it, but not before. It is the orography of the south-eastern margin of the North American Ice Sheet complex that appears to play the most important role in this effect ..."

4. We analyse the distribution of jet changes rather than just changes to the mean and standard deviation, which allows us to separate the jet shifts from the jet distribution changes more effectively.

"Additionally, instead of characterizing the jet via the Gaussian statistics of its mean and standard deviation, we directly examine the jet distribution. Through frequency plots, we present changes in time of the jet distribution itself by showing the percentage of winter months that the jet spends at each latitude for every decade of simulation."

5. We have pointed out the usefulness of examining the eddy-driven jet over its western and eastern regions separately rather than averaging over both regions and using the more obtuse tilt metric, since their mechanisms and timing of changes are different.

"Interpreting the detected jet changes is complicated by the fact that the NAtl jet calculated over longitudes 270°E to 360°E mixes characteristics of the upstream and downstream regions,. The evolution of the jet in these two regions differs in timing, abruptness and dependence on background model conditions. "

6. Finally, this study is carried out an ensemble of 4 model runs along with various sensitivity runs compared to previous studies that have either relied on snapshots or the single transient TraCE-21ka simulation.

"The novelty of this present study compared to those previously mentioned is that it analyses transient changes in the jet over the entire deglaciation in multiple experiments."

*Response to rebuttal:*
*In the first round of review I raised a concern about the claim that the subtropical and eddy driven jets entered a merged state in the North Atlantic at the LGM. The authors doubled down on this claim in their rebuttal, and pointed out a number of examples to support this assertion (it should be said that they removed the references in the manuscript text to make it clear that this is their own interpretation). However, I maintain that the sentence on page 4, line 19 is problematic and should be removed from the manuscript because: (1) it is demonstrably wrong (see below); (2) it gives a false impression that may be perpetuated in other studies since there is a most unfortunate tendency nowadays of mechanically quoting earlier studies without looking up the original source.*

*First of all, the examples from the lower troposphere (everything below 500 hPa) can be dismissed right away, since the subtropical jet is by definition an upper tropospheric entity (it is primarily due to Coriolis acceleration in the poleward branch of the Hadley Circulation) and therefore has no signature (actually often negative u-wind) in the lower troposphere. Second, the cross sections pointed out in the rebuttal clearly show a wind maximum in the subtropics. For example Fig. 4 in Merz et al. (2015) shows a very distinct secondary wind maximum in the subtropics (as shown by the shading). The subtropical jet is perhaps weaker than in the PI simulation, but it is still there and is clearly separated from the eddy driven jet in midlatitudes. The other studies cited in the rebuttal show the same thing, and, incidentally, also Fig. 5 in this manuscript, even though the wind profile here looks a bit suspicious (probably explained by the small number of model levels and perhaps also the simplified modeling approach).*

Firstly, the reviewer and we appear to be working from two different understandings of a merged jet: our impression is that the reviewer sees it as a binary state, either merged or not, whereas on the basis of papers like Barnes and Hartmann (2011), we perceive jet merger as a matter of degree. A more merged jet is situated further south and is less latitudinally variable than a NAtl eddy-driven jet such as we have today. We concede that given the reviewer's definition, there is not strong evidence for jet merger in the papers we cited, and based on the LGM wind profile in Figure 5 of this paper, we agree that there is no evidence of jet merger at LGM in the PlaSim simulations. One option to reconcile these differences is to employ a jet merger statistic like the zonal jet index (Harnick et al, 2014). However, we do not think that is needed in this case, both because the evidence for jet merger is so weak in the PlaSim simulations at LGM and because it is not important to the main conclusions of the paper.

Since we do not want any statements about whether jet is merged or not to distract from the focus of the paper, we have chosen to remove all instances where it is mentioned at all.

"under a strong subtropical jet as in the North Pacific (NPac) today, the eddy-driven jet tends to lie along the poleward flank of the subtropical jet in a "merged" state. " => "under a strong subtropical jet as in the North Pacific (NPac) today, the eddy-driven jet tends to lie along the poleward 15 flank of the subtropical jet. Its variability tends to be dominated by fluctuations in strength rather than latitudinal position. "

"These changes are consistent with what would be expected if the NAtl eddy-driven jet entered a more "merged" state at LGM, although not 20 all of these studies show evidence of this. " => "The reasons for these jet characteristics thus remain unexplained. "

*Line-by-line comments:*

*Page 1, line 5:*
*Should be "intermediate complexity model" instead of "intermediate resolution model". T42 is a pretty low resolution by modern standards, but it is much more important to point up that Plasim is very simple compared to the majority of the PMIP models.*
The reviewer only points out the first part of how we describe PlaSim, "an intermediate-resolution earth system model **with simplified physical parametrizations**." We are not suggesting that it is comparable to present generation PMIP models for modelling paleo climate. But it is not a priori clear whether the higher complexity but lower resolution TRACE simulation (using the clearly non-EMIC CCSM3) provides a more accurate representation of atmospheric circulation during the deglaciation. Furthermore, as explained in our first round of revisions, we intentionally avoid describing PlaSim as an EMIC (Earth system Model of Intermediate Complexity), because that term has come to encompass such a wide range of models that it is not particularly informative. Unlike current models that are self-described as EMICs, PLASIM solves the primitive equations for atmospheric dynamics and at T42 also operates at twice ore more higher spatial resolution than any self-described EMIC. To be even more accurate and precise, while reducing the length of too long a sentence, we now state (the details of model approximations do not belong in the abstract) :

*"using the PlaSim earth system model (run at T42 resolution) and the TraCE-21ka (T31) simulation."*

*Page 3, line 18:*
*More spatially localized?*
Done.

*Page 4, line 19:*
*See comment above*
Response above.

*Page 5, line 10:*
*I would say "simplified physics parameterizations" to be more precise*
Done.

*Page 6, line 1:*
*There is no such thing as wet primitive equations. The dynamical core of Plasim is identical to PUMA, which is a traditional primitive equation model (dry dynamics). Plasim also includes a simplified physics parameterization with equations describing water dynamics.*
Revised text "whose dynamical core is based on the primitive equations."

*Page 12, line 12:*
*Fig. 5 shows the whole troposphere so low level must be a typo*
Revised text "according to wind profiles in Figure 5 and histograms for lower-level jet latitudes in Figure 6."

*Page 13, line 6:*
*Sentence starting "The wave..." is note clear. Equatorward propagating waves tend to break anticyclonically, which yields a poleward momentum flux convergence and thus a meridionally shift in the jet structure. However, since you don't actually show any dynamics related to Rossby wave breaking, I would recommend staying away from this topic since it adds an unnecessary level of complexity that is: (1) speculative at best; and (2) only tangentially related to the topics discussed here.*

Comments regarding the mechanisms whereby the subtropical jet restricts the position of the eddy-driven jet have been removed.  Revised text "This asymmetry is consistent with the subtropical jet providing a dynamical limit on the southernmost position of the eddy-driven jet (Barnes and Hartmann, 2011).  Such asymmetry is apparent on both sides of the jet..."

*Page 16, line 22:*
*What do you mean by "preferred latitude"? Explain*
There is a difficulty in wording here.  Since the jet distributions are not all Gaussian, the mean jet latitude does not always correspond to the most commonly-occupied or "preferred" jet latitude.  The jet distributions start out very peaked, but the peakedness of the distribution changes as well as the latitudinal position of the jet range.  Thus, we diagnose jet shifts via changes to the position of the peak jet latitude, and the change in distribution primarily via the shift away from a single peak latitude.  Since "preferred" did not communicate these ideas clearly, we have replace that word as indicated by the changes below.
"In Figure 9, peak winds occur at the same latitude for more than two-thirds of the months from 21ka BP to approximately 19ka BP, whereas the jet occupies no latitude for more than 40% of the time from 8ka BP onward."
"Ensemble statistics for the NAtl jet tilt in the FullyTrans runs in Figure 9 show oscillations between the most common jet tilt of LGM and a more zonal configuration."
"The most commonly-occupied latitude shifts northward twice in this region,..."
"The transition away from a single, commonly-occupied jet latitude over eastern NAmer occurs more abruptly than for the jet as a whole at 10.8ka BP."
"...the eastern side of the jet over the eastern NAtl is less focussed than the west, with the jet occupying its most common latitude 50 to 70% of the time."
"The eastern side of the jet moves away from any single, commonly-occupied latitude after 11.4ka BP,..."
"Simulations with fixed LGM ice sheets… reproduce neither the deglacial changes to the most commonly-occupied jet latitude nor the bulk of changes to its variability..."
"In the east, the most common position of the NAtl jet does not change under fixed LGM ice sheets,..."
"The most commonly-occupied latitude of the western side of the jet shifts northward..."
"Without the ice sheet barrier effect, the most common tilt in the PDTopo experiments is ..."
"...and the jet shifts away from a single, most commonly-occupied jet location later than before..."
"...appear to be key for the transition of the jet away from a narrow, peaked latitudinal distribution to a broader, less-peaked distribution, ..."
"The TraCE-21ka experiment exhibits two shifts in the most common position of the jet..."
"...the transition to a more distributed jet without a single, most commonly-occupied latitude occurs much earlier over the eastern NAtl..."
"Thus, allowing the ocean to adjust to the changing climate conditions changes the timing of the jet shifts over the eastern NAtl and the timing of the transition away from a single, most commonly-occupied jet latitude."

*Page 16, line 32:*
*What? Why do you get a split jet long after, but not during the LGM? This peculiarity is opposite to what other studies have found and should be explained.*
The reviewer appears to be concluding that because we first detect the jet in the northern branch of a split jet during the middle of the deglaciation, that means the split jet was not present earlier.  In Figure 4b, there exists a more northern branch of winds at LGM whose winds are much weaker than the winds in the more southern branch.  Our jet detection algorithm only identifies the jet in the region of this northern branch during months when the winds in the northern branch are faster than in the southern

branch. This situation only arises mid-way through the deglaciation, when the mean winds in the southern branch have slowed down enough to allow this possibility to occur.

Revised text. "As in Woollings et al (2010), the algorithm we use to detect the latitude of the eddy-driven jet for the remainder of this analysis finds the location of maximum..."
"Note that the development of an isolated northern branch of mean jet latitudes sometimes between 15 and 14ka BP in Figure 9 does not indicate that winds passing south of the ice sheet are undergoing large latitudinal shifts. Rather, it is an artifact due to the strongest zonal winds being identified in the northern branch of a split jet stream over North America. Although the split jet stream is present from LGM (see Figure 4b), the northern wind branch is weaker than the southern branch until this time, so the jet detection algorithm does not identify the jet there."

*Page 21, line 3:*
*The wording here is clunky and hard to parse (I had to read this sentence several times before understanding your point). Simplify and say that you use modern topography with glacial surface albedo.*
Done. Revised text "The PDTopo experiment isolates the thermal forcing associated with ice sheets' relatively high albedo from the orographic forcing due to their elevation by imposing present-day land topography in combination with time-varying glacial surface albedo."

*Page 23, line 21:*
*Point here is not clear. The abrupt circulation changes in Trace are directly linked to specific (i.e., key) changes in the planetary boundary conditions --- as shown here, and also discussed in earlier studies. However, since you use a different boundary condition, it is only natural to see circulation changes at different times than in Trace.*
It appears that our intent in this sentence is being misunderstood. We're not discussing whether PlaSim and TraCE-21ka show changes at the same time, but rather whether the jet changes we detect in the PlaSim simulations led to large-scale abrupt climate changes in those same simulations, similar to those that are inferred to have occurred.
Revised text "Since none of the NAtl jet changes in the PlaSim simulations lead to large-scale, abrupt climate changes in those simulations, our results provide no support for a direct, causal role for jet changes in triggering historical abrupt climate changes."

*Page 23, line 29:*
*Well yes, although finite grid spacing is an inherent limitation of all models.*
Revised text "In assessing the abruptness of jet latitudinal changes, we must also address the fact that the wind data is gridded; a latitudinal change in the position of the NAtl jet will always involve a discrete step from one latitude band to another."

*This argument is also halting because, as you pointed out earlier in the manuscript, the jet can transition smoothly between two distinctly different states, even on a comparatively coarse grid. If the jet maximum changes from being in one grid cell 100% of the time, to a different grid cell 100% of the time, but occupies both grid cells (or all cells in between if they are spatially separated) with some distribution (say 50% in each) for some period of time, we would perceive it as a smooth transition. On the other hand, if it changes from one grid cell to the next in a discrete jump, most people would perceive it as an abrupt transition. I don't think this is particularly controversial. Also, what constitutes a smooth or abrupt change is obviously context sensitive. A change over a few decades or a century can be viewed as abrupt in a 21,000-year simulation, but would probably be viewed as a more gradual change in a shorter simulation.*

It appears the reviewer agrees with our criteria for abrupt changes in this context. This is precisely what we argue in Section 3.2. However, to reiterate this point, we added the following text. "However, as discussed in Section 3.2, we can distinguish between gradual latitudinal changes where the jet evenly splits its time between two adjacent latitude bands while it transitions from one to the other and more abrupt changes where there is no such mixed state."

*Page 25, line 2:*
*Obviously, it is meaningless to talk about wind inside the ice sheet... I am glad to see that you didn't extrapolate the Plasim and Trace data where the surface topography intersects the pressure level of interest.*
The reviewer appears to be suggesting that the result that low-level winds are varying only south of the ice sheet margin is trivial, because otherwise the winds would have to be travelling inside the ice sheet. However, from Figure 4b, it is clear that there are actually very few locations even at LGM where the ice sheet surface exceeds the top of our range for low-level winds, 700hPa, and none in the south-east corner of the North American Ice Sheet complex. To make this point clear, we have modified the text. "In both the PlaSim and TraCE-21ka experiments, low-level winds over eastern North America are highly focussed to the latitude of the ice sheet margin and vary only south of this margin, even though the ice sheet surface lies below the top of our low-level wind range."

*Page 25, line 6:*
*That is a reasonable hypothesis. However, I am not sure that the lower 1000 m are as important as the authors claim. Ice sheets generally have steep margins, which is something that climate models cannot resolve properly because of the finite grid spacing. I therefore think that this argument should be downplayed a little bit because it is contingent upon grid resolution, method used for discretizing the equations on the grid (horizontal and vertical), how sub-gridscale topography is incorporated in surface drag parameterizations, etc.*
In order to simplify the argument we are making about the elevation threshold, we repeated the scatter plots with pressure thresholds and with the ice sheet margins defined by the ice sheet mask. We find that the position of the ice sheet masks provide sufficient constraint on the position of the jet, so we replaced the plots in Figure 13 with the NAIS position determined in this way. We also removed all mentions in the text of a particular elevation threshold value. However, we think it is important to emphasize that the ice sheet mask does not provide a similar constraint on the jet position when the margin is not elevated.

"Our results suggest that the position of ice of above a critical threshold (in our case 725 m) proximal to the south-eastern margin of the North American ice complex strongly constrains the deglacial position of the jet over eastern North America and the western North Atlantic as well as its variability." ==> "Our results suggest that the presence of an elevated ice sheet margin in the south-eastern sector of the North American ice complex strongly constrains the deglacial position of the jet over eastern North America and the western North Atlantic as well as its variability. "

"The western side of the jet either reaches or lies south of the 725m contour of the south-eastern NAIS margin at all times in 5 the FullyTrans simulations and south of the 1000m contour in the TraCE-21ka experiment. " ==> "The western side of the jet either reaches or lies south of the south-eastern NAIS margin at all times in the FullyTrans 20 simulations and in the TraCE-21ka experiment. "

"This number was arrived at empirically and represents the lowest tested threshold that did not have instances of the western side of the NAtl jet exceeding its location. " This sentence was removed.

"Overall, the results presented here indicate that changes to the western side of the jet appear to be predominantly controlled by changes in the position ice above a critical threshold (725 m in our case) proximal to the south-eastern NAIS margin, ..." ==> "Overall, the results presented here indicate that changes to the western side of the jet appear to be predominantly controlled by changes in the position of the south-eastern NAIS margin, ..."

"If this hypothesis is correct and 10 this mechanism is as important in reality as it is in models, then the uncertainty in our understanding of the historical position of the jet over eastern NAmer during this period is dominated by the uncertainty in our knowledge of the southernmost position of the regional 1000m (or so) high ice. " ==> "If this hypothesis is correct and this mechanism is as important in reality as it is in these two models, then our knowledge of the southernmost position of regional ice provides a northern bound on the historical position of the jet. "

*Page 25, line 15:*
*These are probably not mutually exclusive*
Revised text "Whether this control is exerted via changes to the stationary waves (..), transient eddies (…), surface thermal gradients from sea ice and sea surface temperatures (…), some combination of these or some other mechanism altogether is not entirely apparent from this study."

*Page 25, line 2:*
*Possibly, but the fact remains that there appears to be a key change in the boundary condition --- both in Plasim and Trace --- that yields a disproportionally large change in the circulation.*
Revised text "When this margin retreats, the winds tend to shift northward with it, but not before."

*Page 26, line 4:*
*This information is important and should be moved up to the description of the simulations and how you analyze the results.*
The following sentence was added to the start of Section 3.1, where the jet metrics are defined.
"Where ice sheet elevation exceeds 925hPa, we interpolate the winds to these levels to not bias the resulting average.  This choice does not alter the conclusions of our study substantially and is discussed in further detail in Section 4."

*Page 26, line 9:*
*Perhaps an argument for using middle to upper tropospheric winds instead?*
Yes, analysing upper tropospheric winds would prevent any issues with the elevated ice sheet surface impinging on the wind's pressure levels.  However, upper tropospheric winds are less helpful for discerning wind stresses on sea ice and ocean surfaces.

*Page 26, line 25:*
*Trace was run with CCSM3, which includes both a dynamic ocean and sea-ice model…*
True.  This statement was written with the sensitivity studies we performed using PlaSim in mind, but that is not clear from the wording.  Revised text. "Better understanding of possible climate system sensitivities to NAtl jet changes will require additional sensitivity experiments that include dynamic sea ice."

*The manuscript is a great improvement on the previous version. There is far more detail and the reader is not left confused as to what's going on. Well done!*

*Below are some typos/odd uses of English that should be corrected. In general the manuscript is fine, but there are some places where a reader has to stop, re-read the sentence, stop again and try to work out what you mean. This is not because the concept is complicated but because it's not clear what the clauses in the sentence refers to. This does the paper a great disservice at is full of useful information.*

*p4 l20 irregardless not a word.*

Irregardless is recognized as a (controversial) word in the Oxford English Dictionary, but to avoid future confusion for other readers, the wording has been changed. "Regardless of the categorization of the jet as merged or not, changes to the path and variability of the jet ..."

*p5 l18. Need to introduce ensemble before this sentence. how many members? I think that I questioned this in the first version of the paper. It is still not clear how many ensemble members there are.*

Two sentences prior to the one mentioned by the reviewer, we state, "The experiments discussed in this paper consist of four transient simulations of the last deglaciation..."  In order to clarify that these four simulations make up the main ensemble, the revised wording is "The experiments discussed in this paper consist of an ensemble of four transient simulations of the last deglaciation..."

*p6 l4 therefore not thereby.*

This request appears to be stylistic rather than substantive.  We prefer "thereby", since its definition ("by that means") more precisely fits the context of the sentence.  By employing a simpler radiative transfer scheme than is used in current state-of-the-art GCMs, the code is made faster.

*p17 l10 clarify that the transition you are discussing at the beginning of the sentence is in the Western Atlantic.*

We discuss the characteristics of the western side of the jet over eastern North America in this paragraph and those of the eastern side of the jet over the eastern North Atlantic in the following paragraph.  Since this was not clear, we added some reminders.  Revised text "The preferred jet latitude shifts northward twice in this region at 19.3ka BP..."  "...shifts in the basin-averaged jet and are consistent..."  "...from a preferred jet latitude over eastern Namer occurs more abruptly..."

*p17 l7 twice is redundant, sounds as though jet moves 2 times at 19.3ka and once at 14.6ka*

We added another comma to clarify that the years correspond to the two shifts mentioned.  "The preferred latitude shifts northward twice in this region, at 19.3ka BP and 14.6ka BP, and each shift is completed..."

*p17 l15 remind the reader what the two types of jet change are.*

Done.  Revised text is "The TraCE-21ka experiment exhibits both latitudinal jet shifts and a transition away from a single, most commonly-occupied jet latitude, atlhough the timings of these jet changes are not the same as in the PlaSim simulations."

*p21 l5 likely no need to cite this here, but a paper wasn't recently published looking at this question: Roberts et al. 2019 The Mechanisms that Determine the Response of the Northern Hemisphere's Stationary Waves to North American Ice Sheets https://doi.org/10.1175/JCLI-D-18-0586.1*

Thank-you for drawing our attention to this paper.  Since this paper focusses on stationary wave changes and not changes to the jet, we chose not to cite it here.

*p22 l10 - barrier effect. Need to explain what you mean by "barrier effect". Also I don't understand*

*what the sentence "more likely that the barrier effect depends on the pressure level at which it is applied". This sounds like the "barrier effect" is some external prescribed forcing, which I don't think is what you mean.*

The reviewer is correct that the mechanism by which the prescribed ice sheets affect the winds over eastern North America is not being specified externally. It involves dynamics internal to the climate system. However, we have not identified the means by which the ice sheet is forcing the bulk of the low-level winds south of its southern margin. Describing the ice sheet as acting as a "barrier" is primarily diagnostic, rather than mechanistic.

Since we removed any discussion of an elevation threshold, we removed the sentences which were problematic for the reviewer.

*p22 l30 - "As long as ... northernmost position". It's not clear what you are trying to explain here. Is it changes to the Western Atlantic Jet?*

Yes, we are explaining why the jet distribution over eastern North America is strongly skewed, whereas the jet over the eastern Atlantic is distributed more symmetrically. Revised text is "The ice sheet only acts as a barrier to western region of the jet, which creates a strong wind shear along the northern edge of the jet there. This shear causes eddies to break along this boundary and accelerate the flow locally, which keeps the jet preferentially in its northernmost position. Since there is no such constraint on the jet in the eastern regions, it is free to vary equally in both directions around its most common position."

*p23 l24 - "we do detect at times" do you mean sometimes? "at times" is confusing because it suggests, or at least it did to me, that there are specific times at which this occurs, rather than the more general "on occasion this occurs" which is the sense that you mean I think.*

Revised text is "Regardless, since the jet undergoes abrupt changes over the deglaciation, we can not discount the possibility that the jet set the stage for such events to occur."

*p23 l33 - "when the jet ... to the next". I don't understand this sentence.*

We have added additional text to clarify our argument here. Revised text is "However, as discussed in Section 3.2, we can distinguish between gradual latitudinal changes where the jet evenly splits its time between two adjacent latitude bands while it transitions from one to the other and more abrupt changes where there is no such mixed state. Thus, we consider a jet change to be abrupt (given the decadal resolution of our jet diagnostics) when the most commonly-occupied jet position shifts from one grid cell to the next without an intermediate mixed state."

*p24 l17 - "we present ...", Don't understand what you mean here.*

We are discussing the fact that rather than derive statistics from the jet data to describe its distribution (e.g. its mean and standard deviation), we have plotted the distribution directly via the frequency plots. The frequency plots are equivalent to a succession of jet histograms. When the jet is very skewed, this becomes apparent in the frequency plots (e.g. in the FullyTrans plot of Figure 10 for the first 9kyr at least). Due to the high percentage of time that the jet spent in its most common position, if we had plotted a mean and standard deviation calculated from this data (which is not appropriate for non-Gaussian distributions), changes to the mean would likely have resembled those of the most common jet position. However, readers would likely have been under the mistaken impression that the jet varied equally north and south of this position.

Revised text is "Additionally, we do not assume a Gaussian distribution of jet characteristics which is the implicit assumption of studies that solely characterize the jet via its mean and standard deviation. Instead, through frequency plots, we present changes in time of the jet distribution itself by showing

the percentage of winter months that the jet spends at each latitude for every decade of simulation."

*p24 l23 - "mixes characteristics of ... model conditions". Odd sentence. Why not break it up into 2:*
*"mixes characteristics of the upstream and downstream regions. We find that the evolution of the jet*
*in these two regions is differently timed ...". As it is written it's not clear what the clause beginning*
*"which we find" relates to.*
Done. Revised text is "Interpreting the detected jet changes is complicated by the fact that the NAtl jet
calculated over longitudes 270E to 360E mixes characteristics of the upstream and downstream
regions. The evolution of the jet in these two regions differs in timing, abruptness and dependence on
background model conditions."

*p26 l4. I'm confused what this paragraph relates to. It seems to be rather tangential to the rest of this*
*section. It's important, but a reader is confused by what's happenning in the text at this point. Either*
*introduce what this paragraph is about at the start or attach to the end of the paragraph to which it*
*relates.*
Revised text is "Finally, given the constraint provided by the ice sheet on the western side of the jet, we
can imagine scenarios whereby abrupt jet changes could have occurred in this region in the past."

*p26 l14. You need to explain what the difference between a causal or enabling role is. I have no idea*
*what you mean by the ice sheet playing an enabling role.*

[revised manuscript text omitted]